Seawater pH reconstruction using boron isotopes in multiple planktonic foraminifera species with
different depth habitats and their potential to constrain pH and pCO$_2$ gradients

Maxence Guillermic[1,2], Sambuddha Misra[3,4], Robert Eagle[1,2], Alexandra Villa[2,5], Fengming Chang[6],
Aradhna Tripati [1,2]
[1] Department of Earth, Planetary, and Space Sciences, Department of Atmospheric and Oceanic
Sciences, Institute of the Environment and Sustainability, UCLA, University of California – Los
Angeles, Los Angeles, CA 90095 USA
[2] Laboratoire Géosciences Océan UMR6538, UBO, Institut Universitaire Européen de la Mer, Rue
Dumont d'Urville, 29280, Plouzané, France
[3] Indian Institute of Science, Centre for Earth Sciences, Bengaluru, Karnataka 560012, India
[4] The Godwin Laboratory for Palaeoclimate Research, Department of Earth Sciences, University of
Cambridge, UK
[5] Department of Geology, University of Wisconsin-Madison, Madison, WI 53706 USA
[6] Key Laboratory of Marine Geology and Environment, Institute of Oceanology, Chinese Academy of
Sciences, Qingdao 266071, China

Submitted to Biogeosciences

*Corresponding authors:
E-mail address: maxence.guillermic@gmail.com, atripati@g.ucla.edu

## ABSTRACT

Boron isotope systematics of planktonic foraminifera from core-top sediments and culture experiments have been studied to investigate the sensitivity of $\delta^{11}B$ of calcite tests to seawater pH. However, our knowledge of the relationship between $\delta^{11}B$ and pH remains incomplete for many taxa. Thus, to expand the potential scope of application of this proxy, we report $\delta^{11}B$ data for 7 different species of planktonic foraminifera from sediment core-tops. We utilize a method for the measurement of small samples of foraminifera and calculate the $\delta^{11}B$-calcite sensitivity to pH for *Globigerinoides ruber*, *Trilobus sacculifer* (sacc or w/o sacc), *Orbulina universa*, *Pulleniatina obliquiloculata*, *Neogloboquadrina dutertrei*, *Globorotalia menardii* and *Globorotalia tumida,* including for unstudied core-tops and species. These taxa have diverse ecological preferences and are from sites that span a range of oceanographic regimes, including some that are in regions of air-sea equilibrium and others that are out of equilibrium with the atmosphere. The sensitivity of $\delta^{11}B_{carbonate}$ to $\delta^{11}B_{borate}$ (eg. $\Delta\delta^{11}B_{carbonate}/\Delta\delta^{11}B_{borate}$) in core-tops is consistent with previous studies for *T. sacculifer* and *G. ruber* and close to unity for *N. dutertrei*, *O. universa* and combined deep-dwelling species. Deep-dwelling species closely follow the core-top calibration for *O. universa,* which is attributed to respiration-driven microenvironments likely caused by light limitation and/or symbiont/host interactions. Our data support the premise that utilizing boron isotope measurements of multiple species within a sediment core can be utilized to constrain vertical profiles of pH and $pCO_2$ at sites spanning different oceanic regimes, thereby constraining changes in vertical pH gradients and yielding insights into the past behavior of the oceanic carbon pumps.

## 1. Introduction

The oceans are absorbing a substantial fraction of anthropogenic carbon emissions resulting in declining surface ocean pH (IPCC, 2014). Yet there is a considerable uncertainty over the magnitude of future pH change in different parts of the ocean and the response of marine biogeochemical cycles to physio-chemical parameters (T, pH) caused by climate change (Bijma et al., 2002; Ries et al., 2009). Therefore, there is an increased interest in reconstructing past seawater pH (Hönisch and Hemming, 2004; Liu et al., 2009; Wei et al., 2009; Douville et al., 2010), in understanding spatial variability in aqueous pH and carbon dioxide ($pCO_2$) (Foster et al., 2008; Martinez-Boti et al., 2015b; Raitzsch et al., 2018), and in studying the response of the biological carbon pump using geochemical proxies (Yu et al., 2007, 2010, 2016).

Although all proxies for carbon cycle reconstruction are complex in nature (Pagani et al., 2005; Tripati et al., 2009, 2011; Allen and Hönisch, 2012), the boron isotope composition of foraminiferal tests (expressed as $\delta^{11}B_{carbonate}$) is emerging as one of the more robust available tools (Ni et al., 2007; Foster et al., 2008, 2012; Henehan et al., 2013; Martinez-Boti et al., 2015b; Chalk et al., 2017). The study of laboratory-cultured foraminifera has demonstrated a systematic dependence of the boron isotope composition of tests on solution pH (Sanyal et al., 1996, 2001; Henehan et al., 2013, 2016). Core-top measurements on globally distributed samples also show a boron isotope ratio sensitivity to pH with taxa-specific offsets from the theoretical fractionation line of borate ion (Rae et al., 2011; Henehan et al., 2016; Raitzsch et al., 2018).

Knowledge of seawater pH, in conjunction with constraints on one other carbonate system parameter (Total Alkalinity (TA), DIC (dissolved inorganic carbon), $[HCO_3^-]$, $[CO_3^{2-}]$), can be utilized to constrain aqueous $pCO_2$. Application of empirical calibrations for boron isotope ratio, determined for select species of foraminifera from core-tops and laboratory cultures, has resulted in accurate reconstructions of $pCO_2$ utilizing downcore samples from sites that are currently in quasi-equilibrium with the atmosphere at present. Values of $pCO_2$ reconstructed from planktonic foraminifera boron isotope ratios are analytically indistinguishable from ice core $CO_2$ records (Foster et al., 2008; Henehan et al., 2013; Chalk et al., 2017).

The last decade has produced several studies aiming at reconstructing past seawater pH using boron isotopes to constrain atmospheric $pCO_2$ in order to understand the changes in the global carbon cycle (Hönisch et al., 2005, 2009; Foster et al., 2008, 2012, 2014; Seki et al., 2010; Bartoli et al., 2011; Henehan et al., 2013; Martinez-Boti et al., 2015a, 2015b; Chalk et al., 2017). In addition to reconstructing atmospheric pCO2, the boron isotopes proxy has been applied to mixed-layer planktonic foraminifera at sites out of equilibrium with the atmosphere to constrain past air-sea fluxes (Foster et al., 2014; Martinez-Boti et al., 2015b). A small body of work has examined whether data for multiple species in core-top (Foster et al., 2008) and down-core samples could be used to constrain vertical profiles of pH through time (Palmer et al., 1998; Pearson and Palmer, 1999; Anagnostou et al., 2016).

Here we add to the emerging pool of boron isotope data in planktonic foraminifera from different oceanographic regimes, including data for species that have not previously been examined. We utilize a low-blank (15 pg B to 65 pg B), high precision (2sd on the international standard JCp-1 is 0.20 ‰, n=6) $\delta^{11}B_{carbonate}$ analysis method for small samples (down to ~250 µg $CaCO_3$), modified after Misra et al. (2014), to study multiple species of planktonic foraminifera. The studied sediment core-tops span a range of oceanographic regimes, including open-ocean oligotrophic settings and marginal seas. We constrain calibrations for different species, and compare results to published work (Foster et al., 2008; Henehan et al., 2013; Henehan et al., 2016; Martinez-Boti et al., 2015b;

Raitzsch et al., 2018). We also test whether these data support the application of boron isotope measurements of
multiple species within a sediment core as a proxy for constraining vertical profiles of pH and $pCO_2$.

## 2. Background

### 2.1 Planktonic foraminifera as archives of seawater pH

Planktonic foraminifera are used as archives of past environmental conditions within the mixed layer and

thermocline, as their chemical composition is correlated with the physio-chemical parameters of their calcification
environment (Ravelo and Fairbanks, 1992; Elderfield and Ganssen, 2000; Dekens et al., 2002; Anand et al., 2003;
Sanyal et al., 2001; Ni et al., 2007; Henehan et al., 2013, 2015, 2016; Howes et al., 2017; Raitzch et al., 2018).
The utilization of geochemical data for multiple planktonic foraminifera species with different ecological
preferences to constrain vertical gradients has been explored in several studies. The framework for such an
approach was first developed using modern samples of planktonic foraminifera for oxygen isotopes, where it was
proposed as a tool to constrain vertical temperature gradients and study physical oceanographic conditions during
periods of calcification (Ravelo and Fairbanks, 1992).

Because planktonic foraminifera species complete their lifecycle in a particular depth habitat due to their

ecological preference (Ravelo and Fairbanks, 1992; Farmer et al., 2007), it is theoretically possible to reconstruct
water column profiles of pH using boron isotope ratios data from multiple taxa (Palmer and Pearson, 1998;
Anagnostou et al., 2016). The potential use of an analogous approach to reconstruct past profiles of seawater pH
was first highlighted by Palmer and Pearson (1998) on Eocene samples to constrain pH-depth gradients. However,
in these boron isotope-based studies, it was assumed that boron isotope offset from seawater and foraminiferal
carbonate were constant, which is an assumption not supported by subsequent studies (e.g., Hönisch et al., 2003;
Foster et al., 2008; Henehan et al., 2013, 2016; Raitszch et al., 2018; Rae, 2018). Furthermore, boron isotope ratio
differences between foraminifera species inhabiting waters of the same pH makes the acquisition of more core-
top and culture data essential for applications of the proxy.

### 2.2 Boron systematics in seawater

Boron is a conservative element in seawater with a long residence time ($\tau_B \sim 14$ Myr) (Lemarchand et al.,

2002a). In seawater, boron exists as trigonal boric acid $B(OH)_3$ and tetrahedral borate ion $B(OH)_4^-$ (borate). The
relative abundance of boric acid and borate ion is a function of the ambient seawater pH. At standard open ocean
conditions (T = 25 °C and S = 35), the dissociation constant of boric acid is 8.60 (Dickson, 1990), implying that
boron mainly exists in the form of boric acid in seawater. Since the $pK_B$ and seawater pH (e.g., ~8.1, NBS) values
are similar, it implies that small changes in seawater pH will induce strong variations in the abundance of the two
boron species (Fig. 1).

Boron has two stable isotopes, $^{10}B$ and $^{11}B$, with average relative abundances of 19.9 and 80.1 %,

respectively. Variations in B isotope ratio are expressed in conventional delta ($\delta$) notation:

$$\delta^{11}B\ (‰) = 1000\ \text{x} \left( \frac{^{11}B/^{10}B_{Sample}}{^{11}B/^{10}B_{NIST\ SRM\ 951}} - 1 \right) \tag{1}$$


where positive values represent enrichment in the heavy isotope [11]B, and negative values enrichment in the light
isotope [10]B, relative to the standard reference material. Boron isotope values are reported versus the NIST SRM
951 boric acid standard (Cantazaro et al., 1970).
$B(OH)_3$ is enriched in [11]B compared to $B(OH)_4^-$ with a constant offset between the two chemical
species, within the range of physio-chemical variation observed in seawater, given by the fraction factor ($\alpha$). The
fractionation ($\varepsilon$) between $B(OH)_3$ and $B(OH)_4^-$ of $27.2 \pm 0.6$ ‰ has been empirically determined by Klochko et
al. (2006) in seawater. Note, Nir et al. (2015) calculate this fractionation, using an independent method, to be 26
$\pm 1$ ‰, which is within the analytical uncertainty of the Klochko et al. (2006) value. We use a fractionation of 27.2
‰ determined by Klochko et al. (2006) in this study.

**2.3 Boron isotopes in planktonic foraminifera calcite**
Many biogenic carbonate-based geochemical proxies are affected by "vital effects" or biological
fractionations (Urey et al., 1951). The $\delta^{11}B_{carbonate}$ in foraminifera exhibits species-specific offsets (see Rae et al.,
2018 for review) compared to theoretical predictions for the boron isotopic composition of $B(OH)_4^-$ (expressed as
$\delta^{11}B_{borate}$, $\alpha=1.0272$, Klochko et al., 2006). As the analytical and technical aspects of boron isotope measurements
have improved (Foster et al., 2008; Rae et al., 2011; Misra et al., 2014; Lloyd et al., 2018), evidence for taxonomic
differences have not been eliminated, but have become increasingly apparent (Foster et al., 2008, 2018; Henehan
et al 2013, 2016; Foster et al., 2016; Rae et al., 2018; Raitzsch et al., 2018).
At present, culture and core-top calibrations have been published for several planktonic species including
*Trilobatus sacculifer, Globigerinoides ruber, Globigerina bulloides, Neogloboquadrina pachyderma, Orbulina*
*universa* (Foster et al., 2008; Henehan et al., 2013; Henehan et al., 2015; Sanyal et al., 1996; Sanyal et al., 2001).
Although the boron isotopic composition of several species of foraminifera is now commonly used for
reconstructing surface seawater pH, for other species, there is a lack of data constraining the sensitivity of boron
isotopes in foraminiferal carbonate and borate ion in seawater.

**2.4 Origin of biological fractionations in foraminifera**
Perforate foraminifera are calcifying organisms that maintain a large degree of biological control over
their calcification space, and thus, mechanisms of biomineralization may be of significant importance in
controlling the $\delta^{11}B$ of the biogenic calcite. The biomineralization of foraminifera is based on seawater
vacuolization (Erez, 2003; de Nooijer et al., 2014) with parcels of seawater being isolated by an organic matrix
thereby creating a vacuole filled with seawater. Recent work has also demonstrated that even if the chemical
composition of the reservoirs is modified by the organism, seawater is directly involved in the calcification process
with vacuoles formed at the periphery of the shell (de Nooijer et al., 2014). Culture experiments by Rollion-Bard
and Erez (2010) have proposed that the pH at the site of biomineralization is elevated to an upper pH limit of ~9
for the shallow-water, symbiont-bearing benthic foraminifera *Amphistegina lobifera*, which would support a pH
modulation of a calcifying fluid in foraminifera. The extent to which these results apply to planktonic foraminifera
is not known, although pH modulation of calcifying fluid may influence the $\delta^{11}B$ of planktonic foraminifera.
For taxa with symbionts, the microenvironment surrounding the foraminifera is chemically different from
seawater due to photosynthetic activity (Jorgensen et al., 1985; Rink et al., 1998; Köhler-Rink and Kühl, 2000).
Photosynthesis by symbionts elevates the pH of microenvironments (Jorgensen et al., 1985; Rink et al., 1998;
Wolf-Gladrow et al., 1999; Köhler-Rink and Kühl, 2000), while calcification and respiration decrease
microenvironment pH (Equation 2 and 3).
$$Ca^{2+} + 2HCO_3^- \leftrightarrow CaCO_3 + H_2O + CO_2 \text{ or } Ca^{2+} + CO_3^{2-} \leftrightarrow CaCO_3 \qquad \text{[calcification]} \quad (2)$$
$$CH_2O + O_2 \leftrightarrow CO_2 + H_2O \qquad\qquad\qquad \text{[respiration/photosynthesis]} \quad (3)$$

$\delta^{11}B$ in foraminifera is primary controlled by seawater pH, but also depends on the pH alteration of
microenvironments due to calcification, respiration and symbiont photosynthesis. $\delta^{11}B_{carbonate}$ should therefore
reflect the relative dominance of these processes and may account for species-specific $\delta^{11}B$ offsets. Theoretical
predictions from Zeebe et al. (2003) and foraminiferal data from Hönisch et al. (2003) explored the influence of
microenvironment pH in $\delta^{11}B$ signature of foraminifera. Their work also suggested that for a given species, there
should be a constant offset observed between the boron isotope composition of foraminifera and borate ion over a
large range of pH, imparting confidence in utilizing species-specific boron isotope data as a proxy for seawater
pH.
Comparison of boron isotope data for multiple planktonic foraminiferal species indicate that taxa with
high levels of symbiont activity such as *T. sacculifer* and *G. ruber* show higher $\delta^{11}B$ values than the $\delta^{11}B$ of ambient
borate (Foster et al., 2008, Henehan et al., 2013, Raitzsch et al., 2018). The sensitivities ($\Delta\delta^{11}B_{carbonate}/\Delta\delta^{11}B_{borate}$,
hereafter referred to as the slope) of existing calibrations suggest a different species-specific sensitivity for these
species compared to other taxa (Sanyal et al., 2001; Henehan et al., 2013; Henehan et al.,2015; Raitzsch et al.,
2018). For example, *Orbulina universa* exhibits a lower $\delta^{11}B$ than *in situ* $\delta^{11}B$ values of borate ion (Henehan et
al., 2016), consistent with the species living deeper in the water column characterized by reduced photosynthetic
activity.
It is possible that photosynthetic activity by symbionts might not be able to compensate for changes in
calcification and/or respiration, leading to an acidification of the microenvironment. It is interesting to note that
for *O. universa* the slope determined for the field-collected samples is not statistically different from unity (0.95 ±
0.17) (Henehan et al. 2016), while culture experiments report slopes of ≤ 1 for multiple species including *G. ruber*
(Henehan et al., 2013), *T. sacculifer* (Sanyal et al., 2001), and *O. universa* (Sanyal et al., 1999). More core-top and
culture calibrations are needed to refine those slopes and understand if significant differences are observed, which
is part of the motivation for this study.

**2.5 Planktic foraminifera depth and habitat preferences**

The preferred depth habitat of different species of planktonic foraminifera depends on their ecology,
which in turn is dependent on hydrographic conditions. For example, *G. ruber is* commonly found in the mixed
layer (Fairbanks and Wiebe, 1980; Dekens et al., 2002; Farmer et al., 2007) during the summer (Deuser et al.,
1981) whereas *T. sacculifer* is present in the mixed layer until mid-thermocline depths (Farmer et al., 2007) during
spring and summer (Deuser et al., 1981, 1989). Specimens of *P. obliquiloculata* and *N. dutertrei* are abundant
during winter months (Deuser et al., 1989), with an acme in the mixed layer (~60m) for *P. obliquiloculata* and at
mid-thermocline depths for *N. dutertrei* (Farmer et al., 2007). In contrast, *O. universa* tends to record annual
average conditions within the mixed layer. Specimens of *G. menardii* calcify within the seasonal thermocline
(Fairbanks et al., 1982, Farmer et al., 2007, Regenberg et al., 2009), and in some regions in the upper thermocline
(Farmer et al., 2007), and records annual temperatures. *G. tumida* is found at the lower thermocline or below the
thermocline and records annual average conditions (Fairbanks and Wiebe, 1980; Farmer et al., 2007, Birch et al.,
2013). Although the studies listed above showed evidence for species-specific living depth-habitat affinities, recent
direct observations showed that environmental conditions (e.g. temperature, light) was locally responsible for the
variability in the living depth of certain foraminifera species in the eastern North Atlantic (Rebotim et al., 2017).
**3. Materials and Methods**
**3.1 Localities studied**
Core-top locations were selected to span a broad range of seawater pH, carbonate system parameters, and
oceanic regimes. Samples from Atlantic Ocean (CD107-A), Indian Ocean (FC-01a and FC-02a), Arabian Sea
(FC-13a and FC-12b) and Pacific Ocean (WP07-01, A14, and Ocean Drilling Program 806A and 807A) were
analyzed; characteristics of the sites are summarized in Table 1 and S7, Fig. 2, and Fig. 3.
Atlantic site CD107-a (CD107 site A) was cored in 1997 by the Benthic Boundary Layer program
(BENBO) (K.S. Black et al., 1997 - cruise report RRS Charles Darwin Cruise 107). Arabian Sea sites FC-12b
(CD145 A150) and FC-13a (CD145 A3200) were retrieved by the *Charles Darwin* in the Pakistan Margin in 2004
(B.J. Bett et al., 2003 - cruise report n°50 RRS Charles Darwin Cruise 145). A14 was recovered by box corer in
the southern area of the South China Sea in 2012. Core WP07-01 was obtained from the Ontong Java Plateau using
a giant piston corer during the Warm Pool Subject Cruise in 1993. Holes 806A and 807A were retrieved on Leg
130 by the Ocean Drilling Program (ODP). The top 10 cm of sediment from CD107-A have been radiocarbon
dated to be Holocene <3 ky (Thomson et al., 2000). Samples from multiple box cores from Indian Ocean sites
were radiocarbon dated as Holocene <7.3 ky (Wilson et al., 2012). Samples from western equatorial Pacific Site
806B, close to site WP07-01, are dated to between 7.3-8.6 ky (Lea et al., 2000). Arabian Sea and Pacific core-top
samples were not radiocarbon dated but are assumed to be Holocene.
**3.2 Species**
Around 50-100 foraminifera shells were picked from the 400-500 µm fraction size for *Globorotalia*
*menardii* and *Globorotalia tumida*, >500 µm for *Orbulina universa*, and from the 250-400 µm fraction size for
*Trilobatus sacculifer* (w/o sacc, without sacc-like final chamber), *Trilobatus sacculifer* (sacc, sacc-like final
chamber), *Globigerinoides ruber (*white, sensu stricto), *Neogloboquadrina dutertrei,* and *Pulleniatina*
*obliquiloculata*. The samples picked for analyses were visually well preserved.
**3.3 Sample cleaning**
Briefly, picked foraminifera were gently cracked open, clay removed with successive ultrasonication
steps in MQ water and methanol and then were checked for coarse-grained silicates. The next stages of sample
processing and chemical separation were performed in a class 1000 clean lab equipped with boron-free HEPA
filters. Samples were cleaned using full reductive and oxidative cleaning (Boyle, 1981; Boyle and Keigwin, 1985;
Barker et al., 2003). Samples from the South China Sea (sites A14, E035) presented high Mn and high Fe. Due to
potential Fe-Mn oxide and hydroxides the reductive cleaning was used. Previous comparisons of cleaning methods
have shown there is no impact of the reductive step on B/Ca (Misra et al., 2014b) but there is an impact of the
reductive step on Mg/Ca (Barker et al., 2003 and others), nevertheless, it is possible that Fe-Mn oxide and

hydroxides can result in non-negligible Mg and B contamination. Because this study was designed to investigate boron proxies and in order to be consistent in methodology, the reductive cleaning was used at all sites. Cleaned samples selected for this study did not yield high Mn concentrations (see supplement for discussion on contamination).

A final leaching step with 0.001N HCl was done before dissolution in 1N HCl. Hydrochloric acid was used to allow complete dissolution of the sample including Fe-Mn oxide and hydroxides if present. Each sample was divided into two aliquots: an aliquot for boron purification and one aliquot for trace element analysis.

### 3.4 Reagents

Double-distilled $HNO_3$ and HCl acids (from Merck® grade) and a commercial bottle of HF Ultrapure grade were used at Brest. Double-distilled acids were used at Cambridge. All acids and further dilutions were prepared using double-distilled 18.2 $M\Omega.cm^{-1}$ MQ water. Working standards for isotope ratio and trace element measurements were freshly diluted on a daily basis with the same acids used for sample preparation to avoid any matrix effects.

### 3.5 Boron isotopes

Boron purification for isotopic measurement was done utilizing microdistillation method developed by Gaillardet et al. (2001), for Ca-rich matrices by Wang et al. (2010) and adapted at Cambridge by Misra et al. (2014a). 70 µL of carbonate sample dissolved in 1N HCl was loaded on a cap of a clean fin legged 5 mL conical beaker upside down. The tightly closed beaker was put on a hotplate at 95°C for 15 hours. The beakers were taken off the hotplate and were allowed to cool for 15 min. The cap where the residue formed was replaced by a clean one. Then, 100 µL of 0.5% HF were added to the distillate.

Boron isotopic measurements were carried out on a Thermo Scientific ®Neptune+ MC-ICP-MS at the University of Cambridge. Neptune+ was equipped with Jet interface and two $10^{13}$ $\Omega$ resistors. The instrumental setup included Savillex® 50µl/min C-flow self-aspirating nebulizer, single pass Teflon® Scott-type spray chamber constructed utilizing Savillex® column components, 2.0 mm Pt injector from ESI®, Thermo® Ni 'normal' type sample cone and 'X' type skimmer cones. Both isotopes of boron were determined utilizing $10^{13}$ $\Omega$ resistors (Misra et al., 2014a; Lloyd et al., 2018).

The sample size for boron isotope analyses typically ranged from 10 ppb B (~5 ng B) to 20 ppb B samples (~10 ng B). Instrumental sensitivity for [11]B was 17 mV/ppb B (eg. 170 mV for 10ppb B) in wet plasma at 50µl/min sample aspiration rate. Intensity of [11]B for a sample at 10ppb B was typically 165mV $\pm$ 5mV, which closely matched the 170mV $\pm$ 5mV of the standard. Due to the low boron content of the samples extreme care was taken to avoid boron contamination during sample preparation and reduce memory effect during analysis. Procedural boron blanks ranged from 15pg B to 65 pg B and contributed to less than <1% of the sample signal. The acid blank during analyses was measured at $\leq$ 1mV on [11]B, meaning a contribution < 1% of the sample intensity, no memory effect was observed within and across sessions. No matrix effect resulting from the mix HCl/HF was observed on the $\delta^{11}$B.

Analyses of external standards were done to ensure data quality. For $\delta^{11}$B measurements one carbonate standard and one coral were utilized: the JCp-1 (Geological Survey of Japan, Tsukuba, Japan) international standard (Gutjahr et al., 2014) and the NEP coral (Porites sp., $\delta^{11}$B = 26.12 $\pm$ 0.92 ‰, 2SD, n=33 Holcomb et al.,

2015 and Sutton et al., 2018, Table S2) from University of Western Australia/Australian National University. A certified boric acid standard, the ERM© AE121 ($\delta^{11}B$ = 19.9 ± 0.6 ‰, SD, certified) was used to monitor reproducibility and drift during each session (Vogl and Rosner, 2011; Foster et al., 2013; Misra et al., 2014). Results for the isotopic composition of the NEP coral are shown in Table S2, average values are $\delta^{11}B_{NEP}$ = 25.70 ± 0.93 ‰ (2SD, n=22) over different 7 analytical sessions with each number representing an ab-initio processed sample. Our results are within error of published values of 26.20 ± 0.88 ‰ (2SD, n = 27) and 25.80 ± 0.89 ‰ (2SD, n = 6) by Holcomb et al. (2015) and Sutton et al. (2018) respectively. Chemically cleaned JCp-1 samples were measured at 24.06 ± 0.20 (2SD, n=6) and is within error of published values of 24.37 ± 0.32 ‰, 24.11± 0.43 ‰ and 24.42 ± 0.28 ‰ by Holcomb et al. (2015), Farmer et al. (2016) and Sutton et al. (2018) respectively.

### 3.6 Trace elements

The calcium concentration of each sample was measured on an ICP-AES ® Ultima 2 HORIBA at the Pôle spectrometrie Océan (PSO), UMR6538 (Plouzané, France). Samples were then diluted to fixed calcium concentrations (typically 10 ppm or 30 ppm Ca) using 0.1 M $HNO_3$ & 0.3 M HF matching multi-element standards Ca concentration to avoid any matrix effects (Misra et al., 2014b). Levels of remaining HCl (<1%) in these diluted samples were negligible and did not contribute to matrix effects. Trace elements (e.g. X/Ca ratios) were analyzed on a Thermo Scientific ® Element XR HR-ICP-MS at the PSO, Ifremer (Plouzané, France).

Trace element analyses were done at a Ca concentration of 10 or 30 ppm. The typical blanks for a 30 ppm Ca session were: $^7Li$ < 2%, $^{11}B$ < 7%, $^{25}Mg$ < 0.2% and $^{43}Ca$ < 0.02%. Additionally, blanks for a 10 ppm Ca session were: $^7Li$ < 2.5%, $^{11}B$ < 10%, $^{25}Mg$ < 0.4% and $^{43}Ca$ < 0.05%. Due to strong memory effect for boron and instrumental drift on the Element XR, long sessions of conditioning were done prior analyses. Boron blanks were driven below 5% of signal intensity usually after 4 to 5 days of continuous analyses of carbonate samples. External reproducibility was determined on the consistency standard Cam-Wuellestorfi (courtesy of the University of Cambridge) (Misra et al., 2014b), Table S3. Our X/Ca ratio measurements on the external standard Cam-Wuellestorfi were all the time within error of the published value (Table S3) validating the robustness of our trace elements data. Analytical uncertainty of a single measurement was calculated from the reproducibility of the Cam-Wuellestorfi, measured during a particular mass spectrometry session. The analytical uncertainties (2SD, n=31, Table S3) on the X/Ca ratios are: ±0.4 µmol/mol for Li/Ca, ±7 µmol/mol for B/Ca and ±0.01 mmol/mol for Mg/Ca respectively.

### 3.7 Oxygen isotopes

Carbonate $\delta^{13}C$ and $\delta^{18}O$ were measured on a Gas Bench II coupled to a Delta V mass spectrometer at the stable isotope facility of Pôle spectrometrie Océan (PSO), Plouzané. Around 20 shells were weighed, crushed and clay removed following the same method described in section 3.3 (Barker et al., 2003). The recovered foraminifera were weighed in tubes and flushed with He gas. Samples were then digested in phosphoric acid and analyzed. Results were calibrated to the VPDB scale by international standard NBS19 and analytical precision on the in-house standard Ca21 was better than ±0.11‰ for $\delta^{18}O$ (1SD, n=5) and ±0.03‰ for $\delta^{13}C$ (1SD, n=5).

### 3.8 Calcification depth determination

We utilized two different chemo-stratigraphic methods to estimate the calcification depth (CD) in this study (Table S6 and S7). The first method (CD1), commonly used in paleoceanography, utilizes $\delta^{18}O$ measurements of the carbonate ($\delta^{18}O_c$) to estimate calcification depths (referred to as $\delta^{18}O$-based calcification depths) (Schmidt et al., 2002; Mortyn et al., 2003; Sime et al., 2005; Farmer et al., 2007; Birsh et al., 2013). Rebotim et al. (2017) also showed good correspondence between living depth habitat and calcification depth derived using CD1. The second method (CD2) utilizes Mg/Ca-based temperature estimates ($T_{Mg/Ca}$) to constrain calcification depths (Quintana Krupinski et al., 2017). However, we note that reductive cleaning leads to a decrease in Mg/Ca that in turn would result in a bias towards deeper calcification depths, which is not the case when we utilize non-Mg/Ca-based methodologies. In both cases, the prerequisite was that vertical profiles of seawater temperature are available for different seasons in ocean atlases and cruise reports, and that hydrographic data and geochemical proxy signatures can be compared to assess the depth in the water column that represents the taxon's maximum abundance.

Because both methods have their uncertainties (in one case, use of taxon-specific calibrations, and in the other, analytical limitations), both estimates of calcification depth were compared to published values for the basin (CD3), and where available, for the same site (Table S6). To select which calcification depth to use for further calculations, we first looked at $CD_1$, $CD_2$ and $CD_3$. If, CD1 and CD2 were similar we selected this calcification depth, if $CD_1$ and $CD_2$ were different we chose literature values, $CD_3$, when available. For some less studied species, like *G. tumida*, *G. menardii* or *P. obliquiloculata*, $CD_3$ was not always available but when available showed good correspondence with our $CD_2$, moreover due to availability of Mg/Ca-temperature taxon-specific calibrations we preferentially use $CD_2$ for those species.

We applied (based on uncertainties of our measurements) an uncertainty of ±10m for calcification depths > 70 m and an uncertainty of ± 20 m when calcification depths < 70 m. Direct observations of living depths of foraminifera remain limited. However, the depth uncertainties reported here are in line with the uncertainties calculated based on direct observations in the eastern North Atlantic which give a standard error on average living depths ranging from 6-22 m for the same species (Rebotim et al., 2017). The decrease in Mg/Ca due to reductive cleaning was not taking into account, because it has not been studied for most of the species used in this study and because the depth uncertainty applied based on $\delta^{18}O$ analytical error is conservative relative to the uncertainty of a 10% decrease in Mg/Ca equivalent that would be equivalent to ~1.2°C. The depth habitats utilized to derive *in situ* parameters are summarized in Table S7.

### 3.9 $\delta^{11}B_{borate}$

Two carbonate system parameters are needed to fully constrain the carbonate system. Following the approach of Foster et al., (2008) we used the GLODAP database (Key et al., 2004) corrected for anthropogenic inputs in order to estimate pre-industrial carbonate system parameters at each site. Temperature, salinity and pressure for each site are from the World Ocean Database 2013 (Boyer et al., 2013). We utilized the R$^{©}$ code in Henehan et al, (2016) (courtesy of Michael Henehan) to calculate the $\delta^{11}B_{borate}$, $\delta^{11}B_{borate}$ uncertainty and derive our calibrations. Uncertainty for $\delta^{11}B_{borate}$ utilizing Henehan's code was similar to uncertainty calculated by applying 2 standard deviations of the $\delta^{11}B_{borate}$ profiles within the limits imposed by our calcification depth.

The Matlab[©] template provided by Zeebe and Wolf-Gladow, (2001) was used to calculate $pCO_2$ from

TA; temperature, salinity and pressure were included in the calculations. Total boron was calculated from Lee et
al., (2010), $K_1$ and $K_2$ were calculated from Mehrbach et al. (1973) refitted by Dickson and Millero (1987).

Statistical tests were made utilizing GraphPad[©] software, linear regressions for calibration where derived

utilizing R[©] code in Henehan et al, (2016) (courtesy of Michael Henehan) with k (number of wild bootstrap
replicates) equal to 500.

**4. Results**

**4.1 Depth habitat**

The calcification depths utilized in this paper are summarized in Tables S6 and S7, including a comparison

of calcification depth determination methods. The calculated calcification depths are consistent with the ecology
of each species and the physical properties of the water column of the sites. Specimens of *G. ruber* and *T. sacculifer*
appear to be living in the shallow mixed layer (0-100 m), with *T. sacculifer* living or migrating deeper than *G.*
*ruber* (down to 125 m). Specimens of *O. universa* and *P. obliquiloculata* are living in the upper thermocline; *G.*
*menardii* is found in the upper thermocline until the thermocline depth specific to the location; *N. dutertrei* is living
near thermocline depths and *G. tumida* is found in the lower thermocline.

Data from the multiple approaches for calculating calcification depth (CD1, CD2 and CD3)  implies that

some species inhabit deeper environments in the Western Equatorial Pacific (WEP) relative to the Arabian Sea,
which in turn are deeper-dwelling than the same morpho-species occurring in the Indian Ocean. In some cases, we
find evidence for differences in habitat depth of up to ~100m between the WEP and the Arabian Sea. This trend is
observed for *G. ruber* and *T. sacculifer*, but not for *O. universa*.

Some differences are observed between the two methods for calcification depth determination that are

based on $\delta^{18}O$ and Mg/Ca (CD1 and CD2, respectively). These differences might be due to the choice of
calibration. Alternatively, our uncertainties for $\delta^{18}O$ implies larger uncertainties on calcification depth
determinations that use this approach, compared to Mg/Ca based estimates.

**4.2 Empirical calibrations of foraminiferal $\delta^{11}B_{carbonate}$ to $\delta^{11}B_{borate}$**

Results for the different species analyzed in this study are presented in Fig. 4, Fig. 5 and summarized in

Table 2; additionally, published calibrations for comparison are summarized in Table 3.

**4.2.1 *G. ruber***

Samples were picked from the 250-300 µm fraction, except for the WEP sites where *G. ruber* shells were

picked from the 250-400 µm fraction. Weight per shell averaged $11 \pm 4$ µg (n=4, SD) although the weight was not
measured on the same sub-sample analyzed for $\delta^{11}B$ and trace elements or at the WEP sites. In comparison to
literature, the size fraction used for this study was smaller: Foster et al. (2008) used the 300-355µm fraction,
Henehan et al. (2013) utilized multiple size fractions (250-300, 250-355, 300-355, 355-400 and 400-455 µm) and
Raitzsch et al. (2018) used the 315-355 µm fraction.

Our results for *G. ruber* (Fig. 4) are in close agreement with published data from other core-tops, sediment

traps, tows, and culture experiments for $\delta^{11}B_{borate}$>19 ‰ (Foster et al., 2008, Henehan et al., 2013, Raitzsch et al.,
2018). However, the two datapoints from $\delta^{11}B_{borate}$<19 ‰ are lower compared to previous studies. Elevated
$\delta^{11}B_{carbonate}$ values relative to $\delta^{11}B_{borate}$ has been explained by the high photosynthetic activity of symbionts
(Hönisch et al., 2003; Zeebe et al., 2003). Three calibrations have been derived (Table 3). Linear regression on our
data alone yields a slope of 1.12 (±1.67). The uncertainty is significant given limited data in our study, and given
this large uncertainty, our sensitivity of $\delta^{11}B_{carbonate}$ to $\delta^{11}B_{borate}$ is also consistent with the low sensitivity trend of
culture experiments from Sanyal et al. (2001) or Henehan et al. (2013). The second calibration made compiling all
data from literature shows a sensitivity similar (e.g. 0.46 (±0.34)) to the one recently published by Raitzsch et al.,
(2018) (e.g. 0.45 (±0.16), Table 3). The third linear regression made only on data from the 250-400 µm fraction
from our study and from the 250-300 µm from Henehan et al. (2013) yields a sensitivity of 0.58 (±0.91) similar to
culture experiments from Henehan et al., (2013) (e.g. 0.6 (±0.16), Table 3). This third calibration is offset by ~ -
0.4 ‰ (p>0.05) compared to culture calibration from Henehan et al. (2013).

**4.2.2 *T. sacculifer***

$\delta^{11}B_{carbonate}$ results for *T. sacculifer* (sacc and w/o sacc) (Fig. 4) are compared to published data (Foster et

al., 2008; Martinez-Boti et al., 2015b, Raitzsch et al., 2018). Results for *T. sacculifer* are in good agreement with
the literature and exhibit higher $\delta^{11}B_{carbonate}$ compared to expected $\delta^{11}B_{borate}$ at their collection location. A linear
regression through our data alone yields a slope of 1.3 ± 0.2 but is not statistically different to the results from
Martinez-Boti et al. (2015b) (Table 3), (p>0.05). However, when compiled with published data using the bootstrap
method a slope of 0.83 ± 0.48 is calculated, with a large uncertainty given the variability in the data. It is also
noticeable that *T. sacculifer* (w/o sacc) samples from the WEP have a $\delta^{11}B_{carbonate}$ close to expected $\delta^{11}B_{borate}$ and
are significantly lower compared to the combined *T. sacculifer* of other sites (p=0.01, unpaired t-test). When
regressing data from the 250-400 µm fraction, our results are not significantly different from the regression through
data that combine all size fractions (Fig. 4).

**4.2.3 *O. universa* and deeper-dwelling species: *N. dutertrei*, *P. obliquiloculata*, *G. menardii* and *G. tumida***

Our results for *O. universa* (Fig. 4)*, N. dutertrei, P. obliquiloculata, G. menardii* and *G. tumida* (Fig. 5)

exhibit lower $\delta^{11}B_{carbonate}$ compared to the expected $\delta^{11}B_{borate}$ at their collection location. These data for *O. universa*
are not statistically different from the Henehan et al. (2016) calibration (p>0.05). Our results for *N. dutertrei*
expand upon the initial measurements presented in Foster et al. (2008). The different environments experienced
by *N. dutertrei* in our study permit us to extend the range and derive a calibration for this species; the slope is close
to unity (0.93 ± 0.55), and is not significantly different (p>0.05) from the *O. universa* calibration previously
reported by Henehan et al. (2016) (e.g. 0.95 ± 0.17). The data for *P. obliquiloculata* exhibits the largest offset from
the theoretical line. The range of $\delta^{11}B_{borate}$ from the samples we have of *G. menardii* and *G. tumida* is not sufficient
to derive calibrations, but the $\delta^{11}B_{carbonate}$ measured for those species are in good agreement with the *N. dutertrei*
calibration and Henehan et al. (2016) calibration for *O. universa*.

For *O. universa* and all deep-dwelling species, the slopes are not statistically different from Henehan et

al. (2016) (p>0.05) and are close to unity. If data for deep-dwelling foraminiferal species are pooled together with
each other and with data from Henehan et al. (2016) and Raitzch et al. (2018), we calculate a slope of 0.95 (± 0.13)
($R^2$=0.7987, p<0.0001); if only our data are used, we calculate a slope that is not significantly different (0.82 ±
0.27; p<0.05).

**4.2.4 Comparison of core-top and culture data**

The data for *G. ruber* and *T. sacculifer* from the core-tops we measured are broadly consistent with previous published results. The calibrations between these core-top derived estimates and culture experiments are not statistically different due to small datasets and uncertainties on the linear regressions (Henehan et al., 2013; Marinez-Boti et al., 2015; Raitzsch et al., 2018; Table 3). The sensitivities of the species analyzed are not statistically different and are close to unity.

**4.3 B/Ca ratios**

B/Ca ratios are presented in Table 2 and Fig. 6. B/Ca data are species-specific and consistent with previous work (e.g., compiled in Henehan et al., 2016) with ratios higher for *G. ruber > T. sacculifer* (sacc) *> T. sacculifer* (w/o sacc) *> P. obliquiloculata > O. universa > > G. menardii > N. dutertrei > G. tumida > G. inflata > N. pachyderma > G. bulloides* (Fig. 6). This study supports species-specific B/Ca ratios as previously published (Yu et al., 2007; Tripati et al., 2009, 2011; Allen and Hönisch, 2012; Henehan et al., 2016). Differences between surface- and deep-dwelling foraminifera are observed, with lower values and a smaller range for the deeper-dwelling taxa (58-126 µmol/mol vs 83-190 µmol/mol for shallow dwellers), however, the trend for the surface-dwellers can also be driven by interspecies B/Ca variability. The B/Ca data for deep-dwelling taxa exhibits a significant correlation with $[B(OH)_4^-]/[HCO_3^-]$ ($p<0.05$), but no correlation with $\delta^{11}B_{carbonate}$ and temperature (Fig. S3). Surface-dwelling species have B/Ca ratios that exhibit significant correlations with $[B(OH)_4^-]/[HCO_3^-]$, $\delta^{11}B_{carbonate}$ and temperature. The sensitivity of B/Ca to $[B(OH)_4^-]/[HCO_3^-]$ is lower for deep-dwelling species compared to surface dwelling species. When all the B/Ca data are compiled, significant trends are observed with $[B(OH)_4^-]/[HCO_3^-]$, $\delta^{11}B_{carbonate}$ and temperature (Fig. S3). When comparing data from all sites together, a weak decrease in B/Ca with increasing calcification depth is observed ($R^2=0.11$, $p<0.05$, Fig. S4). A correlation also exists between B/Ca and the water depths of the cores (not significant, Fig. S4).

**5. Discussion**

**5.1 Sources of uncertainty relating to depth habitat and seasonality at studied sites**

**5.1.1 Depth habitats and $\delta^{11}B_{borate}$**

Because foraminifera will record ambient environmental conditions during calcification, the accurate characterization of *in situ* data is needed not only for calibrations, but also to understand the reconstructed record of pH or $pCO_2$. The species we examined are ordered here from shallower to deeper depth habitats: *G. ruber >T. sacculifer* (sacc) *> T. sacculifer* (w/o sacc) *> O. universa > P. obliquiloculata > G. menardii > N. dutertrei > G. tumida* (this study; Birch et al., 2013; Farmer et al., 2007), although the specific water depth will vary depending on the physical properties of the water column of the site (Kemle-von Mücke and Oberhänsli, 1999). We note that calculation of absolute calcification depths can be challenging in some cases as many species often transition to deeper waters at the end of their life cycle prior to gametogenesis (Steinhardt et al., 2015).

We find that assumptions about the specific depth habitat a species of foraminifera is calcifying over, in a given region, can lead to differences of a few per mil in calculated isotopic compositions of borate (Fig. 3).

Hence this can cause a bias in calibrations if calcification depths are assumed instead of being calculated (i.e., with
$\delta^{18}O$ and/or Mg/Ca). Factors including variations in thermocline depth can impact depth habitats for some taxa.
At the sites we examined, most of the sampled species live in deeper depth habitats in the WEP relative to the
Indian Ocean, which in turn is characterized by deeper depth habitats than in the Arabian Sea. In the tropical
Pacific, *T. sacculifer* is usually found deeper than *G. ruber* except at sites characterized by a shallow thermocline,
in which case both species tend to overlap their habitat (e.g., ODP Site 806 in the WEP which has a deeper
thermocline than at ODP Site 847 in the Eastern Equatorial Pacific; EEP) (Rickaby et al., 2005). The difference in
depth habitats for *T. sacculifer* and *N. dutertrei* between the WEP and EEP can be as much as almost 100 m
(Rickaby et al., 2005).

**5.1.2 Seasonality and *in situ* $\delta^{11}B_{borate}$**

As discussed by Raitzsch et al. (2018), depending of the study area, foraminiferal fluxes can change
throughout the year. Hydrographic parameters related to carbonate chemistry may change across seasons at a given
water depth. We therefore recalculated the theoretical $\delta^{11}B_{borate}$ using seasonal data for temperature and salinity
and annual values for TA and DIC for each depth at each site. The GLODAP (2013) database does not provide
seasonal TA or DIC values.

The low sensitivity of $\delta^{11}B_{borate}$ to temperature and salinity means that calculated $\delta^{11}B_{borate}$ for each water
depth at our sites were not strongly impacted (Fig. S1). Thus, these findings support Raitzsch et al. (2018), who
concluded that calculated $\delta^{11}B_{borate}$ values corrected for seasonality was within error of non-corrected values for
each water depth. As Raitzsch et al. (2018) highlight, seasonality might be more important at high latitude sites
where seasonality is more marked, however, the seasonality of primary production will also be more tightly
constrained due to the seasonal progression of winter light limitation and intense vertical mixing and summer
nutrient limitation.

Data for our sites suggests that most $\delta^{11}B_{borate}$ variability we observe does not come from seasonality but
from the assumed water depths for calcification. With the exception of a few specific areas such as the Red Sea
(Henehan et al., 2016, Raitzsch et al., 2018), at most sites examined, seasonal $\delta^{11}B_{borate}$ at a fixed depth does not
vary by more than ~0.2‰. We conclude that seasonality has a relatively minor impact on the carbonate system
parameters at the sites we examined.

**5.2 $\delta^{11}B$, microenvironment pH and depth habitats**

It is common for planktonic foraminifera to have symbiotic relationships with algae (Gast and Caron,
2001; Shaked and de Vargas, 2006). The family Globigerinidae, including *G. ruber*, *T. sacculifer* and *O. universa*,
commonly have dinoflagellate algal symbionts (Anderson and Be, 1976; Spero, 1987). The families
Pulleniatinidae and Globorotaliidae (e.g. *P. obliquiloculata, G. menardii* and *G. tumida*) have chrysophyte algal
symbionts (Gastrich, 1988) and *N. dutertrei* hosts pelagophyte symbionts (Bird et al., 2018). The relationship
between the symbionts and the host is complex. Nevertheless, this symbiotic relationship provides energy
(Hallock, 1981b) and promotes calcification in foraminifera (Duguay, 1983; Erez et al., 1983) by providing
inorganic carbon to the host (Jorgensen et al., 1985).

There are several studies indicating that the $\delta^{11}B$ signatures in foraminiferal calcite reflect
microenvironment pH (Jorgensen et al., 1985; Rink et al., 1998; Köhler-Rink and Kühl, 2000, Hönisch et al., 2003;
Zeebe et el., 2003). Foraminifera with high photosynthetic activity and symbiont density, such as *G. ruber* and *T.*
*sacculifer,* are expected to have a microenvironment pH higher than ambient seawater, and a $\delta^{11}B_{carbonate}$ higher
than expected $\delta^{11}B_{borate}$, which is the case in our study and in previous studies (Foster et al., 2008, Henehan et al.,
2013, Raitzsch et al., 2018). We also observed in our study that *N. dutertrei*, *G. menardii*, *P. obliquiloculata* and
*G. tumida* record a lower pH than ambient seawater, with $\delta^{11}B_{carbonate}$ lower than expected $\delta^{11}B_{borate}$, and suggest
the results are consistent with lower photosynthetic activity compared to the mixed-layer dwelling species. These
observations, based on $\delta^{11}B_{carbonate}$ measurements, are in line with direct observations from Takagi et al. (2019)
that show dinoflagellate-bearing foraminifera (*G. ruber*, *T. sacculifer* and *O. universa*) tend to have a higher
symbiont density and photosynthesis activity while *P. obliquiloculata*, *G. menardii* and *N. dutertrei* have lower
symbiont density and *P. obliquiloculata*, *N. dutertrei* have the lowest photosynthetic activity. In the same study,
*P. obliquiloculata* exhibited minimum symbiont densities and levels of photosynthetic activity, which may explain
why *P. obliquiloculata* exhibited the lowest microenvironment pH as recorded by $\delta^{11}B$.

Based on the observations of Takagi et al. (2019), we can assume that the low $\delta^{11}B$ of *O. universa* and *T.*

*sacculifer (w/o sacc)* from the WEP is explained by low photosynthetic activity. It has been shown for *T. sacculifer*
and *O. universa* that symbiont photosynthesis increases with higher insolation (Jorgensen et al., 1985; Rink et al.,
1998) and the photosynthetic activity is therefore a function of the light level the symbionts received. This is, in a
natural system, dependent on the depth of the species in the water column. For the purpose of this study, we do
not consider turbidity which also influences the light penetration in the water column. In this case,
photosynthetically-active foraminifera living close to the surface should record microenvironment pH (thus $\delta^{11}B$)
that is more sensitive to water depth changes. A deeper habitat reduces solar insolation, and as a consequence, may
lower symbiont photosynthetic activity, possibly reducing pH in the foraminifera's microenvironment. This is
supported by the significant trend observed between $\Delta^{11}B$ and the calcification depth for *G. ruber* and *T. sacculifer*
at our sites (Fig. S2), where microenvironment pH decreases with calcification depth. We observe a significant
decrease in $\delta^{11}B$ in the WEP for *T. sacculifer (w/o sacc)* compared to the other sites (p<0.05). Additionally, the
$\Delta^{11}B$ ($\Delta^{11}B = \delta^{11}B_{carbonate} - \delta^{11}B_{borate}$) of *G. ruber*, *T. sacculifer* (w/o sacc and sacc) is significantly lower in the
WEP compared to the other sites (p<0.05).

*T. sacculifer* has the potential to support more photosynthesis due to its higher symbiont density, and

higher photosynthetic activity compared to other species, which may support higher symbiont/host interactions
(Takagi et al., 2019). These results would be consistent with a greater sensitivity of *T. sacculifer*'s photosynthetic
activity with changes in insolation/water depth. To test if the low $\delta^{11}B$ signature of *T. sacculifer* (w/o sacc) in the
WEP is related to a decrease in light at greater water depth, we have independently calculated the calcification
depth of the foraminifera based on various light insolation culture experiments (Jorgensen et al., 1985) and the
microenvironment $\Delta$pH derived from our data (Fig. 7A and B). This exercise showed that the low $\delta^{11}B$ of *T.*
*sacculifer* (w/o sacc) from the WEP can be explained by the reduced light environment due to a deeper depth
habitat in the WEP (Fig. 7B). It can also be noted that *T. sacculifer* exhibits the largest variation in symbiont
density versus test size (Takagi et al., 2019), suggesting that lower size fraction reported for the WEP (250-400
µm) compared to the 300-400 µm at the other sites can be related to a decrease in photosynthetic activity and a
lower $\delta^{11}B$. Unfortunately, no weight per shell data were determined on foraminifera samples to constrain whether
test size was significantly different across sites. Future studies could use shell weights to test these relationships.

When the same approach of independently reconstructing calcification depth based on culture experiments is applied to *O. universa*, the boron data suggest a microenvironment pH of 0.10 to 0.20 lower than ambient seawater pH, which would be in line with the species living deeper than 50m (light compensation point (Ec), Rink et al., 1998), which is consistent with our calcification depth reconstructions. The low $\delta^{11}B_{carbonate}$ of *O. universa* compared to *T. sacculifer* for the similar calcification depth at some sites (e.g. FC-02a, WP07-a) might reflect differences in photosynthetic potential between the two species, which is supported by observation of a lower photosynthetic potential in *O. universa* than in *T. sacculifer* (Tagaki et al., 2019).

Microenvironment $\Delta$pH based on our $\delta^{11}B_{carbonate}$ data were calculated for the rest of the species. We observed that microenvironment $\Delta$pH is higher in *T. sacculifer > G. ruber > T. sacculifer* (w/o sacc - WEP) > *O. universa*, *N. dutertrei*, *G. menardii, G. tumida > P. obliquiloculata*. These results are in line with the photosymbiosis findings from Takagi et al., (2019). Also, the higher $\delta^{11}B$ data from the West African upwelling published by Raitzsch et al., (2018) for *G. ruber* and *O. universa* may reflect a higher microenvironment pH due to a relatively shallow habitat, higher insolation and high rates of photosynthesis by symbionts. This could highlight a potential issue with calibration when applied to sites with different oceanic regimes as the $\delta^{11}B$ species-specific calibrations could be also location-specific for the mixed dweller species.

Microenvironment pH for *N. dutertrei, G. menardii* and *G. tumida* are similar to *O. universa* and suggest a threshold for a respiration-driven $\delta^{11}B$ signature. This threshold can be induced by a change of photosynthetic activity at lower light intensity in deeper water and/or differences in symbiont density and/or by the type of symbionts at greater depth (non-dinoflagellate symbionts). We also note that *P. obliquiloculata,* which has the lowest symbiont density and photosynthetic activity (Takagi et al., 2019), has the lowest microenvironment pH compared to other deeper-dweller species, supporting our hypothesis that respiration can control microenvironment pH. The deep-dwelling species sensitivity of $\delta^{11}B_{carbonate}$ to $\delta^{11}B_{borate}$ with values close to unity might also be explained by a relatively stable respiration-driven microenvironments, as the deeper-dweller species do not experience large changes of insolation (e.g. photosynthesis), thereby making them a more direct recorder of environmental pH.

**5.3 $\delta^{11}B$ sensitivity to $\delta^{11}B_{borate}$ and relationship with B/Ca signatures**

In inorganic calcite, $\delta^{11}B_{carbonate}$ and B/Ca data have shown to be sensitive to precipitation rate with at higher precipitation rate increasing $\delta^{11}B_{carbonate}$ (Farmer et al., 2019) and B/Ca (Farmer et al., 2019; Gabitov et al., 2014; Kaczmarek et al., 2016; Mavromatis et al., 2015; Uchikawa et al., 2015). A recent study from Farmer et al, (2019) has proposed that in foraminifera at higher precipitation rates, more borate ion may be incorporated into the carbonate mineral, while more boric acid may be incorporated at lower precipitation rates. The authors also suggest this may explain low sensitivities of culture experiments.

When combining all literature data, *T. sacculifer* and *G. ruber* have sensitivities of $\delta^{11}B_{carbonate}$ to $\delta^{11}B_{borate}$ of $0.83 \pm 0.48$ and $0.46 \pm 0.34$ respectively in line with previous literature and paleo-$CO_2$ reconstructions. Also, if we only take into account our data, and the observation that the sensitivity of $\delta^{11}B_{carbonate}$ to $\delta^{11}B_{borate}$ is not statistically different from unity for most of the species investigated, we can speculate that for these taxa, changes in precipitation rate and contributions of boric acid are not likely to be important. If considering only the data from this study, *G. ruber* ($1.12 \pm 1.67$) and *T. sacculifer* ($1.38\pm 1.35$) present higher sensitivities of $\delta^{11}B_{carbonate}$ to $\delta^{11}B_{borate}$. We can then again speculate that the observed high values for $\delta^{11}B_{carbonate}$ at high seawater pH can be due

to higher precipitation rates. We note this could also be consistent with the higher sensitivity of B/Ca signatures
in these two surface dwelling species to ambient $[B(OH)_4^-]/[HCO_3^-]$ relative to deeper-dwelling species. Those
interspecific differences still remain to be explained, however, part of this variability is likely due to changes in
the carbonate chemistry of the microenvironment resulting in changing competition between borate and
bicarbonate. A caveat is that we can not exclude specific biological processes, and that in taxa with a non
respiration-driven microenvironment, changes in day/night calcification ratios also impacting observed values. As
indicated by Farmer et al., (2019), studies of calcite precipitation rates in foraminifera may help to improve our
understanding of the fundamental basis of boron-based proxies.

**5.4 Evaluation of species for pH reconstructions and water depth pH reconstructions**

This data set allows us to reassess the utility of boron-based proxies for the carbonate system. The main
aim of using boron-based proxies relates to the reconstruction of past oceanic conditions, specifically pH and
$pCO_2$. Mixed-layer species (eg. *G. ruber* and *T. sacculifer)* are potential archives for atmospheric $CO_2$
reconstructions. Other species can shed light on other aspects of the carbon cycle including the physical and
biological carbon pumps.
There are a few main inferences we can make. When integrated with published data, the sensitivities of
$\delta^{11}B_{carbonate}$ to $\delta^{11}B_{borate}$ for *G. ruber* and *T. sacculifer* are similar to previous studies (Martinez-Boti et al., 2015b;
Raitzsch et al., 2018) which supports the fidelity of previous paleo-reconstructions that use published calibrations
between $\delta^{11}B_{carbonate}$ and $\delta^{11}B_{borate}$. The regression we have made for *G. ruber* supports a decrease in $\delta^{11}B_{carbonate}$
with decreasing size fractions (offset of -0.4 ‰, p>0.05) with the sensitivity of $\delta^{11}B_{carbonate}$ to $\delta^{11}B_{borate}$ not being
statistically different from higher size fraction (p<0.05). The variability in our weight per shell for our *G. ruber*,
based data from Henehan et al. (2013), can potentially imply a deviation down to 1‰ relative to calibration line
from Henehan et al. (2013), which can be in line with the maximum deviation observed in our data (~1.2 ‰) and
not inconsistent with a size effect explaining the offset in our calibration. Our $\delta^{11}B_{carbonate}$ data and the sensitivity
to $\delta^{11}B_{borate}$ of *O. universa* supports previous data from Henehan et al. (2016). *N. dutertrei* $\delta^{11}B_{carbonate}$ data span a
large range of pH, allowing us to derive a robust calibration with $\delta^{11}B_{borate}$. It remains premature to assume that a
unique calibration with a slope of ~0.9 can be used for all deeper-dwelling species, more data is needed for *P.*
*obliquiloculata*, *G. menardii* and *G. tumida* to robustly test this assertion.
In order to derive accurate reconstructions of past ambient pH and $pCO_2$, accurate species-specific
calibrations need to be used that are constrained by core-tops or samples from similar types of settings (Fig. 8, 10,
S6). Lower $\delta^{11}B$ signatures in *T. sacculifer* (w/o sacc) are observed in the WEP, which may be explained by the
deeper depth habitat for this taxa, as lower light levels might reduce symbiont photosynthetic activity. Also, we
show that a correction is needed for *T. sacculifer* (w/o sacc) in the WEP in order to accurately reconstruct
atmospheric $CO_2$. When applying calibrations n°2 and 4 to *T. sacculifer* and *G. ruber* (compilation of all data,
Table 3) our data show more variability, especially for *G. ruber* which lead to the larger mismatch compared to *in*
*situ* parameters. The greater divergence of reconstructed values from *in situ* measurements are observed at site
WPO7-01 for both *T. sacculifer* (w/o sacc) and *G. ruber*. More data would be needed to determine a proper
correction for both species and coretop study will be determinant for future downcore reconstructions, especially
in the WEP. We also find that for two species, the boron isotope-pH proxy is a relatively straightforward recorder
of ambient pH, with sensitivities close to unity observed for *O. universa and N. dutertrei*.

There is also promise in using multiple species in a sample from different hydrographic regimes to reconstruct vertical profiles of pH and $pCO_2$. We are able to reproduce pH and $pCO_2$ profiles from multiple sites with different water column structures (Fig. 8) with those reconstructions within error of the *in situ* values, for most sites. In order to avoid circularity, to validate these calibrations, we recalculated ambient pH and $pCO_2$ by first excluding site-specific data and then recalculating species-specific calibrations, followed by application to each specific site. The comparison of the two methods, first using all the data to derive the calibration and recalculate pH and $pCO_2$ (circular) and the second by excluding the site of interest, derive calibrations and calculate pH and $pCO_2$ (not circular), does not show significant differences and validates the robustness of the calibrations (Fig. S5). We utilized the calibrations derived from our data for *G. ruber* (calibration n°1 and 2, Table 3), *T. sacculifer* (calibration n°3 and 4, Table 3), *O. universa* (calibration n°8, Table 3), for *P. obliquiloculata* (calibration n°11, Table 3), and for *N. dutertrei*, *G. tumida* and *G. menardii* the calibration made on the compilation of the deep-dweller (calibration n°13, Table 3). Results are shown in Fig. 8 and evaluated in Fig. 9. For *G. menardii*, more data would be helpful to provide additional constraints. Results for *G. ruber* are the most scattered, potentially due to difference in test sizes (Henehan et al., 2013), or depth habitat. Results reaffirm the importance of working with narrow size fractions (Henehan et al., 2013), the utilization of calibrations derived from the same size fraction or use of offsets to take into account this size fraction effect, and the importance of core-top studies before paleo-application.

## 6. Conclusions and future implications

Our study has extended the boron isotope proxy with data for new species and sites. The work supports previous work showing that depth habitats of foraminifera vary depending on the oceanic regime, and this can impact boron isotope signatures. Low $\delta^{11}B$ values in the WEP compared to other regions for *T. sacculifer* (w/o sacc) may be explained by a reduction in microenvironment pH due to a deeper depth habitat associated with reduced irradiance and thus photosynthetic activity.

In order to accurately develop downcore reconstructions, constraining the depth habitat using core-tops studies is important, as a same species can record the seawater pH at different water depth potentially introducing biases when comparing between different locations. Also, we speculate that a change of the thermocline depth in the past could imply variations of depth habitat and introduce biases in the reconstructions but further work is needed to test this assertion.

The sensitivity of $\delta^{11}B_{carbonate}$ to pH is in line with previously published data for *T. sacculifer*, *G. ruber*. The sensitivity of $\delta^{11}B_{carbonate}$ to pH of *O. universa* (mixed-dweller), *N. dutertrei*. *G. menardii* and *G. tumida* (deep-dwellers) are similar but more data are needed to fully determine those sensitivities. The similarity of boron isotope calibrations for deep-dwelling taxa might be related to similar respiration-driven microenvironments.

Reconstruction of seawater pH and carbonate system parameters is achievable using foraminiferal $\delta^{11}B$ but additional core-top and down-core studies reconstructing depth profiles will be needed in order to further verify calibrations published to date. Past pH and $pCO_2$ water depth profiles can potentially be created by utilizing multiple foraminiferal species in concert with taxon-specific calibrations for similar settings. This approach has much potential for enhancing our understanding of the past workings of the oceanic carbon cycle, and the biological pump.

**Author contribution**

R.E and A.T. wrote the proposals that funded the work. A.T. and F.C. provided the samples. M.G., S.M. and A.T. contributed to the experimental design. A.V. helped for sample preparation. M.G. and S.M contributed to developing the method of boron isotope analysis. M.G. performed the measurements with assistance from S.M. M.G conducted the data analysis. M.G. drafted the paper, which was edited by all authors. Interpretation was led by M.G., A.T., S.M. with input from R.E., A.V. and F.C.

**Competing interests**

The authors declare that they have no conflict of interest.

**Acknowledgments:**

The authors wish to thank Jesse Farmer for his valuable and detailed comments on the actual and a previous version of the manuscript. We wish to thank Michael Henehan for helpful discussion, comments on the manuscript and help with the code. We also want to thank the anonymous reviewer for helpful comments. Lea Bonnin for assistance with picking samples, the IODP repository for provision of samples, the Tripati Laboratory (UCLA) for their technical support, Mervyn Greaves, Madeleine Bohlin (University of Cambridge) for technical support and use of laboratory space, Yoan Germain, Emmanuel Ponzevera and Oanez Lebeau for technical support and use of laboratory space in Brest, Jill Sutton for helpful conversation on the manuscript. Research is supported by DOE BES grant DE-FG02-13ER16402, by the International Research Chair Program that is funded by the French government (LabexMer ANR-10-LABX-19-01), and IAGC student research grant 2017.

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

**Figure caption**

**Figure 1:** (A) Speciation of $B(OH)_3$ and $B(OH)_4^-$ as function of seawater pH (total scale), (B) $\delta^{11}B$ of dissolved inorganic boron species as a function of seawater pH, (C) sensitivity of $\delta^{11}B$ of $B(OH)_4^-$ for a pH ranging from 7.6 to 8.4. T=25°C, S=35, $\delta^{11}B$=39.61 ‰ (Foster et al., 2010), dissociation constant $\alpha$ = 1.0272 (Klochko et al., 2006).

**Figure 2:** Map showing locations of the core-tops used in this study (white diamonds). Red open circles represent the sites used for *in situ* carbonate parameters from GLODAP database (Key et al., 2004).

**Figure 3:** Pre-industrial data versus depth for the sites used in this study. The figure shows seasonal temperatures (extracted from World Ocean Database 2013), density anomaly (kg/m$^3$), pre-industrial pH and pre-industrial $\delta^{11}B$ of $H_4BO_4^-$ (calculated from the GLODAP database and corrected for anthropogenic inputs). Dotted lines are the calculated uncertainties based on errors on TA and DIC from the GLODAP database.

**Figure 4:** Boron isotopic measurements of mixed-layer foraminifera plotted against $\delta^{11}B_{borate}$. $\delta^{11}B_{borate}$ was characterized by determination of the calcification depth of foraminifera utilizing data presented in Fig. 3. A) *G. ruber*, B) *T. sacculifer*, C) *O. universa*. Mono-specific calibrations (Table 3) and error bars on $\delta^{11}B_{borate}$ were derived utilizing the wild bootstrap code from Henehan et al. (2016), while errors on the $\delta^{11}B_{carbonate}$ for this study are reported as 2$\sigma$ of measured AE121 standards during the session of the sample. Calibrations were also derived on the 250-400 size fraction for *G. ruber* and *T. sacculifer* (black dashed lines). Data reported on those graphs have been measured with an MC-ICP-MS.

**Figure 5:** Boron isotopic measurements of deep-dwelling foraminifera ($\delta^{11}B_{carbonate}$) plotted against $\delta^{11}B_{borate}$. $\delta^{11}B_{borate}$ was constrained using foraminiferal calcification depths. A) *P. obliquiloculata*, B) *G. menardii*, C) *N. dutertrei*, D) *G. tumida* and E) Compilation of deep dweller species. Mono-specific calibrations are summarized in Table 3.

**Figure 6:** Boxplots of B/Ca ratios for multiple foraminifera species., including *T. sacculifer* (this study; Foster et al., 2008; Ni et al; 2007; Seki et al., 2010), *G. ruber* (this study; Babila et al., 2014; Foster et al., 2008; Ni et al., 2007), *G. inflata*, *G. bulloides* (Yu et al., 2007), *N. pachyderma* (Hendry et al., 2009; Yu et al., 2013), *N. dutertrei* (this study; Foster et al., 2008), *O. universa*, *P.obliquiloculata, G. menardii, G. tumida* (this study).

**Figure 7:** A) Boxplot showing the calculated microenvironment pH difference ($\Delta$microenvironment pH) between microenvironment and external pH based on the $\delta^{11}B$ data. B) This figure shows that a decrease in insolation can explain the low $\delta^{11}B$ from the WEP. Light penetration profile in the Western Pacific, with $E_0$ in the WEP of 220 J.s$^{-1}$.m$^{-2}$ (Weare et al., 1981) and a light attenuation coefficient of 0.028 (m$^{-1}$) (Wang et al., 2008). Theoretical depths were calculated for a decrease in microenvironment pH of $\Delta pH_1$= -0.02 (e.g. WP07-a); $\Delta pH_1$= -0.04 (e.g. A14), $\Delta pH_2$= -0.06 (e.g. 806A). Light penetration corresponding to Ec is ~12%, $\Delta pH_0$~7%, $\Delta pH_1$ ~5%, $\Delta pH_2$ ~1% respective calcification depth are 75m, 90m, 110m and 150m. Grey band is the calcification depth calculated that

explains the Δ microenvironment pH from $\Delta pH_0$ to $\Delta pH_2$. Dotted lines show the range of the calcification depth
for *T. sacculifer* (w/o sacc) in the WEP utilized in this study.

**Figure 8:** Water depth pH profiles reconstructed at every site applying the mono-specific calibrations derived from
our results (Table 3). Figure is showing measured $\delta^{11}B_{calcite}$, $\delta^{11}B_{borate}$ calculated according to different calibrations
(see Table 3 and text), calculated pH based on $\delta^{11}B$ ($pH_{\delta 11B}$) and $pCO_2$ calculated from $pH_{\delta 11B}$ and alkalinity.

**Figure 9:** Evaluation of the reconstructed parameters, $\delta^{11}B_{borate}$, pH and $pCO_2$ versus *in situ* parameter calculated
in Fig. 8 (based on $\delta^{11}B$ and alkalinity). The recalculated parameters are consistent with *in situ* data, except for *G.*
*ruber*, and this variability might be explained by the different test sizes within measured size fractions.
**Table caption**

**Table 1:** Box-core information

**Table 2:** Analytical results of $\delta^{13}C$, $\delta^{18}O$, $\delta^{11}B$ and elemental ratios Li/Ca, B/Ca and Mg/Ca

**Table 3:** Species-specific $\delta^{11}B_{carbonate}$ to $\delta^{11}B_{borate}$ calibrations from literature and from our data

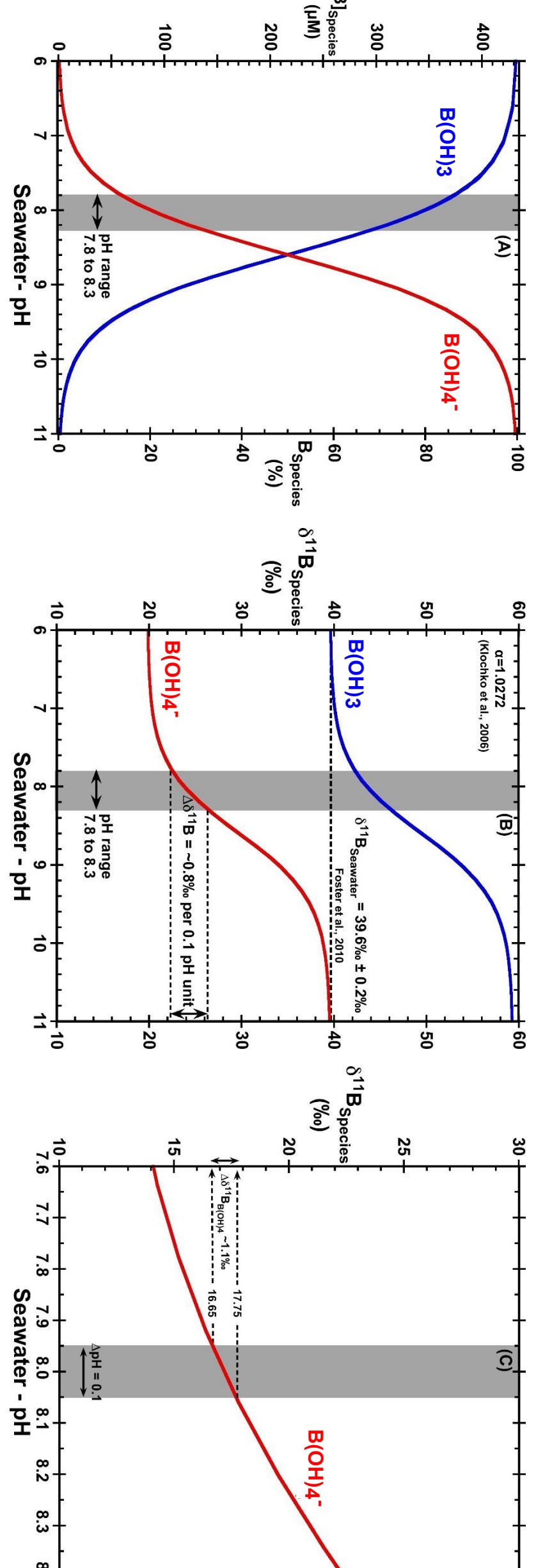

**Figure 1**

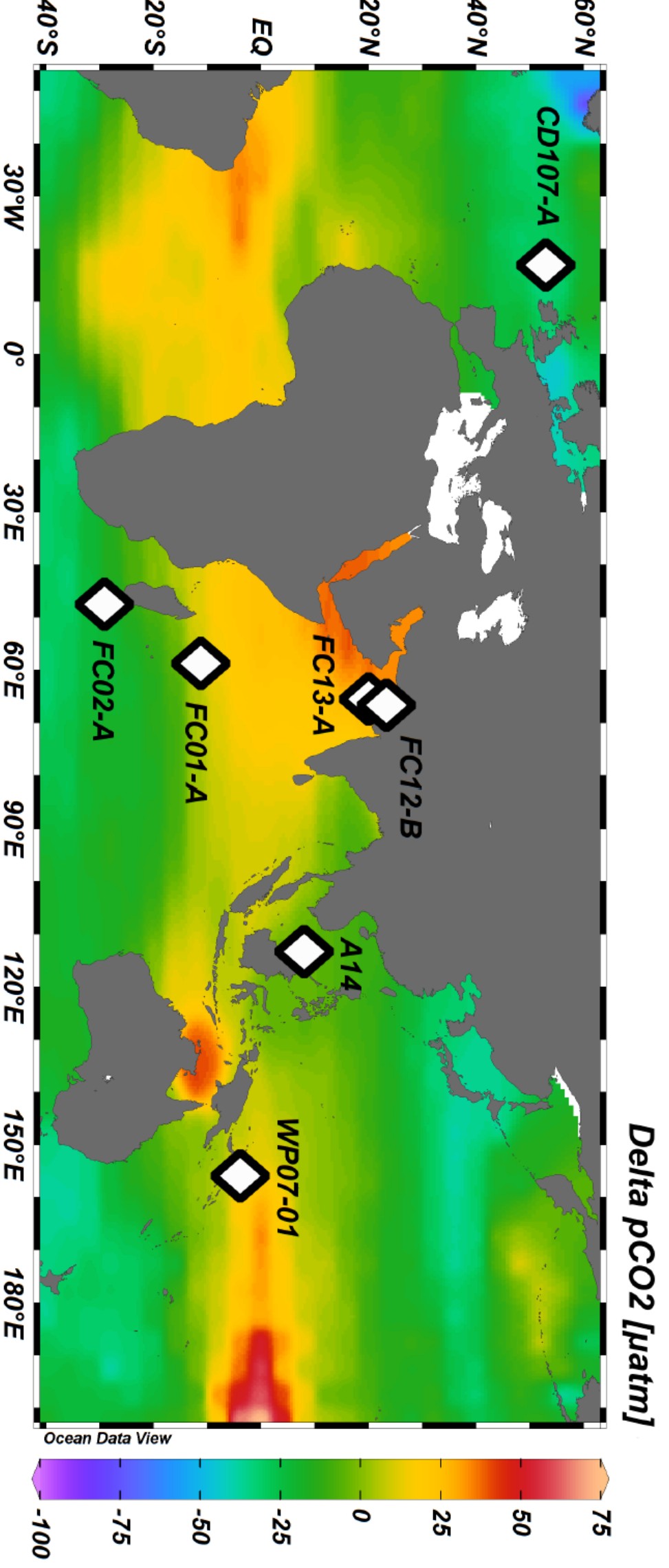

**Figure 2**

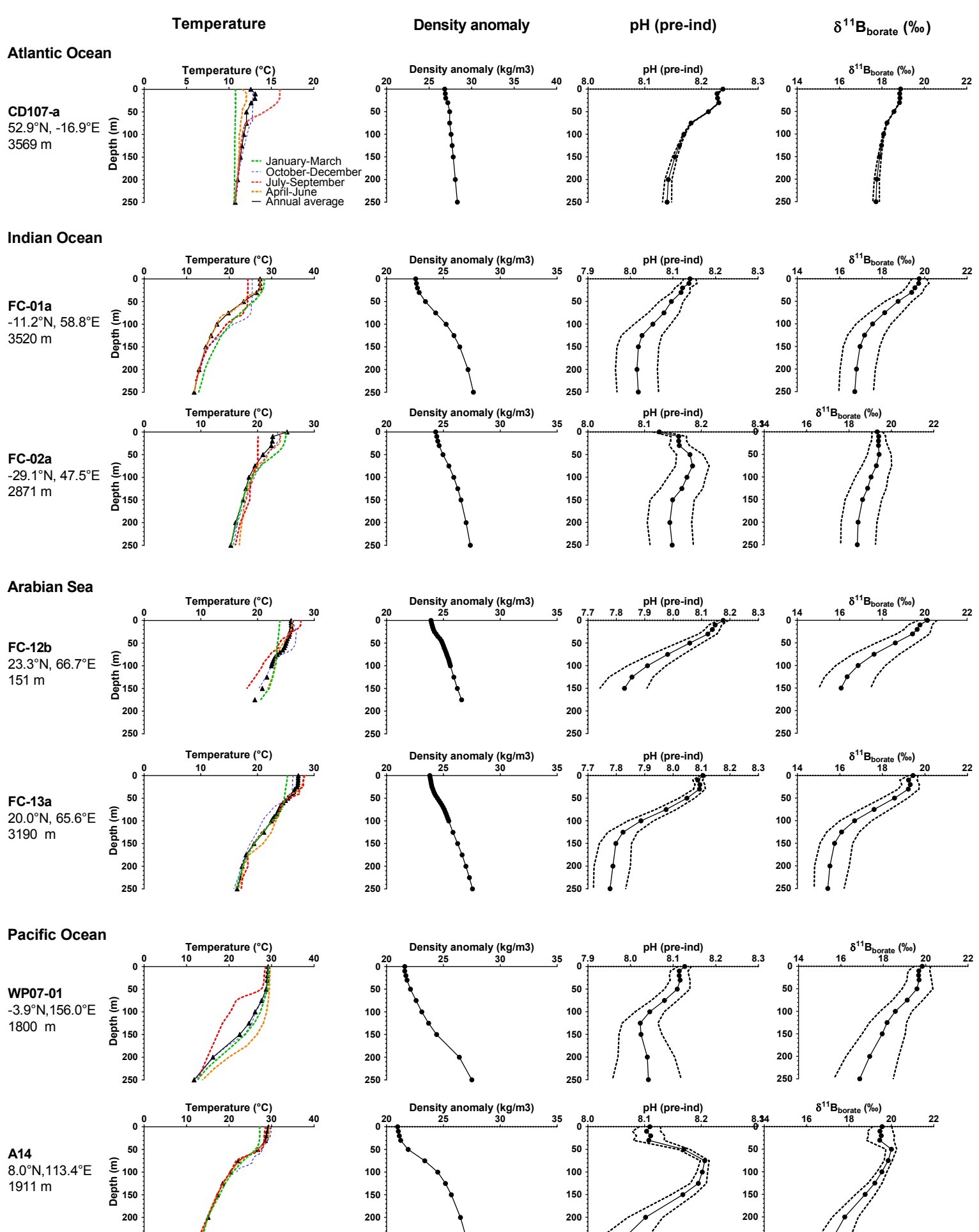

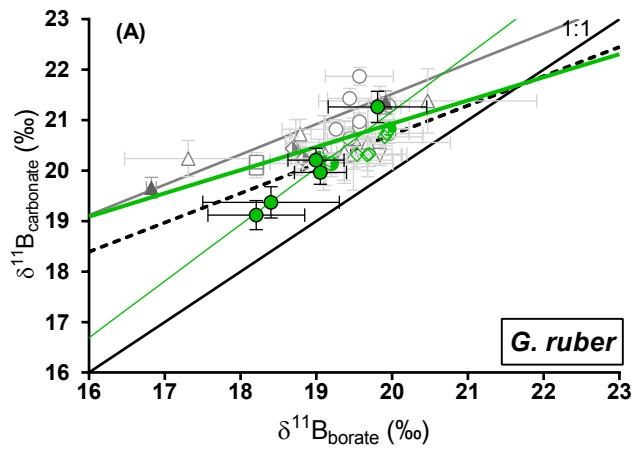

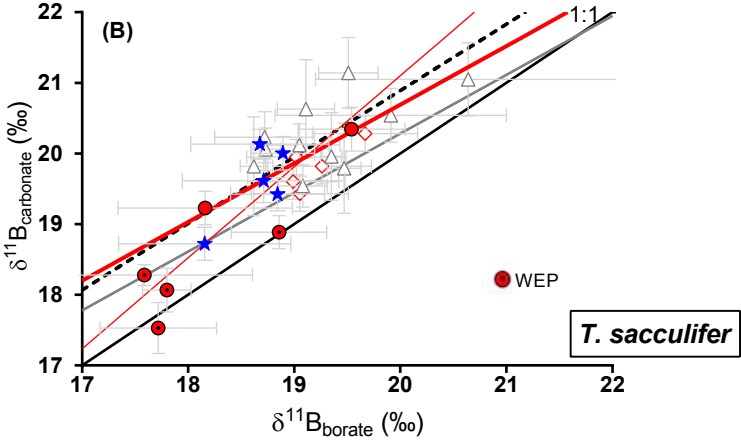

- ● $\delta^{11}B_{G. ruber}$ (core-top, 250-400μm, this study)
- ◇ $\delta^{11}B_{G. ruber}$ (core-top, 300-355μm, Foster et al., 2008)
- ◐ $\delta^{11}B_{G. ruber}$ (core-top, 250-300μm, Henehan et al., 2013)
- ○ $\delta^{11}B_{G. ruber}$ (core-top, 250-455μm, Henehan et al., 2013)
- □ $\delta^{11}B_{G. ruber}$ (sediment trap, 250-355μm, Henehan et al., 2013)
- ◆ $\delta^{11}B_{G. ruber}$ (tow, Henehan et al., 2013)
- ▲ $\delta^{11}B_{G.ruber}$ (culture, Henehan et al., 2013)
- ▽ $\delta^{11}B_{G. ruber}$ (grab sample, 250-355μm, Henehan et al., 2013)
- △ $\delta^{11}B_{G. ruber}$ (core-top, 315-355μm, Raizsch et al., 2018)
- — *G. ruber* calibration line (all data, this study, 250-455 )
- — *G. ruber* calibration line (core-top, this study, 250-400μm)
- — *G. ruber* calibration line (culture, Henehan et al., 2013)
- --- *G. ruber* calibration line (this study, 250-300μm from Henehan et al., 2013)

- ● $\delta^{11}B_{T.sacculifer (w/o sacc)}$ (core-top, 250-400μm, this study)
- △ $\delta^{11}B_{T. sacculifer (w/o sacc)}$ (core-top, 315-355μm, Raitzsch et al., 2018)
- ★ $\delta^{11}B_{T.sacculifer (sacc)}$ (core-top, 250-400μm, this study)
- ◇ $\delta^{11}B_{T. sacculifer (sacc)}$ (core-top, 500-600μm, Foster et al., 2008)
- — *T. sacculifer (w/o sacc and sacc)* calibration line (all data, 250-600μm, this study)
- — *T. sacculifer (w/o sacc and sacc)* calibration line (core-top, 250-400μm, this study)
- — *T. sacculifer (sacc)* calibration line (Martinez-Boti et al., 2015)
- --- *T. sacculifer (w/o sacc and sacc)* calibration line 250-400 μm (this study and Raitzsch et al., 2018)

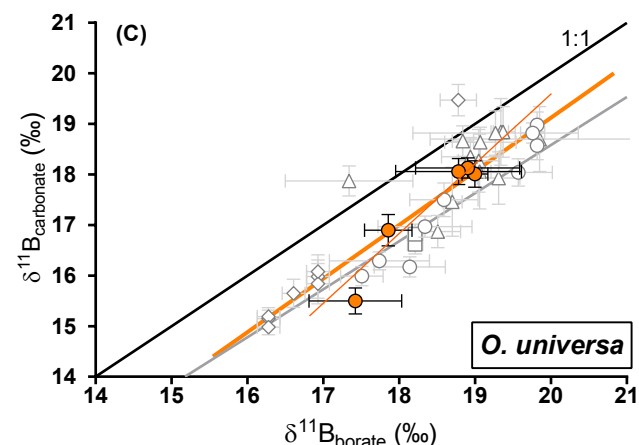

- ● $\delta^{11}B_{O. universa}$ (core-top, this study)
- ○ $\delta^{11}B_{O. universa}$ (core-top, Henehan et al., 2016)
- □ $\delta^{11}B_{O. universa}$ (sediment trap, Henehan et al., 2016)
- ◇ $\delta^{11}B_{O. universa}$ (tow, Henehan et al., 2016)
- △ $\delta^{11}B_{O. universa}$ (core-top, Raitzsch et al., 2018)
- — *O. universa* calibration line (core-top, this study)
- — *O. universa* calibration line (this study,Henehan et al., 2016, Raitzsch et al., 2018)
- — *O. universa* calibration line (wild, Henehan et al., 2016)

**Figure 4**

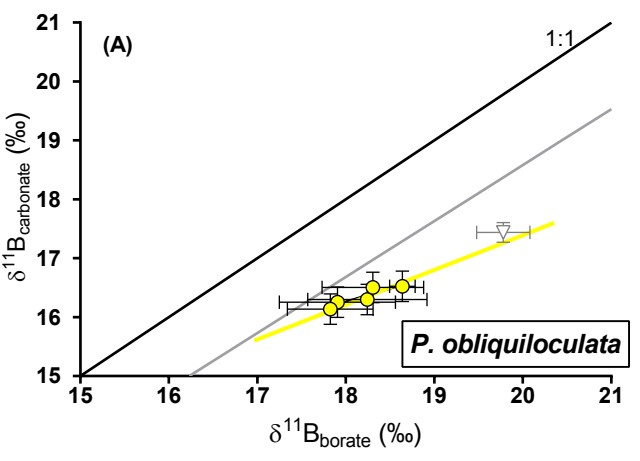

- ○ $\delta^{11}B_{P.obliquiloculata}$ (Core-top, this study)
- ▽ $\delta^{11}B_{P.obliquiloculata}$ (Henehan et al., 2016)
- ── *P. obliquiloculata* calibration line (this study, Henehan et al., 2016)
- ── *O. universa* calibration curve (Henehan et al., 2016)

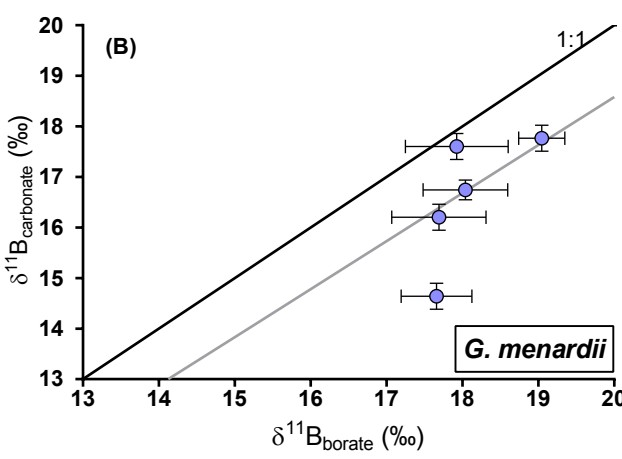

- ● $\delta^{11}B_{G.\ menardii}$ (this study)
- ── *O. universa* calibration curve (Henehan et al., 2016)

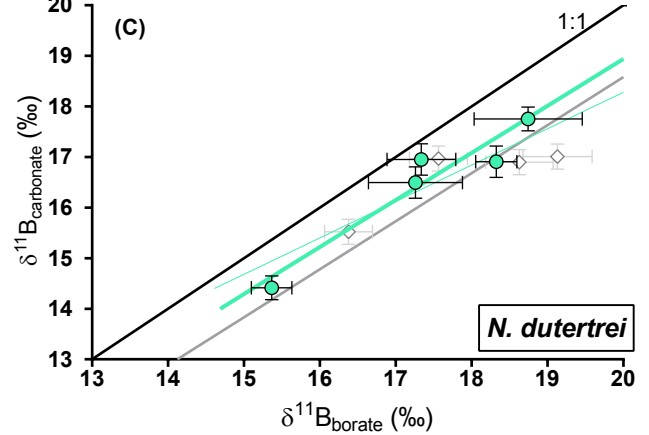

- ● $\delta^{11}B_{N.\ dutertrei}$ (Core-top, this study)
- ◇ $\delta^{11}B_{N.\ dutertrei}$ (Core-top, Foster et al., 2008)
- ── *O. universa* calibration line (This study)
- ── *O. universa* calibration line (This study, Foster et al., 2008)
- ── *O. universa* calibration line (Henehan et al., 2016)

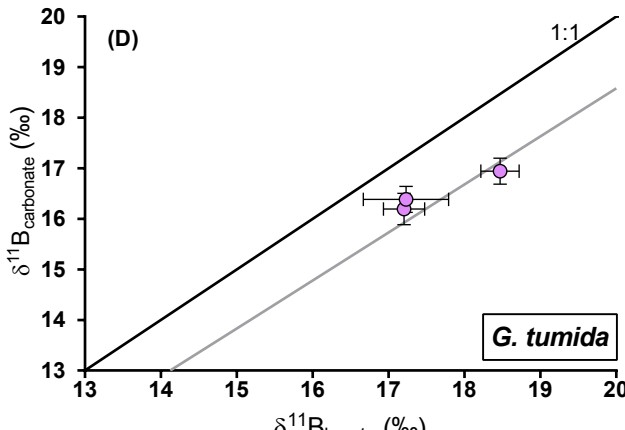

- ● $\delta^{11}B_{G.\ tumida}$ (this study)
- ── *O. universa* calibration curve (Henehan et al., 2016)

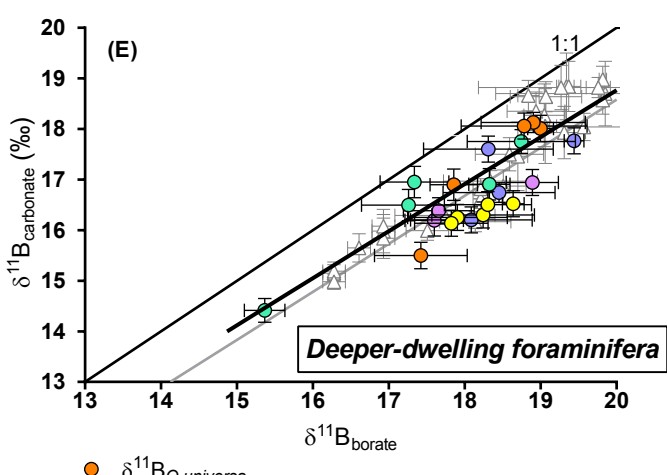

- ● $\delta^{11}B_{O.universa}$
- ● $\delta^{11}B_{P.obliquiloculata}$
- ● $\delta^{11}B_{N.\ dutertrei}$
- ● $\delta^{11}B_{G.\ menardii}$
- ● $\delta^{11}B_{G.\ tumida}$
- △ $\delta^{11}B_{deep-dweller}$ from literature
- ── Deep-dweller calibration line
- ── *O. universa* calibration line (Henehan et al., 2016)

**Figure 5**   36

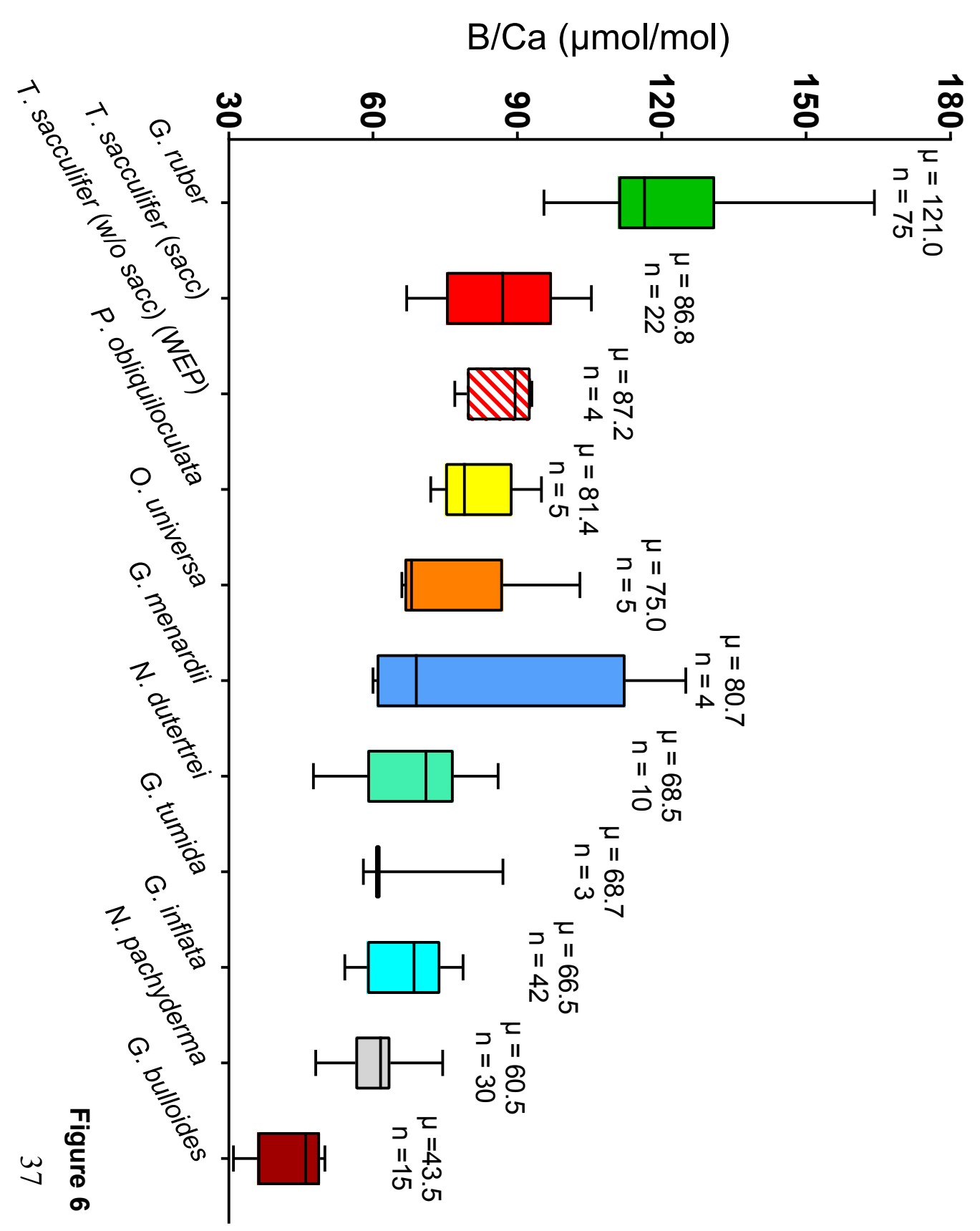

**Figure 6**

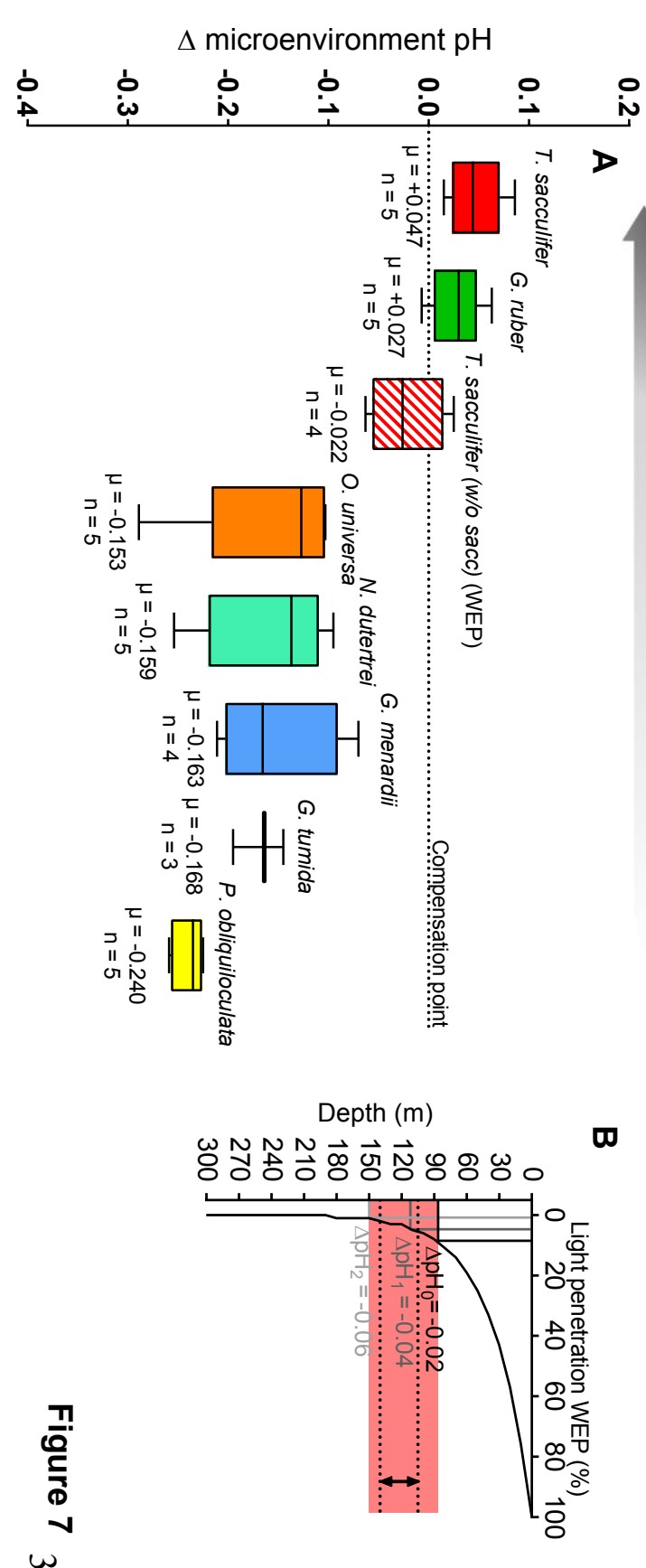

**Figure 7**

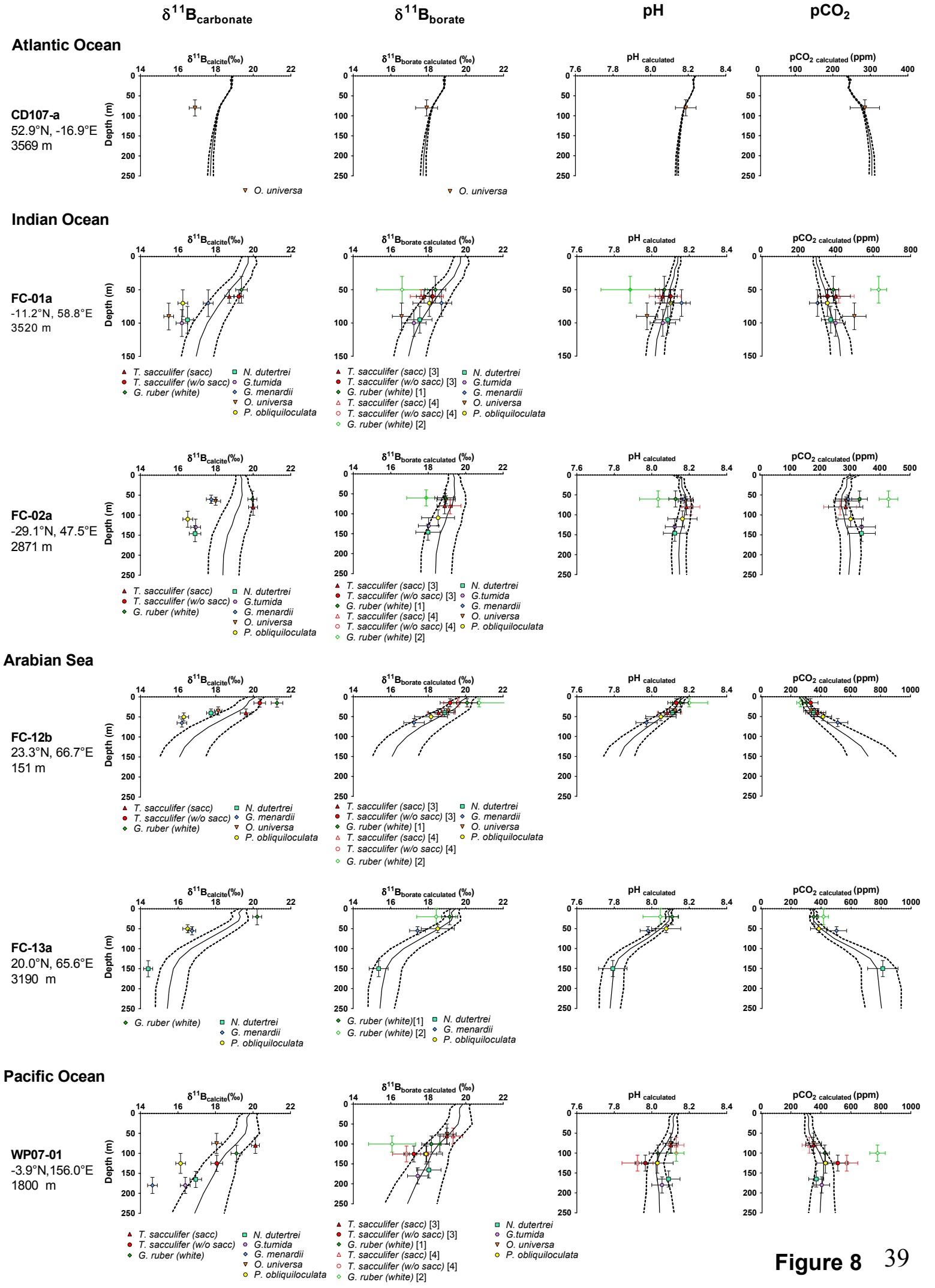

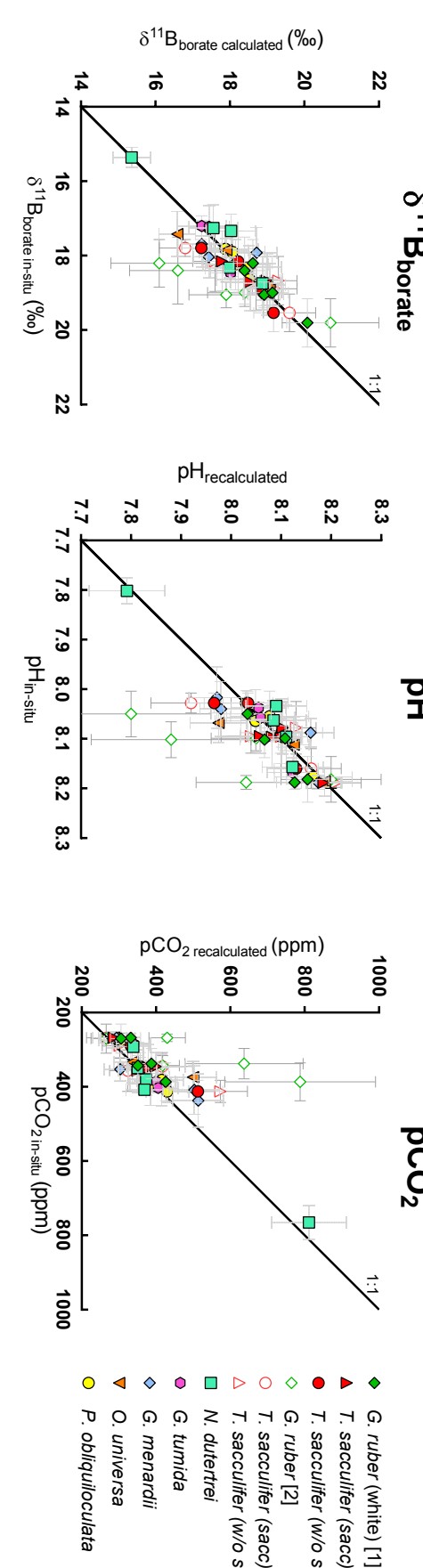

**Figure 9** 40

**Table 1**

| Label | Box-Core | Site | Latitude (N) | Longitude (E) | Depth (mbsl) | Oceanic Regime | $^{14}$C age (year) |
|---|---|---|---|---|---|---|---|
| *Atlantic Ocean* | | | | | | | |
| CD107-a | CD107 | A | 52.92 | -16.92 | 3569 | non-upwelling | <3000[a] |
| *Indian Ocean* | | | | | | | |
| FC-01a | WIND-33B | I | -11.21 | 58.77 | 3520 | non-upwelling | |
| FC-02a | WIND-10B | K | -29.12 | 47.55 | 2871 | non-upwelling | 7252 ± 27[b] |
| *Arabian Sea* | | | | | | | |
| FC-12b | CD145 | A150 | 23.30 | 66.70 | 151 | seasonal upwelling | |
| FC-13a | CD145 | A3200 | 20.00 | 65.58 | 3190 | seasonal upwelling | |
| *Pacific Ocean* | | | | | | | |
| WP07-01 | | | -3.93 | 156.00 | 1800 | non-upwelling | 7300-8600[c] |
| A14 | | | 8.02 | 113.39 | 1911 | non-upwelling | 7300-8600[c] |
| 806 | | A | 0.32 | 159.36 | 2521 | equatorial divergence | 7300-8600[c] |
| 807 | | A | 3.61 | 156.62 | 2804 | equatorial divergence | 7300-8600[c] |

[a] Thomson et al., 2000
[b] Wilson et al., 2012
[c] Age for core-top of site 806B from Lea et al., 2000

**Table 2**

| Core | Species | Fraction size (μm) | $\delta^{13}C^*$ (‰) | $\delta^{18}O^*$ (‰) | $\delta^{11}B_{c1}$ (‰) | $\delta^{11}B_{c2}$ (‰) | $\delta^{11}B_{average}$** (‰) | Li/Ca*** (μmol/mol) | B/Ca*** (μmol/mol) | Mg/Ca*** (mmol/mol) |
|---|---|---|---|---|---|---|---|---|---|---|
| *Atlantic Ocean* | | | | | | | | | | |
| CD107a | O. universa | >500 | 1.99±0.03 | 1.25±0.11 | 16.85±0.31 (2SD, nAE121=11) | 16.95±0.31 (2SD, nAE121=11) | 16.90±0.22 | 13.9±0.4 | 68±7 | 3.60±0.01 |
| *Indian Ocean* | | | | | | | | | | |
| FC-01a | G. ruber (white ss) | 250-300 | 1.37±0.03 | -1.32±0.11 | 19.33±0.31 (2SD, nAE121=11) | 19.41±0.31 (2SD, nAE121=11) | 19.37±0.22 | 15.4±0.4 | 109±7 | 3.98±0.01 |
| FC-01a | T. sacculifer (sacc) | 300-400 | 1.88±0.03 | -2.20±0.11 | 18.71±0.24 (2SD, nAE121=10) | 18.73±0.24 (2SD, nAE121=10) | 18.72±0.17 | 12.1±0.4 | 87±7 | 3.45±0.01 |
| FC-01a | T. sacculifer (w/o sacc) | 300-400 | 2.02±0.03 | -1.05±0.11 | 19.13±0.24 (2SD, nAE121=10) | 19.32±0.24 (2SD, nAE121=10) | 19.23±0.17 | 12.1±0.4 | 82±7 | 3.42±0.01 |
| FC-01a | O. universa | >500 | 1.00±0.03 | -0.55±0.11 | 15.50±0.26 (2SD, nAE121=14) | 15.50±0.26 (2SD, nAE121=14) | 15.50±0.26 | 15.4±0.4 | 78±7 | 2.06±0.01 |
| FC-01a | P. obliquiloculata | 300-400 | 1.64±0.03 | 0.43±0.11 | 16.40±0.26 (2SD, nAE121=14) | 16.10±0.26 (2SD, nAE121=14) | 16.25±0.18 | 12.7±0.4 | 63±7 | 2.26±0.01 |
| FC-01a | G. menardii | 300-400 | 1.28±0.03 | -0.43±0.11 | 17.52±0.26 (2SD, nAE121=14) | 17.69±0.26 (2SD, nAE121=14) | 17.60±0.18 | 12.1±0.4 | 73±7 | 1.81±0.01 |
| FC-01a | N. dutertrei | 300-400 | 1.29±0.03 | -0.53±0.11 | 16.40±0.26 (2SD, nAE121=14) | 16.59±0.26 (2SD, nAE121=14) | 16.50±0.22 | 10.0±0.4 | 61±7 | 1.79±0.01 |
| FC-01a | G. tumida | 300-400 | 0.30±0.03 | -1.40±0.11 | 16.21±0.31 (2SD, nAE121=11) | 16.18±0.31 (2SD, nAE121=11) | 16.20±0.22 | 18.2±0.4 | 125±7 | 3.47±0.01 |
| FC-02a | G. ruber (white ss) | 250-300 | 1.43±0.03 | -1.60±0.11 | 20.02±0.24 (2SD, nAE121=10) | 19.90±0.24 (2SD, nAE121=10) | 19.96±0.17 | 14.2±0.4 | 106±7 | 3.30±0.01 |
| FC-02a | T. sacculifer (sacc) | 300-400 | 1.52±0.03 | -1.40±0.11 | 20.07±0.24 (2SD, nAE121=10) | 19.93±0.24 (2SD, nAE121=10) | 20.00±0.17 | 13.7±0.4 | 106±7 | 3.34±0.01 |
| FC-02a | T. sacculifer (w/o sacc) | 300-400 | 1.79±0.03 | 0.02±0.11 | 23.23±0.24 (2SD, nAE121=10) | 23.22±0.24 (2SD, nAE121=10) | 23.22±0.17 | 14.8±0.4 | 67±7 | 4.40±0.01 |
| FC-02a | O. universa | >500 | 0.34±0.03 | 0.56±0.11 | 18.05±0.26 (2SD, nAE121=14) | 17.97±0.26 (2SD, nAE121=14) | 18.01±0.18 | 16.6±0.4 | 83±7 | 2.33±0.01 |
| FC-02a | P. obliquiloculata | 300-400 | 1.73±0.03 | -0.51±0.11 | 16.35±0.26 (2SD, nAE121=14) | 16.69±0.26 (2SD, nAE121=14) | 16.52±0.18 | 15.8±0.4 | 125±7 | 2.21±0.01 |
| FC-02a | G. menardii | 300-400 | 1.03±0.03 | -0.55±0.11 | 16.77±0.26 (2SD, nAE121=14) | 17.03±0.26 (2SD, nAE121=14) | 16.90±0.22 | 8.6±0.4 | 82±7 | 2.13±0.01 |
| FC-02a | N. dutertrei | 300-400 | 1.64±0.03 | -0.28±0.11 | 16.78±0.26 (2SD, nAE121=14) | 16.91±0.26 (2SD, nAE121=14) | 16.85±0.22 | 15.6±0.4 | 87±7 | 1.90±0.01 |
| FC-02a | G. tumida | 300-400 | | | 16.93±0.26 (2SD, nAE121=14) | 16.95±0.26 (2SD, nAE121=14) | 16.94±0.18 | | | |
| *Arabian Sea* | | | | | | | | | | |
| FC-12b | G. ruber (white ss) | 250-300 | 0.58±0.03 | -2.82±0.11 | 21.30±0.31 (2SD, nAE121=11) | 21.23±0.31 (2SD, nAE121=11) | 21.26±0.22 | 19.5±0.4 | 164±7 | 5.76±0.01 |
| FC-12b | G. sacculifer (s) | 300-400 | 1.76±0.03 | -2.15±0.11 | 19.65±0.31 (2SD, nAE121=11) | 19.57±0.31 (2SD, nAE121=11) | 19.61±0.22 | 14.6±0.4 | 101±7 | 4.28±0.01 |
| FC-12b | T. sacculifer (w/o sacc) | 300-400 | 1.97±0.03 | -2.19±0.11 | 20.32±0.31 (2SD, nAE121=11) | 20.37±0.31 (2SD, nAE121=11) | 20.34±0.22 | 16.7±0.4 | 116±7 | 4.90±0.01 |
| FC-12b | O. universa | >500 | 1.89±0.03 | -1.59±0.11 | 18.13±0.20 (2SD, nAE121=6) | 18.13±0.20 (2SD, nAE121=6) | 18.13±0.20 | 13.6±0.4 | 103±7 | 6.91±0.01 |
| FC-12b | P. obliquiloculata | 300-400 | 0.5±0.03 | -1.58±0.11 | 16.45±0.26 (2SD, nAE121=14) | 16.15±0.26 (2SD, nAE121=14) | 16.30±0.18 | 16.7±0.4 | 95±7 | 3.61±0.01 |
| FC-12b | G. menardii | 300-400 | 1.05±0.03 | -0.97±0.11 | 16.2±0.26 (2SD, nAE121=14) | 16.20±0.26 (2SD, nAE121=14) | 16.20±0.26 | 14.8±0.4 | 75±7 | 3.44±0.01 |
| FC-12b | N. dutertrei | 300-400 | 1.35±0.03 | -1.57±0.11 | 17.77±0.24 (2SD, nAE121=10) | 17.73±0.24 (2SD, nAE121=10) | 17.75±0.17 | 17.1±0.4 | 75±7 | 3.25±0.01 |
| FC-13a | G. ruber (white ss) | 250-300 | 0.08±0.03 | -3.71±0.11 | 20.27±0.24 (2SD, nAE121=10) | 20.15±0.24 (2SD, nAE121=10) | 20.21±0.17 | 16.4±0.4 | 147±7 | 4.52±0.01 |
| FC-13a | T. sacculifer (w/o sacc) | 300-400 | 1.59±0.03 | -2.46±0.11 | 17.85±0.29 (2SD, nAE121=12) | 17.85±0.29 (2SD, nAE121=12) | 17.85±0.29 | 15.7±0.4 | 121±7 | 5.49±0.01 |
| FC-13a | P. obliquiloculata | 300-400 | 0.00±0.03 | -0.97±0.11 | 16.51±0.26 (2SD, nAE121=14) | 16.50±0.26 (2SD, nAE121=14) | 16.51±0.18 | 18.7±0.4 | 79±7 | 4.43±0.01 |
| FC-13a | G. menardii | 300-400 | 0.75±0.03 | -1.07±0.11 | 16.74±0.20 (2SD, nAE121=6) | 16.74±0.20 (2SD, nAE121=6) | 16.74±0.20 | 9.2±0.4 | 60±7 | 1.99±0.01 |
| FC-13a | N. dutertrei | 300-400 | 0.71±0.03 | -1.41±0.11 | 14.43±0.24 (2SD, nAE121=10) | 14.40±0.24 (2SD, nAE121=10) | 14.41±0.17 | 15.7±0.4 | 69±7 | 1.98±0.01 |
| *Pacific Ocean* | | | | | | | | | | |
| WP07-a | G. ruber (white ss) | 250-400 | | | 19.12±0.29 (2SD, nAE121=12) | 19.12±0.29 (2SD, nAE121=12) | 19.12±0.29 | 14.5±0.4 | 144±7 | 4.32±0.01 |
| WP07-a | T. sacculifer (sacc) | 250-400 | | | 20.13±0.21 (2SD, nAE121=11) | 20.13±0.21 (2SD, nAE121=11) | 20.13±0.21 | 12.7±0.4 | 92±7 | 4.44±0.01 |
| WP07-a | T. sacculifer (w/o sacc) | 250-400 | | | 18.10±0.31 (2SD, nAE121=11) | 18.04±0.31 (2SD, nAE121=11) | 18.07±0.22 | 12.3±0.4 | 192±7 | 4.51±0.01 |
| WP07-a | O. universa | 500-630 | | | 18.13±0.26 (2SD, nAE121=14) | 17.99±0.26 (2SD, nAE121=14) | 18.06±0.18 | 11.9±0.4 | 71±7 | 7.52±0.01 |
| WP07-a | P. obliquiloculata | 250-400 | | | 16.08±0.26 (2SD, nAE121=14) | 16.19±0.26 (2SD, nAE121=14) | 16.14±0.18 | 13.4±0.4 | 72±7 | 3.02±0.01 |
| WP07-a | G. menardii | 250-400 | | | 14.74±0.26 (2SD, nAE121=14) | 14.53±0.26 (2SD, nAE121=14) | 14.64±0.18 | 13.5±0.4 | 85±7 | 2.68±0.01 |
| WP07-a | N. dutertrei | 250-400 | | | 16.91±0.31 (2SD, nAE121=11) | 16.99±0.31 (2SD, nAE121=11) | 16.95±0.22 | 21.7±0.4 | 86±7 | 3.66±0.01 |
| WP07-a | G. tumida | 250-400 | | | 16.45±0.26 (2SD, nAE121=14) | 16.32±0.26 (2SD, nAE121=14) | 16.39±0.18 | 10.6±0.4 | 58±7 | 2.55±0.01 |
| 806A | T. sacculifer (w/o sacc) | 250-400 | | | 17.53±0.36 (2SD, nAE121=11) | 17.53±0.36 (2SD, nAE121=11) | 17.53±0.36 | 14.40±0.4 | 77±7 | 3.89±0.01 |
| 807A | T. sacculifer (w/o sacc) | 250-400 | | | 18.38±0.21 (2SD, nAE121=11) | 18.17±0.21 (2SD, nAE121=11) | 18.28±0.15 | 12.54±0.4 | 87±7 | 4.24±0.01 |
| A14 | G. ruber (white ss) | 250-400 | | | 18.91±0.24 (2SD, nAE121=10) | 19.17±0.24 (2SD, nAE121=10) | 19.04±0.17 | 12.0±0.4 | 102±7 | 3.91±0.01 |
| A14 | T. sacculifer (sacc) | 250-400 | | | 19.53±0.24 (2SD, nAE121=10) | 19.32±0.24 (2SD, nAE121=10) | 19.42±0.17 | 12.3±0.4 | 93±7 | 3.76±0.01 |
| A14 | T. sacculifer (w/o sacc) | 250-400 | | | 18.93±0.24 (2SD, nAE121=10) | 18.84±0.24 (2SD, nAE121=10) | 18.88±0.17 | 11.3±0.4 | 66±7 | |
| A14 | O. universa | 500-560 | | | 17.33±0.26 (2SD, nAE121=14) | 17.08±0.26 (2SD, nAE121=14) | 17.20±0.18 | 16.9±0.4 | 75±7 | 6.59±0.01 |
| A14 | N. dutertrei | 250-400 | | | 14.39±0.31 (2SD, nAE121=11) | 14.39±0.31 (2SD, nAE121=11) | 14.39±0.31 | | | 1.99±0.01 |

\* uncertainties given in 1SD (see text)

\*\* When two measurements were carried out uncertainty was calculated with $\Delta a = \sqrt{(1/\sum_i (1/\Delta a_i)^2)}$; with only one measurement the error was determined on reproducibility of the AE121 standard

\*\*\* Uncertainty given in 2SD, calculated on the reproducibility of CamWuellestorfi (see text and table S3; ref in Misra et al., 2014)

**Table 3**

| Species | Size fraction (μm) | Material | Instrument (original) | Regression method | $\delta^{11}B_{borate}= f(\delta^{11}B_{calcite})$ | n | Calibration number | Reference |
|---|---|---|---|---|---|---|---|---|
| G. ruber | ~380 | Culture/core tops/plankton tows | MC-ICP-MS | | $\delta^{11}B_{borate}=(\delta^{11}B_{calcite} - 9.52\ (\pm2.02))/0.6\ (\pm0.11)$ | | | Henehan et al., 2013 |
| G. ruber | 315-355 | Core-tops | MC-ICP-MS | | $\delta^{11}B_{borate}=(\delta^{11}B_{calcite} - 11.78\ (\pm3.20))/0.45\ (\pm0.16)$ | | | Raitzsch et al., 2018 |
| T. sacculifer | n.d. | Culture/artificial seawater enriched in B | N-TIMS | | $\delta^{11}B_{borate}=(\delta^{11}B_{calcite} - 3.94\ (\pm4.02))/0.82\ (\pm0.22)$ | | | Sanyal et al., 2001 refitted Martinez-Botí et al., 2015 |
| T. sacculifer | 315-355 | Core-tops | MC-ICP-MS | | $\delta^{11}B_{borate}=(\delta^{11}B_{calcite} - 8.86\ (\pm5.27))/0.59\ (\pm0.21)$ | | | Raitzsch et al., 2018 |
| O. universa | no effect | Core-tops/plankton tows/sediment traps | MC-ICP-MS | | $\delta^{11}B_{borate}=(\delta^{11}B_{calcite} + 0.42\ (\pm2.85))/0.95\ (\pm0.17)$ | | | Henehan et al., 2016 |
| O. universa | >425 | Core-tops | MC-ICP-MS | | $\delta^{11}B_{borate}=(\delta^{11}B_{calcite} + 5.69\ (\pm7.51))/1.26\ (\pm0.39)$ | | | Raitzsch et al., 2018 |
| G. bulloides | 300-355 | Core-tops/sediment trap | MC-ICP-MS | | $\delta^{11}B_{borate}=(\delta^{11}B_{calcite} + 3.440\ (\pm4.584))/1.074\ (\pm0.252)$ | | | Martinez-Botí et al., 2015 |
| G. bulloides | 315-355 | Core-tops | MC-ICP-MS | | $\delta^{11}B_{borate}=(\delta^{11}B_{calcite} + 3.81\ (\pm13.17))/1.13\ (\pm0.72)$ | | | Raitzsch et al., 2018 |
| N. pachyderma | 150-200 | Core-tops | MC-ICP-MS | | $\delta^{11}B_{borate} = \delta^{11}B_{calcite} + 3.38$ | | | Yu et al., 2013 |
| G. ruber | 250-400 | Core-tops | MC-ICP-MS | Bootstrap | $\delta^{11}B_{borate}=(\delta^{11}B_{calcite} - 9.11\ (\pm0.73)/0.58\ (\pm0.91)$ | 9 | 0 | This study; Henehan et al., 2013 |
| G. ruber | 250-400 | Core-tops | MC-ICP-MS | Bootstrap | $\delta^{11}B_{borate}=(\delta^{11}B_{calcite} + 1.23\ (\pm0.59)/1.12\ (\pm1.67)$ | 5 | 1 | This study |
| G. ruber | 250-455 | Core-tops | MC-ICP-MS | Bootstrap | $\delta^{11}B_{borate}=(\delta^{11}B_{calcite} - 11.73\ (\pm0.83)/0.46\ (\pm0.34)$ | 40 | 2 | This study; Foster et al., 2008; Raitzsch et al., 2018 |
| T. sacculifer (sacc) | 250-400 | Core-tops | MC-ICP-MS | Bootstrap | $\delta^{11}B_{borate}=(\delta^{11}B_{calcite} + 6.06\ (\pm 0.25)/1.38\ (\pm 1.33)$ | 11 | 3 | This study |
| T. sacculifer (sacc and w/o sacc) | 250-400 | Core-tops | MC-ICP-MS | Bootstrap | $\delta^{11}B_{borate}=(\delta^{11}B_{calcite} - 4.09\ (\pm0.86)/0.83\ (\pm0.48)$ | 27 | 4 | This study; Foster et al., 2008; Raitzsch et al., 2018 |
| T. sacculifer (sacc and w/o sacc) | 300-400 | Core-tops | MC-ICP-MS | Bootstrap | $\delta^{11}B_{borate}=(\delta^{11}B_{calcite} - 0.34\ (\pm1.83)/0.93\ (\pm0.55)$ | 5 | 5 | This study |
| N. dutertrei | 300-400 | Core-tops | MC-ICP-MS | Bootstrap | $\delta^{11}B_{borate}=(\delta^{11}B_{calcite} - 3.88\ (\pm0.65)/0.72\ (\pm0.74)$ | 9 | 6 | This study; Foster et al., 2008 |
| O. universa | 400-600 | Core-tops | MC-ICP-MS | Bootstrap | $\delta^{11}B_{borate}=(\delta^{11}B_{calcite} + 8.01\ (\pm23)/1.38\ (\pm2.67)$ | 5 | 7 | This study |
| O. universa | 400-600 | Core-tops | MC-ICP-MS | Bootstrap | $\delta^{11}B_{borate}=(\delta^{11}B_{calcite} + 2.08\ (\pm0.59)/1.06\ (\pm0.13)$ | 36 | 8 | This study; Henehan et al., 2016; Raitzsch et al., 2018 |
| G. menardii | 400-600 | Core-tops | MC-ICP-MS | Bootstrap | $\delta^{11}B_{borate}=(\delta^{11}B_{calcite} - 5.36\ (\pm1.36)/0.65\ (\pm0.76)$ | 5 | 9 | This study |
| G. tumida | 400-600 | Core-tops | MC-ICP-MS | Bootstrap | $\delta^{11}B_{borate}=(\delta^{11}B_{calcite} - 6.33\ (\pm2.52)/0.57\ (\pm1.2)$ | 3 | 10 | This study |
| P. obliquiloculata | 300-400 | Core-tops | MC-ICP-MS | Bootstrap | $\delta^{11}B_{borate}=(\delta^{11}B_{calcite} - 5.59\ (\pm4.16)/0.65)$ | 6 | 11 | This study; Henehan et al., 2016 |
| Deep-dweller | 300-600 | Core-tops | MC-ICP-MS | Bootstrap | $\delta^{11}B_{borate}=(\delta^{11}B_{calcite} -1.99\ (\pm0.13)/0.82\ (\pm0.27)$ | 22 | 12 | This study |
| Deep-dweller | 300-600 | Core-tops | MC-ICP-MS | Bootstrap | $\delta^{11}B_{borate}=(\delta^{11}B_{calcite} - 0.18\ (\pm0.6)/0.95\ (\pm0.13)$ | 54 | 13 | This study; Foster et al., 2008; Henehan et al., 2016; Raitzsch et al., 2018 |