# Peer review of "Seawater pH reconstruction using boron isotopes in multiple planktonic foraminifera species with different depth habitats and their potential to constrain pH and pCO$_2$ gradients"

_Biogeosciences, 2019_

## Short Comment (SC1) · 26 Aug 2019

I read this recent work by Guillermic et al. with great interest. The authors present useful new data from core-top foraminifera, which expands the array of core-top MC-ICPMS data we have in the community. The data are logically presented, and the manuscript is well written for the most part (although there are a fair few typos throughout, which I will leave to the reviewers and copyeditors). These data add more evidence that no modern species of planktic foraminifera measured to date consistently records the d11B of borate faithfully (with implications for palaeo-work), and confirms the pattern of increasing d11B in increasingly shallow-dwelling symbiont-bearing foraminifera, and lighter d11B in deeper-dwelling species.

While in my view this will be worthy of publication, I would like to take this opportunity while the manuscript is still in open discussion to make some suggestions to improve the manuscript.

1) The authors on several occasions highlight the difference between their G. ruber data and the slope obtained by our culture experiments (Lines 385-387; 422-425; lines 525-526; 537-538; 563). In lines 422-425 the picture as presented is particularly confusing, since the authors first suggest "there is a difference in calibrations", then say "this is particularly notable for G. ruber", but then say "the sensitivity of the species analysed are not statistically different". In truth, only the final sentence is true (with the exception of the clause "and are close to unity". A slope of $1.12 \pm 1.67$ is within uncertainty of our culture slope (0.6), and could technically allow a slope as low as -0.55: i.e. there is no significant difference between the slope they suggest and the slope that we observe in culture.

Framing this as a difference seems even more odd given the authors do not draw any distinction between their T. sacculifer slope and that of previous calibrations, because their bounds of uncertainty do not allow it- so it seems logically inconsistent to draw distinctions for ruber where the statistical difference is equally unfounded.

I would suggest the authors go through the manuscript and revise their phrasing to reflect this lack of statistically significant difference. Including "the sensitivity of d11B to pH is not statistically different from unity for G. ruber" as a main conclusion in line 563, for example, implies this study is in disagreement with cultures (which it isn't), and that we should consider a slope of 1 to be potentially suitable for this species. In reality, were we to calculate pCO2 with the slope and intercept that the authors suggest (m=1.12, c=-1.23), the fit of the downcore record of G. ruber from Chalk et al. 2017 with ice-core pCO2 would be considerably worse (see attached Fig. 1). The magnitude of pCO2

change between glacials and interglacials (i.e. the parameter that is driven by the slope of the calibration) is underestimated, with pCO2 too high in glacials by ∼50 ppm. The improved fit of the down-core record with pCO2 from ice-cores when our ruber calibration is used, however (see Chalk et al. 2017), offers support for the shallower-than-unity slope we observed in this species. Incidentally, also, an R-squared of 0.98 for the ruber core-top data presented in this study seems anomalously high relative to the scatter/uncertainty bounds in the dataset- can the authors be clear how this R-squared is computed? Is it the average R-squared of Monte Carlo regressions plotted through datapoints randomly subsampled from within the x- and y-uncertainties? Or is this simply a least-squares linear regression through the central tendencies of the datapoints? The former might be more representative, but as long as the authors are clear about what they are describing that is the main thing.

2) The authors report our generic culture intercept for ruber in their Table 3 (9.52), but erroneously list the size fraction as ∼250 $\mu$m. I would like to point them to Fig. 6 of this 2013 paper, where we give the average size fraction of our cultures to be ∼380 $\mu$m. A suggested size fraction correction on the intercept is given, such that for 300-355$\mu$m it would be 8.87. On this same note, the authors also combine a wide size fraction (250-400$\mu$m) for G. ruber, which given size-related offsets from the culture calibration (as shown in Fig. 6 of Henehan et al. 2013) has the potential over such a large range to skew the data, due to size-related changes in d11B. Can the authors give some estimate as to the distribution of test sizes within their broad sample range, so as to make them more easily comparable to published data?

3) The authors screened for clay contamination using Ti/Ca ratios, as Al/Ca values were difficult to measure with their introduction system. However, they do not provide these data. Clay may carry isotopically-light sorbed d11B with it, and introduce bias towards lighter values. To allow maximum confidence in the data, and see which datapoints if any might have some influence of clay, can the authors please provide the Ti/Ca ratios in Table 2?

4) In Section 4.2.3 (note the paper skips 4.2.4 and goes straight on to 4.2.5?), the authors pool 'deeper-dwelling' foraminiferal species together, but this seems a bit unfounded since these foraminifera don't even have the same symbiont types (crysophyte vs. dinoflagellate), and have quite different ecologies. I'm not convinced there's enough of an a priori reason to even do this in the first place. However, I see that the authors do already concede this may be unfounded.

5) In section 3.9, the authors make no mention of how they calculated pK*B for each foraminifera. I take it they did indeed account for changes in pK*B with temperature, salinity and pressure? It may sound blindingly obvious, but I'm constantly amazed at how many people make this error.. On a similar theme Fig S3 the pH lines are no doubt helpful, but I'm not sure how the authors managed to calculate them, given the pK*B is different for each foram. Is this calculated using the mean pK*B of each of the forams plotted? This figure makes me worry that the authors just chose a single value of pK*B for all forams in all calculations of the paper, which would be wrong.

6) I think the decision to group 'shallow-dwelling' foraminifera (note it is not clearly defined what species this includes in any caption) in Fig. S4 (and in the text where this is referenced) is I think unfounded. It produces a correlation between B/Ca and Borate/DIC, sure, but it's entirely driven by the interspecies difference between ruber and sacculifer, and we know from Kat Allen's work for example that these species have fundamentally different B/Ca-Borate/DIC relationships.. they shouldn't be lumped together in one group. As it is, it makes this look like a carbonate system relationship when on an intra-species level there is no significant correlation with the carbonate system (as we also observed elsewhere).

I hope these comments prove useful and help with the development of the manuscript.

[Figure]

**Fig. 1.**

---

## Referee Comment (RC1) · Jesse Farmer (Referee) · 9 Sep 2019

Guillermic et al. present boron isotope measurements of seven species of planktonic foraminifera from seven coretop sediment samples. In addition to providing new and expanded boron isotope calibrations for these taxa, the authors interpret variations in foraminifer $\delta$11B (and to some extent, B/Ca) data in terms of varying calcification depths and microenvironment influences, with particular focus on how light availability at the depth of calcification may impact symbiont photosynthesis.

[Figure]

The introduction, background, materials and methods, and results are well written and clear, and I think the authors' data are a welcome addition to the $\delta$11B community. However, I have significant reservations on aspects of the discussion that will require detailed revisions before this manuscript is suitable for publication.

Three general comments:

1) Foraminifera depth habitats and thermocline depth. To what extent are differences in foraminifera depth habitats between the different studied oceanographic regions simply a function of variations in thermocline depth? This point could be clarified throughout the manuscript. If true, it seems a particularly important outcome of this study is a need to combine planktonic foraminifera $\delta$11B with thermocline depth reconstructions.

2) In section 5.2 (L482-514), the authors posit a primary control of light availability on foraminiferal $\delta$11B deviations from $\delta$11Bborate. While an interesting idea, unfortunately I think the weight of this section is not supported by the authors' data. This whole discussion is essentially predicated on a single T. sacculifer $\delta$11B measurement from the western Equatorial Pacific, which has anomalously low $\delta$11Bforam value (relative to $\delta$11Bborate) compared to previous studies. The authors use this single observation to make a complex argument about how foraminifera calcification depth impacts light availability, which impacts symbiont photosynthesis, which affects microenvironment pH and hence foram $\delta$11B. Not only does this strike me as insufficient evidence to justify a discussion of this length, there are numerous assumptions and issues within the discussion that require referencing and clarification (see detailed comments below).

3) Section 5.4 gives a rather hasty overview of how measuring $\delta$11B in different foraminifera species can reconstruct the upper water column pH and pCO2 gradients. I believe this is one of the big strengths of this paper and would like to see an expanded discussion of the depth profiles. One concern is that there may be some circularity here. If the authors have calibrated $\delta$11Bforam to modern profiles of $\delta$11Bborate, then by default they would correctly reconstruct the pH and pCO2 profile of the water column

with the calibration dataset. There are no free parameters.

Detailed comments

L49. Change "several" to "many" taxa. L50. Specify "we report $\delta$11B data" L62. Either change to carbon pumps, or specify which pump to which you refer (biological?) L54-61: The primary results are a bit vague. I encourage greater specificity in the key results (but see general comments above).

L64-66. Change to "resulting in declining surface ocean pH". I would caution against calling this decline "steady".

L73. Change reference to Allen and Hönisch, 2012 for formatting consistency.

L81. also spell out DIC

L89-90. remove Rae et al. 2011 reference here, as this study of benthic forams was not intended to directly constrain atmospheric pCO2.

L90 and 93. These are two separate Martínez-Boti et al. (2015) papers and should be referenced as 2015a (Pliocene) and 2015b (eastern Equatorial Pacific & Subantarctic)

L96-97. Delete first sentence, and change second sentence to "In this study, we make critical additions to the emerging pool of boron isotope data for coretop..."

L100. Subscript on 3; e.g., CaCO3

L101 vs. L98. Pick either core-top or coretop throughout the manuscript

L118-120. While interesting, the PETM $\delta$ 18O profile work is quite tangential to the current study and should be removed.

L121. Reword to either "Because planktonic foraminifera species..., it is thus..." or "Planktonic foraminifera species..., therefore it is thus..."

L124-125. Either Palmer and Pearson (1998) pioneered this approach and the "perhaps" should be deleted, or someone else did and should be referenced accordingly.

Please revise.

L128-130. This comes across a bit awkward; suggest rewording to something like "Furthermore, $\delta$11B differences between foraminifera species from the same pH makes the acquisition of more modern…"

L136. Remove "equal to" (repetitive)

L148-153. The terminology for isotopic fractionation factors and fractionations is incorrect. For Klochko et al. (2006), the fractionation factor, $\alpha$B, is 1.0272, and the fractionation, $\varepsilon$B, is the per mil value of 27.2±0.6‰. See, e.g., Table 1 in Farmer et al. (2019) GCA. Please change to correct terminology throughout this paragraph.

L166-167. Benthic foraminifera $\delta$ 11B are only tangentially relevant to the results of this manuscript, so I recommend deleting this clause and associated references. Unless the benthic $\delta$ 11B results directly shed light on your interpretation (as is the case for Amphestegina, below).

L180. Specify that Amphistegina lobifera is a shallow-water, symbiont-bearing benthic foraminifer.

L182. Change "taxon" to "taxa"

L187/Equation 2. To me, this equation is an odd depiction of the cumulative effects of calcification, photosynthesis, respiration and dissolution. Suggest separating into two equations: one for calcification/dissolution, and another for photosynthesis/respiration.

L189-195. Please clarify this paragraph. I think the authors are trying to make the point that, while seawater pH provides a primary control on foraminifer $\delta$11B, microenvironment pH alterations from calcification, respiration, and symbiont photosynthesis also contribute to foraminifer $\delta$11B, and may account for species-specific $\delta$11B offsets. But it is not clear as written.

L212-224. It may be worth noting here (or perhaps earlier) that this manuscript largely

focuses on tropical/subtropical foraminifera.

L255. Were the samples dissolved in 1N HCl or HNO3? And why the two different acid matrices? HCl causes interference issues with ICP-MS measurements. Maybe this does not matter with the microdistillation step, but I'd like to see some explanation (see L260).

L283-286. Recommend splitting this into two sentences, one on the procedural blanks and one on the acid backgrounds/memory effect.

L288. As a field, we need to stop considering NEP as a "standard". It is not sufficiently homogenous to be useful for $\delta^{11}B$ analyses in foraminifera, where precisions of «1‰ are absolutely necessary for the vast majority of paleoceanographic purposes.

L296-298. You should cite the Gutjahr et al. Goldschmidt intercalibration abstract that defines multiple laboratory values for JCp-1: "Boron Isotope Intercomparison Project (BIIP): Development of a new carbonate standard for stable isotopic analyses".

L308-310. I'm surprised that the HF matrix prevents a B memory effect on the Neptune+, but not on the Element XR. To my knowledge, both instruments have effectively the same frontend plasma setup. Can you comment more on this? Does this high B background result from other measurements on the XR, e.g., rock digestions with really high B content? Did you swap out cones, etc? (This comment does not necessarily need to be addressed within the manuscript, I'm just curious).

L344. Rephrase to "Two carbonate system parameters are essential to calculate the entire carbonate system".

L349-350 and Figure 4. Please plot the $\delta^{11}B_{borate}$ uncertainties on Figure 4. If they are too small to be observed, please note that in the figure caption.

L360-366. It would be useful to indicate the general ranges of depths for these different foraminifera here. E.g., L363 "shallow mixed layer (0-100 m), with T. sacculifer living or migrating deeper than G. ruber (down to 125 m)."

L380-382. Except your G. ruber results are not consistent at the low $\delta$11Bborate end; they are $\sim$1‰ lighter than Henehan et al. (2013) found in sediment traps. It is very important that you state this observation because this is the principal reason for your elevated $\delta$11Bforam to $\delta$11Bborate slope relative to Henehan and Sanyal.

L397-398. Careful overinterpreting limited data here. If you calculate $\delta$11Bforam-$\delta$11Bborate for all sacc and w/o sacc specimens, I doubt you will find a significant difference in $\delta$11Bforam elevation between the two T. sacculifer forms.

L400-417. This is well put.

L422-424. See above comment on L380-382.

L427-439. It could be worth making an updated version of Henehan et al. EPSL 2016's Figure 7 and including this in the main text.

L433. $p < 0.05$

L438-439. One option- you could test the influence of sediment core water depth and foraminifer water depth with a multiple linear regression and see if either influence dominates B/Ca.

L453. Suggest starting a new paragraph with the sentence "We find…" and merging with the below paragraph starting on L456.

L457-459 and again L462-463. This is just another way of saying that the depth of the thermocline differs at each location, correct? If so, really what we need are proxies for thermocline depth.

L474-475. Yes, but this might not matter as much for foraminifera $\delta$11B. At higher latitudes, the seasonality of primary production (& hence foraminifera growing seasons) will be more tightly constrained due to the seasonal progression of winter light limitation and intense vertical mixing and summer nutrient limitation.

L478. Specify seasonal $\delta$11Bborate at a fixed depth

L483. Do not use articles when you can be more specific; it creates unnecessary ambiguity. Here, state "foraminifera $\delta$11B" instead of "the $\delta$11B signature".

Section 5.2 (L482-514) detailed comments:

1) You have not yet discussed symbionts in these different foraminifera species until near the end of the section (L509-512). Be explicit about what is known about symbionts in all studies species with an introductory paragraph at the beginning of this section. How biologically similar are the symbiont assemblages in different foraminifera species? Is their concentration, photosynthetic activity, etc. similar? Do they show the same dependence on light intensity to maintain photosynthetic activity?

2) Following on the above, it is not clear to the reader whether "weaker photosynthetic activity" (L487) corresponds to an absence of symbionts, less active symbionts, lower symbiont density, lower light levels, etc. Please clarify.

3) L490. Is symbiont photosynthetic activity a function of light level alone? What about symbiont composition? This comes across as highly speculative without references; please either include references or phrase as a speculation. (In general, it is fine to speculate a little, as long as the reader is aware that you are speculating).

4) Light intensity in the ocean is not a function of water depth alone; turbidity matters quite a lot. Is it reasonable to compare the light intensity in different oceanographic regions as a function of water depth alone?

4) L492. The negative relationship between $\Delta$11B and water depth in Figure S2 is driven only by the low $\delta$11B measured in western equatorial Pacific T. sacculifer. Have you propagated your uncertainty in depth habitat to $\delta$11Bborate in your calculation of $\Delta$11B as plotted in Figure S2? Looking at the western equatorial Pacific (WP7-01), the 80$\pm$20m depth range from CD2 corresponds to a ~1‰ $\delta$11Bborate range (Fig. 4). (If you include the CD3 estimate of 125$\pm$15 m, the total possible $\delta$11Bborate increases to ~1.5‰. Add on the $\delta$11Bforam measurement uncertainty of $\pm$0.22‰ and I cannot see

how you get a total Δ11B uncertainty of <1‰

5) I do not think linear regression is the appropriate test statistical test for the significance of the Δ11B-water depth relationship, because depending on how you performed the regression, it may not account for the uncertainty on each datapoint. A t-test for mean difference between Arabian Sea, Indian Ocean, and western equatorial Pacific T. sacculifer Δ11B would be more robust. I would bet that such a test will indicate no significant Δ11B difference between the different regions given the small sample sizes. If true, this detailed line of discussion is unnecessary; instead, you can present this as an observation that requires future study to confirm or deny.

L525-527. What do you mean by "higher values " here? Slopes of δ11Bforam to δ11Bborate regressions, or Δ11B, or something else?

Any precipitation rate implications need to be very carefully phrased. The higher "values" for G. ruber and T. sacculifer have large uncertainties, so they are probably not robustly higher than other species (unless a statistical test confirms this to be true). Moreover, it is unclear the extent to which the growth rates of different foraminifera species differ from one another. Nevertheless, your point on higher B/Ca sensitivity to borate/bicarbonate in the shallower species is very interesting.

L540-541. There is no Figure 10?

L541-542. Given the speculative nature of the depth/light effect on symbiont photosynthesis, foraminifer microenvironment pH and thus foraminifera δ11B, change to "which may be explained by the deeper depth habitat for these taxa in the WEP, where lower light levels might reduce symbiont photosynthetic activity". Or remove.

L543-548. See general comment #3

Figure 1. Is this necessary?

Figure 2. Please make the text font larger; the labels and annotations are difficult to read at this size.

Figure 3. It would be interesting to also plot thermocline depth, contoured or otherwise. There are global datasets of mixed layer depth (see https://odv.awi.de/data/ocean/mixed-layer-depths/), which should be a reasonable proxy.

Figure 4. See Figure 2 comment and above comment about including $\delta$11Bborate uncertainty.

Figure 7. I really cannot follow this figure at all. Should there be a pH axis?

Figure 8. See Figure 2 comment.

Figure 9. No caption?

---

## Referee Comment (RC2) · Anonymous Referee #2 · 13 Sep 2019

The authors of this manuscript present new boron isotope data for multiple species of planktic foraminifera, which adds data to and expands existing calibrations for many of these species. The manuscript is well written, logically presented, and the results will be of interest to many in the paleoceanography community. While I have mostly minor suggestions for improving the manuscript (detailed below), I have a few more moderate concerns that may improve the manuscript prior to final publication.

Depth habitats of the foraminifers are used to link the habitat to oceanographic conditions at the core locations. The supplement has a fairly thorough explanation of how the

various depths were determined (using d18Oc, MgCa-derived T, and published depth habitats; detailed in Table 6), but I couldn't determine which depth the authors chose to use for each species (or at least is wasn't consistently clear in the text) without seeking the information in Table 7. In the intro of the MS, the authors point the reader to table 3 for the depth habitats but table 3 doesn't include this information. Also, I don't understand why sometimes the authors used published references for depth habitats and in other instances is d18O or Mg/Ca derived temperatures. For example: For core FC-01a, the Sime reference is cited for the depths used for G. ruber, T. sacc, O. universa, but oxygen isotopes are used for P. obliquiloculata and Mg/Ca derived Ts are used for tumida and menardii. Clarification here is needed.

I don't understand why O. universa is considered a deeper dweller in this MS. There are some major assumptions made about why the d11B of this species falls below the 1:1 line – this is also discussed in Henehan et al., 2013. Why O. universa d11B is more like the deeper dweller non-spinose forams is indeed puzzling (esp. since it has same symbionts as G. ruber and G. sacculifer), but I don't think it can be attributed to a deep depth, especially given the size fraction used (>500). The larger size fraction of the samples used would suggest that these are living at a shallower depth (See Spero and Parker, 2003) and likely in the mixed layer. The correct depth habitat will impact the calibration.

Samples were cleaned using the full cleaning method including the reductive step (cite Boyle and Kiegwin 1985/1986 as well, since they developed the method). Why was the reductive step included here? Yu et al., 2007 suggests the reductive step isn't detrimental to B/Ca ratios, but the effect of this cleaning step on for d11B analysis is unknown and according to Rae et al., 2018, the reductive step is typically not used during sample cleaning. There are documented dissolution effects that the authors discuss in the supplement (preferential dissolution of ontogenetic calcite occurs relative to the light d11B of gam calcite) and if cleaning preferentially removes ontogenetic calcite, then the primary d11B signal has been altered by the reductive cleaning. Additionally, the reductive cleaning step IS detrimental to other elements, like Mg/Ca ratios (decreases ratios by up to 15%), which were used to estimate depth habitat for some of the species investigated here. Including the reductive step should be justified with an explanation of how this additional cleaning step could have affected results.

Li/Ca ratios are included in table 2, but never discussed?? Are they relevant for this MS? If not, perhaps remove?

Figure comments: Some of the fonts are very small, though this may be due to the orientation on the screen for the current version. I don't think Biogeosciences has figure font-size recommendations, but generally not smaller than 9pt.

Figure 1: Probably not needed, there are already many figures in the MS Figure 2: Fonts are v. small Figure 4: plots should be enlarged and figure moved to supplement Figure 5: the faint gray lines/symbols are quite hard to see. In print, nearly impossible. Please check font sizes, again they are quite small Figure 6: Check font sizes, they are very small. I recommend forgoing the use of yellow in figures. Too hard to see. Figure 7: I'm not sure I find this figure useful. Light percent? Perhaps change to PAR or something that can be related to a preferred habitat? Figure 8: I don't have specific suggestions for this figure, but there is a LOT of data with very small fonts. Would it be better to break this up into basins and put in the supplement? Figure 9: lacks a caption Figure 10 is mentioned I the text, but there is no figure 10.

Table 3: In the text on line 329 it is stated that chemo stratigraphic data is used to constrain depths, but this information is not summarized in the table, probably just a typo?

Other suggestions: Line 56-57: Not sure I agree with the statement in the abstract that the other species follow O. universa because of light limitation by symbiont bearing foraminifera. All of the deep dwellers have symbionts, all live in the photic zone.

Line 128: This sentence is poorly structured.

Secition 2.4: Origin of biological fractionation Paragraph beginning on line 172: very speculative and based upon benthic foraminifer experiments.

Section 2.5: The annual vs seasonal preferences for forams is largely dependent on temperatures. For example, in some regions G. ruber and G. sacculifer can be present throughout the year if T > 25C, but will have a summer/fall preference when T drops below 15. Their choice on seasonal vs. annual presence of these species will affect the hydrographic data used and perhaps impact results.

Section 3.2: Size fractions listed in this paragraph don't agree with size fractions in the Table.

Section 3.8: This section could/should include some of the information in the supplement regarding the depth used to obtain hydrographic information.

Line 339: Forams don't migrate in the water column (See Meiland et al., 2019), but deep dwellers may crust at depth during the END of their lifecycle, this should be clarified. This is later explained correctly (lines 452-453).

Lines 509-514: The concept of facultative symbiosis is outdated – all forams with symbionts are likely obligate and not facultative. See https://www.biogeosciences.net/16/3377/2019/. G. tumida doesn't have symbionts at all, so why does it align with the other species? Please discuss.

---

## Author Comment (AC1) · 23 Oct 2019

We wish to thank Dr. Jesse Farmer for his thorough review of our manuscript and his helpful comments. We believe that we addressed all of the major comments indicated by Dr. Farmer as indicated in the discussion below and the updated manuscript (see supplement).

Comment 1: Foraminifera depth habitats and thermocline depth. To what extent are differences in foraminifera depth habitats between the different studied oceanographic regions simply a function of variations in thermocline depth? This point could be clarified throughout the manuscript. If true, it seems a particularly important outcome of this study is a need to combine planktonic foraminifera _11B with thermocline depth reconstructions.

Response 1: We have added a statement to line 460 acknowledging that "Factors including variations in thermocline depth can impact depth habitats for some taxa." Variations in thermocline depth are certainly one of several factors that impact the differences in foraminiferal depth habitats, but not the only factor, both in the present and in the past. Only some species follow the thermocline depth (like G. tumida), while others (such as N. dutertrei) are not and are usually found around the same water depth. So it will be dependent on the studied species.

Comment 2: In section 5.2 (L482-514), the authors posit a primary control of light availability on foraminiferal _11B deviations from _11Bborate. While an interesting idea, unfortunately I think the weight of this section is not supported by the authors' data. This whole discussion is essentially predicated on a single T. sacculifer _11B measurement from the western Equatorial Pacific, which has anomalously low _11Bforam value (relative to _11Bborate) compared to previous studies. The authors use this single observation to make a complex argument about how foraminifera calcification depth impacts light availability, which impacts symbiont photosynthesis, which affects microenvironment pH and hence foram _11B. Not only does this strike me as insufficient evidence to justify a discussion of this length, there are numerous assumptions and issues within the discussion that require referencing and clarification (see detailed comments below).

Response 2: We understand this point. Further support for this comes from data for ODP Sites 806A and 807A which we have added to this study. The Holocene data for Sites 806A and record similar low values to what we reported for site WPO7. This strengthens and justifies the length of the discussion.

Comment 3: Section 5.4 gives a rather hasty overview of how measuring _11B in different foraminifera species can reconstruct the upper water column pH and pCO2 gradients. I believe this is one of the big strengths of this paper and would like to see an expanded discussion of the depth profiles. One concern is that there may be some circularity here. If the authors have calibrated _11Bforam to modern profiles of _11Bborate, then by default they would correctly reconstruct the pH and pCO2 profile of the water column with the calibration dataset. There are no free parameters.

Response 3: In order to avoid this circularity, we recalculated one calibration for each species excluding the site of interest. Meaning that the recalculated site is not into the calibration. We also use this subset of calibration data to reconstruct vertical profiles in figure 9 which validates the work. When we compare both differences between calibration with or avoiding circularity (figure below or S6), the difference of $\delta$11B is on average (0.8% on $\delta$11Bborate, 0.2% on pH and 5% on pCO2). The results are similar to the whole calibration dataset, subset of the data for calibration purposes which validates this work. Maximum divergence can be observed for G. ruber and T. sacculifer when utilizing the calibration [2] and [4]. We also have added a statement to Line 618-620 stating "Reconstruction of seawater pH and carbonate system parameters is achievable using foraminiferal $\delta$11B but additional coretop and down-core studies reconstructing depth profiles will be needed in order to further verify those calibrations."

Detailed review

RC1: L148-153. The terminology for isotopic fractionation factors and fractionations is incorrect. For Klochko et al. (2006), the fractionation factor, _B, is 1.0272, and the fractionation, "B, is the per mil value of 27.2_0.6‰§ËŹ ee, e.g., Table 1 in Farmer et al. (2019) GCA. Please change to correct terminology throughout this paragraph.

Response: Corrected Line 146 "given by the fraction factor ($\alpha$). The fractionation ($\varepsilon$) between B(OH)3 and B(OH) 4- of 27.2 $\pm$ 0.6 ‰ has been empirically determined by Klochko et al., (2006) in seawater."

[Figure]

RC1: L166-167. Benthic foraminifera _ 11B are only tangentially relevant to the results of this manuscript, so I recommend deleting this clause and associated references. Unless the benthic _ 11B results directly shed light on your interpretation (as is the case for Amphestegina, below).

Response: I removed the benthic species. Line 160-163: "At present, culture and core-top calibrations have been published for several planktonic species including Trilobatus sacculifer, Globigerinoides ruber, Globigerina bulloides, Neogloboquadrina pachyderma, Orbulina universa (Foster et al., 2008; Henehan et al., 2013; Henehan et al., 2015; Sanyal et al., 1996; Sanyal et al., 2001)."

RC1: L212-224. It may be worth noting here (or perhaps earlier) that this manuscript largely focuses on tropical/subtropical foraminifera.

Response: We have added this information in the abstract (line 50).

RC1: L255. Were the samples dissolved in 1N HCl or HNO3? And why the two different acid matrices? HCl causes interference issues with ICP-MS measurements. Maybe this does not matter with the microdistillation step, but I'd like to see some explanation (see L260).

Response: The samples were dissolved in 1N HCl, results from the microdistillation in order to get the entire dissolution of the sample, including Fe-Mn oxide and hydroxides (reductive acid). Different dissolution after microdistillation resulting in mix HCl/HNO3 matrices were tested resulting in no significant $\delta$11B differences on standards, however, working with really low concentration samples resulting in running samples in HCl matrix might be an issue at some point.

RC1: L283-286. Recommend splitting this into two sentences, one on the procedural blanks and one on the acid backgrounds/memory effect.

Response: Lines 279-286: "The sample size for boron isotope analyses typically ranged from 10 ppb B ($\sim$5 ng B) to 20 ppb B samples ($\sim$10 ng B). Instrumental sensitivity for 11B was 17 mV/ppb B (eg. 170 mV for 10ppb B) in wet plasma at $50\mu$l/min sample aspiration rate. Intensity of 11B for a sample at 10ppb B was typically 165mV $\pm$ 5mV closely matched the 170mV $\pm$ 5mV of the standard. Due to the low boron content of the samples extreme care was taken to avoid boron contamination during sample preparation and reduce memory effect during analysis. Procedural boron blanks ranged from 15pg B to 65 pg B (contributed to less than <1% of the sample signal). The acid blank during analyses was measured at $\leq$ 1mV on the 11B, meaning a contribution < 1% of the sample intensity, no memory effect was observed within and across sessions."

RC1: L288. As a field, we need to stop considering NEP as a "standard". It is not sufficiently homogenous to be useful for _11B analyses in foraminifera, where precisions of Âń1‰ are absolutely necessary for the vast majority of paleoceanographic purposes.

Response: We agree but due to the specifics of these analytical runs, unfortunately the NEP was the only "relevant" carbonate standard we had. We also ran seawaters as well along with carbonate standards though they did not have the same matrix.

RC1: L349-350 and Figure 4. Please plot the 11Bborate uncertainties on Figure 4. If they are too small to be observed, please note that in the figure caption.

Response: We recalculated the uncertainties based on the errors of Alkalinity and DIC from the Glodap database. Temperature, salinity and pressure were taking into account for all calculations.

RC1: L308-310. I'm surprised that the HF matrix prevents a B memory effect on the Neptune+, but not on the Element XR. To my knowledge, both instruments have effectively the same frontend plasma setup. Can you comment more on this? Does this high B background result from other measurements on the XR, e.g., rock digestions with really high B content? Did you swap out cones, etc? (This comment does not necessarily need to be addressed within the manuscript, I'm just curious).

Response: HF matrix does prevent memory effects, but with the Element XR we are measuring X/Ca ratios, so working only in HF matrix will directly precipitate CaF2, a mix of HF and HNO3 prevents precipitation. The boron background comes from other measurements. The shared instrument is used to run a lot of seawater samples, rocks etc. . . However, the cones were properly cleaned. We suspect the extraction lengt was the issue. You can still try to decontaminate with HF solution before conditioning the cones but results are not sufficient; One way to decrease the boron background is to condition the cones with the [Ca] standards (HNO3 and HF) for few hours, and switching the boron concentration of the standards.

RC1: L380-382. Except your G. ruber results are not consistent at the low 11Bborate end; they are 1‰ lighter than Henehan et al. (2013) found in sediment traps. It is very important that you state this observation because this is the principal reason for your elevated 11Bforam to 11Bborate slope relative to Henehan and Sanyal.

Response:

Lines 381-391 "Our results for G. ruber (Fig. 5) are in good agreement with published data from other core-tops, sediment traps, tows, and culture experiments for $\delta$11Bborate>19 ‰ (Foster et al., 2008, Henehan et al., 2013, Raitzsch et al., 2018). However, for $\delta$11Bborate<19 ‰ our results show lighter $\delta$11Bcarbonate compared to published values. Whilst this species has been widely studied previously, the sites selected in this study allow us to extend the calibration. The positive offset from the 1:1 curve has been explained by the high photosynthetic activity (Hönisch et al., 2003; Zeebe et al., 2003). Two calibrations have been derived. Utilizing only our data, the sensitivity of $\delta$11Bcarbonate to $\delta$11Bborate of our linear regression is not statistically different from 1 and do not follow the low sensitivity trend of the culture experiments from Sanyal et al., (2001) or Henehan et al., (2013), (p<0.05). The uncertainty on this regression is important due to our small dataset and not inconsistent with the second calibration made compiling all data from literature. The sensitivity of this regression is similar (e.g. 0.46 ($\pm$0.34) to the one recently published by Raitzsch et al., (2018) (e.g.

0.45 ($\pm$0.16), Table 3)."

It was also highlighted by Michael Henehan (SC1), he was especially asking about the size fraction of our samples, I replied: "Most of the samples have been picked in the 250-300 size fraction (average weight/ shell of 11 $\pm$ 4 $\mu$g (n=4, SD) only when measurements were realized), we chose a restrained size fraction to avoid this size-related variability (at least in our calibration), we also chose this lower size fraction because some of the sites did not present shells in the higher size fractions and we wanted to stay consistent. Now, when compiling all the data this variability is not constrained. From your paper, our weight/ shell variability could lead to an offset up to $\sim$1% that we acknowledged can explain most of the variability for our G. ruber data. I have added line 582 "Henehan et al., (2013) reported a lighter $\delta$11B with smaller test size, our sample add a weight/shell of 11 $\pm$ 4 $\mu$g (n=4, SD) which could also explain this variability."

RC1: L427-439. It could be worth making an updated version of Henehan et al. EPSL 2016's Figure 7 and including this in the main text.

We updated made this figure (below) and added it to the main text.

RC1: L438-439. One option- you could test the influence of sediment core water depth and foraminifer water depth with a multiple linear regression and see if either influence dominates B/Ca.

Response: At the end, linear regression was only significant when plotting B/Ca and the calcification depth not significant with water depth. From those results, the calcification depth is the parameter influencing B/Ca.

RC1: L457-459 and again L462-463. This is just another way of saying that the depth of the thermocline differs at each location, correct? If so, really what we need are proxies for thermocline depth.

Response: Please see our response to Comment 1.

RC1: L474-475. Yes, but this might not matter as much for foraminifera _11B. At higher latitudes, the seasonality of primary production (& hence foraminifera growing seasons) will be more tightly onstrained due to the seasonal progression of winter light limitation and intense vertical mixing and summer nutrient limitation.

Response: Yes, I agree, I changed for:

Line 478-481: "As Raitzsch et al, (2018) highlight, seasonality might be more important at high latitude sites where seasonality is more marked, however, the seasonality of primary production will also be more tightly constrained due to the seasonal progression of winter light limitation and intense vertical mixing and summer nutrient limitation."

RC1:1) You have not yet discussed symbionts in these different foraminifera species until near the end of the section (L509-512). Be explicit about what is known about symbionts in all studies species with an introductory paragraph at the beginning of this section. How biologically similar are the symbiont assemblages in different foraminifera species? Is their concentration, photosynthetic activity, etc. similar? Do they show the same dependence on light intensity to maintain photosynthetic activity? 2) Following on the above, it is not clear to the reader whether "weaker photosynthetic activity" (L487) corresponds to an absence of symbionts, less active symbionts, lower symbiont density, lower light levels, etc. Please clarify. 3) L490. Is symbiont photosynthetic activity a function of light level alone? What about symbiont composition? This comes across as highly speculative without references; please either include references or phrase as a speculation. (In general, it is fine to speculate a little, as long as the reader is aware that you are speculating).

Response: We have expanded the discussion around the types of symbionts/photosynthetic activity/symbiont density. Although there is not a lot of data in the literature, there is a recent study from Takagi et al., (2019) – published this September – that was helpful in strengthening the discussion.

Note it seems that the nutrient concentrations were not impacting the photosynthesis activity for T. sacculifer, and that is why we did not mention it.

The following text was added,

Lines 489-501: "In planktonic foraminifera, algal symbiosis is the more common symbiotic relationship. For most of planktonic foraminifera, the host presents only one species of symbionts (Gast and Caron, 2001). The family Globigerinidae, including G. ruber, T. sacculifer and O. universa, commonly have dinoflagellates or chrysophyte algal symbionts (Anderson and Be, 1976; Spero, 1987). The families Pulleniatinidae, Globorotaliidae, including N. dutertrei, P. obliquiloculata, G. menardii and G. tumida, have chrysophyte algal symbionts (Gastrich, 1988). The relationship between the symbionts and the host is complex by nature. Nevertheless, this symbiotic relationship provides energy (Hallock, 1981b) and promotes calcification of the foraminifera (Duguay, 1983; Erez et al., 1983) by providing the inorganic carbon to the host (Jorgensen et al., 1985). Also, for T. sacculifer and O. universa photosynthesis increases with higher insolation (Jorgensen et al., 1985; Rink et al., 1998). Dinoflagellate-bearing foraminifera (G. ruber, T. sacculifer and O. universa) tend to have a higher symbiont density and photosynthesis activity while P. obliquiloculata, G. menardii and N. dutertrei have lowered symbiont density and P. obliquiloculata, N. dutertrei lower photosynthetic activity (Takagi et al., 2019). P. obliquiloculata showed the minimum symbiont density and photosynthetic activity (Takagi et al., 2019)."

Lines 526-530: "Also, T. sacculifer has the potential to support more photosynthesis due to its higher symbiont density. Higher photosynthetic activity is observed compared to other species potentially supporting higher symbiont/host interactions. Those results could be in line with a greater sensitivity of T. sacculifer photosynthetic activity with changes in insolation/water depth. It can also be noted that this species presents the most important variations in symbiont density versus its test size. Microenvironment pH results for N. dutertrei, G. menardii, G. tumida, are similar to O. universa and suggest a threshold for respiration driven $\delta$11B signature. This threshold can be driven by a change of photosynthetic activity due to lower light intensity at deeper depth and/or a change in the symbiont assemblage with non-dinoflagellate symbionts at deeper depth. We can explain this threshold because deep dweller species do not experience important changes of insolation at those depths so their microenvironments should be respiration driven and relatively stable. We can also note that P. obliquiloculata which has the lowest symbiont density and photosynthetic activity has the lowered microenvironment pH compared to other deeper dweller species supporting this respiration driven microenvironment."

RC1: 4) Light intensity in the ocean is not a function of water depth alone; turbidity matters quite a lot. Is it reasonable to compare the light intensity in different oceanographic regions as a function of water depth alone?

Response: This is true, but we think it is still reasonable for the calculations because we used a light attenuation coefficient from Wang et al., (2008) which should take most of the parameters into account even if no variability is constrained.

RC1: 4) L492. The negative relationship between _11B and water depth in Figure S2 is driven only by the low _11B measured in western equatorial Pacific T. sacculifer. Have you propagated your uncertainty in depth habitat to _11Bborate in your calculation of _11B as plotted in Figure S2? Looking at the western equatorial Pacific (WP7-01), the 80_20m depth range from CD2 corresponds to a _1‰ _11Bborate range (Fig. 4). (If you include the CD3 estimate of 125_15 m, the total possible _11Bborate increases to _1.5‰Ȧdd on the _11Bforam measurement uncertainty of _0.22‰ and I cannot see how you get a total _11B uncertainty of <1‰

Response: The uncertainties on the $\delta$11Bborate are calculated using Michael Henehan's code which takes into account the uncertainties on pH, temperature, salinity and $\delta$11Bseawater. Uncertainties of pH, temperature and salinity were calculated integrating the parameters between uncertainties of the calcification depth. This is the same approach that was used in Raitzsch et al. (2018). Then the uncertainty on $\Delta$11B was measured by the square root of the sum of the square uncertainties on $\delta$11Bborate and $\delta$11Bcarbonate.

RC1: 5) I do not think linear regression is the appropriate test statistical test for the significance of the _11B-water depth relationship, because depending on how you performed the regression, it may not account for the uncertainty on each datapoint. A t-test for mean difference between Arabian Sea, Indian Ocean, and western equatorial Pacific T. sacculifer _11B would be more robust. I would bet that such a test will indicate no significant _11B difference between the different regions given the small sample sizes. If true, this detailed line of discussion is unnecessary; instead, you can present this as an observation that requires future study to confirm or deny.

Response: We have added two data points from sites 806A and 807A which makes the t-test more robust.

For $\delta$11B. Unpaired t-test between the WEP and the rest of the samples for T. sacculifer (w/o sacc) is significantly different (p=0.01). For $\Delta$11B. Unpaired t-test between the WEP and the rest of the samples for T. sacculifer (w/o sacc) is not significantly different (p=0.067). If now we look at both G. ruber and T. sacculifer (w/o sacc) which are presented in the plot, difference between Arabian Sea+Indian Ocean and WEP is significantly different (p=0.0075). If we also look at T. sacculifer (w/o sacc and sacc) + G.ruber between Arabian Sea+Indian Ocean and WEP the difference is significantly different (p=0.0427).

Lines 520-522: "Especially, we observe an important decrease of $\delta$11B in the WEP for T. sacculifer (w/o sacc), significantly different from the other sites (p<0.05) and a calculated $\Delta$11B of G. ruber, T. sacculifer (w/o sacc and sacc) significantly lower in the WEP compared to the other sites (p<0.05)."

RC1: Any precipitation rate implications need to be very carefully phrased. The higher "values" for G. ruber and T. sacculifer have large uncertainties, so they are probably not robustly higher than other species (unless a statistical test confirms this to be true). Moreover, it is unclear the extent to which the growth rates of different foraminifera species differ from one another. Nevertheless, your point on higher B/Ca sensitivity to borate/bicarbonate in the shallower species is very interesting.

Response: We mitigated this paragraph.

Lines 556-567: "When combining all literature data, T. sacculifer and G. ruber have sensitivities of $\delta$11Bcarbonate to $\delta$11Bborate of 0.83 $\pm$ 0.48 and 0.46 $\pm$ 0.34 respectively in line with previous literature and paleo-CO2 reconstructions. Also, if we only take into account our data, the observation that the sensitivity of $\delta$11Bcarbonate to $\delta$11Bborate are not statistically different from unity for most of the species investigated we can speculate that for these taxa, changes in precipitation rate and contributions of boric acid are not likely to be important. If considering only the data from this study, G. ruber (1.12 $\pm$ 1.67) and T. sacculifer (1.38$\pm$ 1.35) present higher sensitivities of $\delta$11Bcarbonate to $\delta$11Bborate. We can then again speculate that the observed high values for $\delta$11Bcarbonate at high seawater pH can be due to higher precipitation rates. We note this could also be consistent with the higher sensitivity of B/Ca signatures in these two surface dwelling species to ambient [B(OH)4-]/[HCO3-] relative to deeper dwelling species. As indicated by Farmer et al., (2019), studies of calcite precipitation rates in foraminifera could help to test this hypothesis and improve our understanding of the fundamental basis of boron-based proxies."

RC1: L525-527. What do you mean by "higher values "here? Slopes of $\delta$11Bforam to $\delta$11Bborate regressions, or $\Delta$11B, or something else?

Response: Lines 561-562:" G. ruber (1.12 $\pm$ 1.67) and T. sacculifer (1.38$\pm$ 1.35) present higher sensitivities of $\delta$11Bcarbonate to $\delta$11Bborate."

RC1: L541-542. Given the speculative nature of the depth/light effect on symbiont photosynthesis, foraminifer microenvironment pH and thus foraminifera _11B, change to "which may be explained by the deeper depth habitat for these taxa in the WEP, where lower light levels might reduce symbiont photosynthetic activity". Or remove.

Response: Changed.

RC1: Fig.2/Fig.4

Response: Front made bigger

RC1: Figure 7 (or below). I really cannot follow this figure at all. Should there be a pH axis?

Response: For more clarity, we combined in Figure 8 (or below) the $\Delta$microenvironment pH and this depth profile. This depth profile only aims to explain the low $\delta$11B in the WEP. This is based on the calculation of insolation needed to explain the decrease in $\Delta$pH0, $\Delta$pH1 and $\Delta$pH2 observed from the $\delta$11B. This is an independent way to calculate the depth habitat explaining the $\delta$11B. The insolation is calculated based on Jorgensen et al., (1985). I found the insolation needed to decrease of $\Delta$pH0, $\Delta$pH1 and $\Delta$pH2 and converted in % insolation in the water column. What is interesting is that the decrease of insolation around 125m can explain the low $\delta$11B values.

Please also note the supplement to this comment:
https://www.biogeosciences-discuss.net/bg-2019-266/bg-2019-266-AC1-supplement.pdf

———————————————

[Figure]

**Fig. 1.**

**Fig. 2.**

[Figure]

**Fig. 3.**

**Supplement:**

We wish to thank Dr. Jesse Farmer for his thorough review of our manuscript and his helpful comments. We believe that we addressed all of the major comments indicated by Dr. Farmer as indicated in the discussion below and the updated document.

The updated manuscript and figures can be found at the end after this response.

**Comment 1:** Foraminifera depth habitats and thermocline depth. To what extent are differences in foraminifera depth habitats between the different studied oceanographic regions simply a function of variations in thermocline depth? This point could be clarified throughout the manuscript. If true, it seems a particularly important outcome of this study is a need to combine planktonic foraminifera _11B with thermocline depth reconstructions.

*Response 1: We have added a statement to line 460 acknowledging that "Factors including variations in thermocline depth can impact depth habitats for some taxa." Variations in thermocline depth are certainly one of several factors that impact the differences in foraminiferal depth habitats, but not the only factor, both in the present and in the past. Only some species follow the thermocline depth (like G. tumida), while others (such as N. dutertrei) are not and are usually found around the same water depth. So it will be dependent on the studied species.*

**Comment 2:** In section 5.2 (L482-514), the authors posit a primary control of light availability on foraminiferal _11B deviations from _11Bborate. While an interesting idea, unfortunately I think the weight of this section is not supported by the authors' data. This whole discussion is essentially predicated on a single T. sacculifer _11B measurement from the western Equatorial Pacific, which has anomalously low _11Bforam value (relative to _11Bborate) compared to previous studies. The authors use this single observation to make a complex argument about how foraminifera calcification depth impacts light availability, which impacts symbiont photosynthesis, which affects microenvironment pH and hence foram _11B. Not only does this strike me as insufficient evidence to justify a discussion of this length, there are numerous assumptions and issues within the discussion that require referencing and clarification (see detailed comments below).

*Response 2: We understand this point. Further support for this comes from data for ODP Sites 806A and 807A which we have added to this study. The Holocene data for Sites 806A and record similar low values to what we reported for site WPO7. This strengthens and justifies the length of the discussion.*

**Comment 3:** Section 5.4 gives a rather hasty overview of how measuring _11B in different foraminifera species can reconstruct the upper water column pH and pCO2 gradients. I believe this is one of the big strengths of this paper and would like to see an expanded discussion of the depth profiles. One concern is that there may be some circularity here. If the authors have calibrated _11Bforam to modern profiles of _11Bborate, then by default they would correctly reconstruct the pH and pCO2 profile of the water column with the calibration dataset. There are no free parameters.

*Response 3: In order to avoid this circularity, we recalculated one calibration for each species excluding the site of interest. Meaning that the recalculated site is not into the calibration. We also use this subset of calibration data to reconstruct vertical profiles in figure 9 which validates the work. When we compare both differences between calibration with or avoiding circularity (figure below or S6), the difference of $\delta 11B$ is on average (0.8% on $\delta^{11}B_{borate}$, 0.2% on pH and 5% on $pCO_2$). The results are similar to the whole*

*calibration dataset, subset of the data for calibration purposes which validates this work. Maximum divergence can be observed for G. ruber and T. sacculifer when utilizing the calibration [2] and [4].*

[Figure]

*We also have added a statement to Line 618-620 stating "Reconstruction of seawater pH and carbonate system parameters is achievable using foraminiferal δ11B but additional coretop and down-core studies reconstructing depth profiles will be needed in order to further verify those calibrations."*

**Detailed review**

**RC1:** L148-153. The terminology for isotopic fractionation factors and fractionations is incorrect. For Klochko et al. (2006), the fractionation factor, _B, is 1.0272, and the fractionation, "B, is the per mil value of 27.2_0.6‰S˙ ee, e.g., Table 1 in Farmer et al. (2019) GCA. Please change to correct terminology throughout this paragraph.

**Response:** *Corrected*

*Line 146 "given by the fraction factor (α). The fractionation (ε) between $B(OH)_3$ and $B(OH)_4^-$ of 27.2 ± 0.6 ‰ has been empirically determined by Klochko et al., (2006) in seawater."*

**RC1:** L166-167. Benthic foraminifera _ 11B are only tangentially relevant to the results of this manuscript, so I recommend deleting this clause and associated references. Unless the benthic _ 11B results directly shed light on your interpretation (as is the case for Amphestegina, below).

**Response:** *I removed the benthic species.*
*Line 160-163: "At present, culture and core-top calibrations have been published for several planktonic species including Trilobatus sacculifer, Globigerinoides ruber, Globigerina bulloides, Neogloboquadrina pachyderma, Orbulina universa (Foster et al., 2008; Henehan et al., 2013; Henehan et al., 2015; Sanyal et al., 1996; Sanyal et al., 2001)."*

**RC1:** L212-224. It may be worth noting here (or perhaps earlier) that this manuscript largely focuses on tropical/subtropical foraminifera.

**Response:** *We have added this information in the abstract (line 50).*

**RC1:** L255. Were the samples dissolved in 1N HCl or HNO3? And why the two different acid matrices? HCl causes interference issues with ICP-MS measurements. Maybe this does not matter with the microdistillation step, but I'd like to see some explanation (see L260).

**Response:** *The samples were dissolved in 1N HCl, results from the microdistillation in order to get the entire dissolution of the sample, including Fe-Mn oxide and hydroxides (reductive acid).*
*Different dissolution after microdistillation resulting in mix HCl/HNO₃ matrices were tested resulting in no significant $\delta^{11}B$ differences on standards, however, working with really low concentration samples resulting in running samples in HCl matrix might be an issue at some point.*

**RC1:** L283-286. Recommend splitting this into two sentences, one on the procedural blanks and one on the acid backgrounds/memory effect.

**Response:** *Lines 279-286: "The sample size for boron isotope analyses typically ranged from 10 ppb B (~5 ng B) to 20 ppb B samples (~10 ng B). Instrumental sensitivity for 11B was 17 mV/ppb B (eg. 170 mV for 10ppb B) in wet plasma at 50μl/min sample aspiration rate. Intensity of 11B for a sample at 10ppb B was typically 165mV ± 5mV closely matched the 170mV ± 5mV of the standard. Due to the low boron content of the samples extreme care was taken to avoid boron contamination during sample preparation and reduce memory effect during analysis. Procedural boron blanks ranged from 15pg B to 65 pg B (contributed to less than <1% of the sample signal). The acid blank during analyses was measured at ≤ 1mV on the 11B, meaning a contribution < 1% of the sample intensity, no memory effect was observed within and across sessions."*

**RC1:** L288. As a field, we need to stop considering NEP as a "standard". It is not sufficiently homogenous to be useful for _11B analyses in foraminifera, where precisions of «1‰ are absolutely necessary for the vast majority of paleoceanographic purposes.

**Response:** *We agree but due to the specifics of these analytical runs, unfortunately the NEP was the only "relevant" carbonate standard we had. We also ran seawaters as well along with carbonate standards though they did not have the same matrix.*

**RC1:** L349-350 and Figure 4. Please plot the 11Bborate uncertainties on Figure 4. If they are too small to be observed, please note that in the figure caption.

**Response:** *We recalculated the uncertainties based on the errors of Alkalinity and DIC from the Glodap database. Temperature, salinity and pressure were taking into account for all calculations.*

**RC1:** L308-310. I'm surprised that the HF matrix prevents a B memory effect on the Neptune+, but not on the Element XR. To my knowledge, both instruments have effectively the same frontend plasma setup. Can you comment more on this? Does this high B background result from other measurements on the XR, e.g., rock digestions with really high B content? Did you swap out cones, etc? (This comment does not necessarily need to be addressed within the manuscript, I'm just curious).

**Response:** *HF matrix does prevent memory effects, but with the Element XR we are measuring X/Ca ratios, so working only in HF matrix will directly precipitate CaF₂, a mix of HF and HNO₃ prevents precipitation. The boron background comes from other measurements. The shared instrument is used to run a lot of seawater samples, rocks etc... However, the cones were properly cleaned. We suspect the extraction lengt was the issue. You can still try to decontaminate with HF solution before conditioning the cones but results are not sufficient; One way to decrease the boron background is to condition the cones with the [Ca] standards (HNO3 and HF) for few hours, and switching the boron concentration of the standards.*

**RC1:** L380-382. Except your G. ruber results are not consistent at the low 11Bborate end; they are 1‰ lighter than Henehan et al. (2013) found in sediment traps. It is very important that you state this observation because this is the principal reason for your elevated 11Bforam to 11Bborate slope relative to Henehan and Sanyal.

**Response:**

*Lines 381-391 "Our results for G. ruber (Fig. 5) are in good agreement with published data from other core-tops, sediment traps, tows, and culture experiments for δ11Bborate>19 ‰ (Foster et al., 2008, Henehan et al., 2013, Raitzsch et al., 2018). However, for δ11Bborate<19 ‰ our results show lighter δ11Bcarbonate compared to published values. Whilst this species has been widely studied previously, the sites selected in this study allow us to extend the calibration. The positive offset from the 1:1 curve has been explained by the high photosynthetic activity (Hönisch et al., 2003; Zeebe et al., 2003). Two calibrations have been derived. Utilizing only our data, the sensitivity of δ11Bcarbonate to δ11Bborate of our linear regression is not statistically different from 1 and do not follow the low sensitivity trend of the culture experiments from Sanyal et al., (2001) or Henehan et al., (2013), (p<0.05). The uncertainty on this regression is important due to our small dataset and not inconsistent with the second calibration made compiling all data from literature. The sensitivity of this regression is similar (e.g. 0.46 (±0.34) to the one recently published by Raitzsch et al., (2018) (e.g. 0.45 (±0.16), Table 3)."*

*It was also highlighted by Michael Henehan (SC1), he was especially asking about the size fraction of our samples, I replied:*
*"Most of the samples have been picked in the 250-300 size fraction (average weight/ shell of 11 ± 4 µg (n=4, SD) only when measurements were realized), we chose a restrained size fraction to avoid this size-related variability (at least in our calibration), we also chose this lower size fraction because some of the sites did not present shells in the higher size fractions and we wanted to stay consistent. Now, when compiling all the data this variability is not constrained.*
*From your paper, our weight/ shell variability could lead to an offset up to ~1% that we acknowledged can explain most of the variability for our G. ruber data.*
*I have added line 582 "Henehan et al., (2013) reported a lighter δ11B with smaller test size, our sample add a weight/shell of 11 ± 4 µg (n=4, SD) which could also explain this variability."*

**RC1:** L427-439. It could be worth making an updated version of Henehan et al. EPSL 2016's Figure 7 and including this in the main text.

*We updated made this figure (below) and added it to the main text.*

[Figure]

**RC1:** L438-439. One option- you could test the influence of sediment core water depth and foraminifer water depth with a multiple linear regression and see if either influence dominates B/Ca.

**Response:** *At the end, linear regression was only significant when plotting B/Ca and the calcification depth not significant with water depth. From those results, the calcification depth is the parameter influencing B/Ca.*

**RC1:** L457-459 and again L462-463. This is just another way of saying that the depth of the thermocline differs at each location, correct? If so, really what we need are proxies for thermocline depth.

**Response:** *Please see our response to Comment 1.*

**RC1:** L474-475. Yes, but this might not matter as much for foraminifera _11B. At higher latitudes, the seasonality of primary production (& hence foraminifera growing seasons) will be more tightly onstrained due to the seasonal progression of winter light limitation and intense vertical mixing and summer nutrient limitation.

**Response:** *Yes, I agree, I changed for:*

*Line 478-481: "As Raitzsch et al, (2018) highlight, seasonality might be more important at high latitude sites where seasonality is more marked, however, the seasonality of primary production will also be more tightly constrained due to the seasonal progression of winter light limitation and intense vertical mixing and summer nutrient limitation."*
* * *
**RC1:**1) You have not yet discussed symbionts in these different foraminifera species until near the end of the section (L509-512). Be explicit about what is known about symbionts in all studies species with an introductory paragraph at the beginning of this section. How biologically similar are the symbiont assemblages in different foraminifera species? Is their concentration, photosynthetic activity, etc. similar? Do they show the same dependence on light intensity to maintain photosynthetic activity?

2) Following on the above, it is not clear to the reader whether "weaker photosynthetic activity" (L487) corresponds to an absence of symbionts, less active symbionts, lower symbiont density, lower light levels, etc. Please clarify.

3) L490. Is symbiont photosynthetic activity a function of light level alone? What about symbiont composition? This comes across as highly speculative without references; please either include references or phrase as a speculation. (In general, it is fine to speculate a little, as long as the reader is aware that you are speculating).

**Response:** *We have expanded the discussion around the types of symbionts/photosynthetic activity/symbiont density. Although there is not a lot of data in the literature, there is a recent study from Takagi et al., (2019) – published this September – that was helpful in strengthening the discussion.*

*Note it seems that the nutrient concentrations were not impacting the photosynthesis activity for T. sacculifer, and that is why we did not mention it.*

*The following text was added,*

*Lines 489-501: "In planktonic foraminifera, algal symbiosis is the more common symbiotic relationship. For most of planktonic foraminifera, the host presents only one species of symbionts (Gast and Caron, 2001). The family Globigerinidae, including G. ruber, T. sacculifer and O. universa, commonly have dinoflagellates or chrysophyte algal symbionts (Anderson and Be, 1976; Spero, 1987). The families Pulleniatinidae, Globorotaliidae, including N. dutertrei, P. obliquiloculata, G. menardii and G. tumida, have chrysophyte algal symbionts (Gastrich, 1988).*
*The relationship between the symbionts and the host is complex by nature. Nevertheless, this symbiotic relationship provides energy (Hallock, 1981b) and promotes calcification of the foraminifera (Duguay, 1983; Erez et al., 1983) by providing the inorganic carbon to the host (Jorgensen et al., 1985). Also, for T. sacculifer and O. universa photosynthesis increases with higher insolation (Jorgensen et al., 1985; Rink et al., 1998).*
*Dinoflagellate-bearing foraminifera (G. ruber, T. sacculifer and O. universa) tend to have a higher symbiont density and photosynthesis activity while P. obliquiloculata, G. menardii and N. dutertrei have lowered symbiont density and P. obliquiloculata, N. dutertrei lower photosynthetic activity (Takagi et al., 2019). P. obliquiloculata showed the minimum symbiont density and photosynthetic activity (Takagi et al., 2019)."*

*Lines 526-530: "Also, T. sacculifer has the potential to support more photosynthesis due to its higher symbiont density. Higher photosynthetic activity is observed compared to other species potentially supporting higher symbiont/host interactions. Those results could be in line with a greater sensitivity of T. sacculifer photosynthetic activity with changes in insolation/water depth. It can also be noted that this species presents the most important variations in symbiont density versus its test size.*
*Microenvironment pH results for N. dutertrei, G. menardii, G. tumida, are similar to O. universa and suggest a threshold for respiration driven δ11B signature. This threshold can be driven by a change of photosynthetic activity due to lower light intensity at deeper depth and/or a change in the symbiont assemblage with non-dinoflagellate symbionts at deeper depth. We can explain this threshold because deep dweller species do not experience important changes of insolation at those depths so their microenvironments should be respiration driven and relatively stable. We can also note that P. obliquiloculata which has the lowest symbiont density and photosynthetic activity has the lowered microenvironment pH compared to other deeper dweller species supporting this respiration driven microenvironment."*

**RC1:** 4) Light intensity in the ocean is not a function of water depth alone; turbidity matters quite a lot. Is it reasonable to compare the light intensity in different oceanographic regions as a function of water depth alone?

**Response:** *This is true, but we think it is still reasonable for the calculations because we used a light attenuation coefficient from Wang et al., (2008) which should take most of the parameters into account even if no variability is constrained.*

**RC1:** 4) L492. The negative relationship between _11B and water depth in Figure S2 is driven only by the low _11B measured in western equatorial Pacific T. sacculifer. Have you propagated your uncertainty in depth habitat to _11Bborate in your calculation of _11B as plotted in Figure S2? Looking at the western equatorial Pacific (WP7-01), the 80_20m depth range from CD2 corresponds to a _1‰ _11Bborate range (Fig. 4). (If you include the CD3 estimate of 125_15 m, the total possible _11Bborate increases to _1.5‰. Add on the _11Bforam measurement uncertainty of _0.22‰ and I cannot see how you get a total _11B uncertainty of <1‰.

**Response:** *The uncertainties on the $\delta 11B$borate are calculated using Michael Henehan's code which takes into account the uncertainties on pH, temperature, salinity and $\delta 11B$seawater. Uncertainties of pH, temperature and salinity were calculated integrating the parameters between uncertainties of the calcification depth. This is the same approach that was used in Raitzsch et al. (2018). Then the uncertainty on $\Delta 11B$ was measured by the square root of the sum of the square uncertainties on $\delta 11B$borate and $\delta 11B$carbonate.*

**RC1:** 5) I do not think linear regression is the appropriate test statistical test for the significance of the _11B-water depth relationship, because depending on how you performed the regression, it may not account for the uncertainty on each datapoint. A t-test for mean difference between Arabian Sea, Indian Ocean, and western equatorial Pacific T. sacculifer _11B would be more robust. I would bet that such a test will indicate no significant _11B difference between the different regions given the small sample sizes. If true, this detailed line of discussion is unnecessary; instead, you can present this as an observation that requires future study to confirm or deny.

**Response:** *We have added two data points from sites 806A and 807A which makes the t-test more robust.*

*For $\delta 11B$.*
*Unpaired t-test between the WEP and the rest of the samples for T. sacculifer (w/o sacc) is significantly different (p=0.01).*
*For $\Delta 11B$.*
*Unpaired t-test between the WEP and the rest of the samples for T. sacculifer (w/o sacc) is not significantly different (p=0.067). If now we look at both G. ruber and T. sacculifer (w/o sacc) which are presented in the plot, difference between Arabian Sea+Indian Ocean and WEP is significantly different (p=0.0075). If we also look at T. sacculifer (w/o sacc and sacc) + G.ruber between Arabian Sea+Indian Ocean and WEP the difference is significantly different (p=0.0427).*

*Lines 520-522: "Especially, we observe an important decrease of $\delta^{11}B$ in the WEP for T. sacculifer (w/o sacc), significantly different from the other sites (p<0.05) and a calculated $\Delta^{11}B$ of G. ruber, T. sacculifer (w/o sacc and sacc) significantly lower in the WEP compared to the other sites (p<0.05)."*

**RC1:** Any precipitation rate implications need to be very carefully phrased. The higher "values" for G. ruber and T. sacculifer have large uncertainties, so they are probably not robustly higher than other species (unless a statistical test confirms this to be true). Moreover, it is unclear the extent to which the growth rates of different foraminifera species differ from one another. Nevertheless, your point on higher B/Ca sensitivity to borate/bicarbonate in the shallower species is very interesting.

**Response:** *We mitigated this paragraph.*

*Lines 556-567: "When combining all literature data, T. sacculifer and G. ruber have sensitivities of δ11Bcarbonate to δ11Bborate of 0.83 ± 0.48 and 0.46 ± 0.34 respectively in line with previous literature and paleo-CO2 reconstructions. Also, if we only take into account our data, the observation that the sensitivity of δ11Bcarbonate to δ11Bborate are not statistically different from unity for most of the species investigated we can speculate that for these taxa, changes in precipitation rate and contributions of boric acid are not likely to be important. If considering only the data from this study, G. ruber (1.12 ± 1.67) and T. sacculifer (1.38± 1.35) present higher sensitivities of δ11Bcarbonate to δ11Bborate. We can then again speculate that the observed high values for δ11Bcarbonate at high seawater pH can be due to higher precipitation rates. We note this could also be consistent with the higher sensitivity of B/Ca signatures in these two surface dwelling species to ambient [B(OH)4-]/[HCO3-] relative to deeper dwelling species. As indicated by Farmer et al., (2019), studies of calcite precipitation rates in foraminifera could help to test this hypothesis and improve our understanding of the fundamental basis of boron-based proxies."*

**RC1:** L525-527. What do you mean by "higher values "here? Slopes of δ11Bforam to δ11Bborate regressions, or Δ11B, or something else?

**Response:** *Lines 561-562:" G. ruber (1.12 ± 1.67) and T. sacculifer (1.38± 1.35) present higher sensitivities of δ11Bcarbonate to δ11Bborate."*

**RC1:** L541-542. Given the speculative nature of the depth/light effect on symbiont photosynthesis, foraminifer microenvironment pH and thus foraminifera _11B, change to "which may be explained by the deeper depth habitat for these taxa in the WEP, where lower light levels might reduce symbiont photosynthetic activity". Or remove.

**Response:** *Changed.*

**RC1:** Fig.2/Fig.4

**Response:** *Front made bigger*

**RC1:** Figure 7. I really cannot follow this figure at all. Should there be a pH axis?

**Response:** *For more clarity, we combined in Figure 8 (or below) the Δmicroenvironment pH and this depth profile. This depth profile only aims to explain the low $\delta^{11}B$ in the WEP.*
*This is based on the calculation of insolation needed to explain the decrease in ΔpH0, ΔpH1 and ΔpH2 observed from the $\delta^{11}B$. This is an independent way to calculate the depth habitat explaining the $\delta^{11}B$.*
*The insolation is calculated based on Jorgensen et al., (1985). I found the insolation needed to decrease of ΔpH0, ΔpH1 and ΔpH2 and converted in % insolation in the water column. What is interesting is that the decrease of insolation around 125m can explain the low $\delta^{11}B$ values.*

[revised manuscript text omitted]

In planktonic foraminifera, algal symbiosis is the more common symbiotic relationship. For most of planktonic foraminifera, the host presents only one species of symbionts (Gast and Caron, 2001). The family

Globigerinidae, including *G. ruber*, *T. sacculifer* and *O. universa*, commonly have dinoflagellates or chrysophyte algal symbionts (Anderson and Be, 1976; Spero, 1987). The families Pulleniatinidae, Globorotaliidae, including

*N. dutertrei*, *P. obliquiloculata, G. menardii* and *G. tumida*, have chrysophyte algal symbionts (Gastrich, 1988).

The relationship between the symbionts and the host is complex by nature. Nevertheless, this symbiotic relationship provides energy (Hallock, 1981b) and promotes calcification of the foraminifera (Duguay, 1983;

Erez et al., 1983) by providing the inorganic carbon to the host (Jorgensen et al., 1985). Also, for *T. sacculifer*

and *O. universa* photosynthesis increases with higher insolation (Jorgensen et al., 1985; Rink et al., 1998).

Dinoflagellate-bearing foraminifera (*G. ruber*, *T. sacculifer* and *O. universa*) tend to have a higher symbiont density and photosynthesis activity while *P. obliquiloculata*, *G. menardii* and *N. dutertrei* have lowered symbiont density and *P. obliquiloculata*, *N. dutertrei* lower photosynthetic activity (Takagi et al., 2019).

*P. obliquiloculata* showed the minimum symbiont density and photosynthetic activity (Takagi et al., 2019).

It is now accepted that the foraminifera $\delta^{11}$B signature comes from the microenvironment pH

(Jorgensen et al., 1985; Rink et al., 1998; Köhler-Rink and Kühl, 2000, Hönisch et al., 2003; Zeebe et el., 2003).

Foraminifera with high photosynthetic activity and symbiont density like *G. ruber* and *T. sacculifer* present a pH

of microenvironment higher than ambient seawater, $\delta^{11}$B higher than 1:1 line (Foster et al., 2008, Henehan et al.,

2013, Raitzsch et al., 2018). The opposite can also be true, from our study, species with lower photosynthetic activity and lower symbiont density present microenvironments lower than ambient seawater, $\delta^{11}$B lower than

1:1 line (Martinez-Boti et al., 2015b; Henehan et al., 2016), this is the case in our data for *N. dutertrei*, *G.*

*menardii* and *P. obliquiloculata* and likely *G. tumida*. Nevertheless, the low $\delta^{11}$B of *O. universa* and *T.*

*sacculifer (w/o sacc)* from the WEP are difficult to reconcile with a high photosynthetic activity compared to *T.*

*sacculifer* et *G. ruber*.

The photosynthetic activity is also function of the light level they received which is, in the natural system, dependent of their depth in the water column, for the purpose of this study we will not consider turbidity which also influences the light penetration in the water column. In this case, the photosynthetically active foraminifera living close to the surface should see their microenvironment pH (thus $\delta^{11}$B) more sensitive to water depth changes. A deeper depth habitat will change the light intensity they received and as a consequence may lower their photosynthetic activity reducing their microenvironment pH. This thought is supported by the significant trend observed between our $\Delta^{11}$B and the calcification depth for *G. ruber* and *T. sacculifer* of our sites (Fig. S2). This trend basically supports the fact that the microenvironment pH decrease with calcification depth.

We observe a decrease of $\delta^{11}$B in the WEP for *T. sacculifer (w/o sacc)*, significantly different from the other sites (p<0.05). The $\Delta^{11}$B of *G. ruber*, *T. sacculifer* (w/o sacc and sacc) is also significantly lower in the WEP

compared to the other sites (p<0.05). To test if the $\delta^{11}$B signature was inferred to a light driven, we have been able to independently calculate the depth of the foraminifera based on various light insolation culture experiments (Jorgensen et al., 1985) and the $\Delta$microenvironment pH derived from our data (Fig. 8A and B). This exercise verified that this low $\delta^{11}$B can be explained by the reduced light environment due to a deeper depth habitat in the WEP (Fig. 8B). Also, *T. sacculifer* has the potential to support more photosynthesis due to its higher symbiont density. Higher photosynthetic activity is observed compared to other species potentially supporting higher symbiont/host interactions. Those results could be in line with a greater sensitivity of *T.*

*sacculifer* photosynthetic activity with changes in insolation/water depth. It can also be noted that this species presents the largest variations in symbiont density versus its test size. When applied to the other species *O.*

[revised manuscript text omitted]

There are a few main inferences we can make. When compiled with data from the literature, sensitivities of $\delta^{11}B_{carbonate}$ to $\delta^{11}B_{borate}$ for *G. ruber* and *T. sacculifer* are similar to previous studies (Martinez-

Boti et al., 2015b; Raitzsch et al., 2018) which is also supporting of previous paleo-reconstructions. Our data also support the observations of Henehan et al., (2016) for *O. universa*.

In order to derive accurate reconstructions of past ambient pH and $pCO_2$, accurate species-specific calibrations need to be used that are constrained by core-tops or samples from similar types of settings (Fig. 9,

10, S6). Lighter $\delta^{11}B$ signatures in *T. sacculifer* (w/o sacc) are observed in the WEP, which may be explained by the deeper depth habitat for these taxa, where lower light levels might reduce symbiont photosynthetic activity.

Also, correction will be needed for *T. sacculifer* (w/o sacc) in the WEP. When applying the calibrations n°2 and

4 to *T. sacculifer* and *G. ruber* (compilation of all data, Table 3) our data show more variability, especially for *G.*

*ruber* which lead to the larger mismatch compared to *in-situ* parameters. Henehan et al., (2013) reported a lighter

$\delta^{11}B$ with smaller test size, our sample add a weight/shell of 11 ± 4 µg (n=4, SD) which, despite a narrow range, could still explain this variability. The higher 
[revised manuscript text omitted]

[Figure]

[Figure]

[Figure]

[Figure]

| Temperature | Density anomaly | pH (pre-ind) | $\delta^{11}B_{borate}$ (‰) |
|---|---|---|---|

**Atlantic Ocean**

CD107-a
52.9°N, -16.9°E
m

January-March
October-December
July-September
April-June
Annual average

**Indian Ocean**

FC-01a
-11.2°N, 58.8°E
m

FC-02a
-29.1°N, 47.5°E
m

**Arabian Sea**

FC-12b
23.3°N, 66.7°E
m

FC-13a
20.0°N, 65.6°E
m

**Pacific Ocean**

WP07-01
-3.9°N,156.0°E
m

A14
8.0°N,113.4°E
m

[Figure]

[Figure]

- ● $\delta^{11}B_{G.\ ruber}$ (core-top, this study)
- ◇ $\delta^{11}B_{G.\ ruber}$ (core-top, Foster et al., 2008)
- ◐ $\delta^{11}B_{G.\ ruber}$ (core-top, 250-300μm, Henehan et al., 2013)
- ○ $\delta^{11}B_{G.\ ruber}$ (core-top, Henehan et al., 2013)
- □ $\delta^{11}B_{G.\ ruber}$ (sediment trap, Henehan et al., 2013)
- ◆ $\delta^{11}B_{G.\ ruber}$ (tow, Henehan et al., 2013)
- ▽ $\delta^{11}B_{G.\ ruber}$ (grab sample, Henehan et al., 2013)
- △ $\delta^{11}B_{G.\ ruber}$ (Raizsch et al., 2018)
- ── *G. ruber* calibration line (All data, this study)
- ── *G. ruber* calibration line (Core-top, this study)
- ── *G. ruber* calibration line (Culture, Henehan et al., 2013)

- ● $\delta^{11}B_{T.sacculifer\ (w/o\ sacc)}$ (core-top, this study)
- △ $\delta^{11}B_{T.\ sacculifer\ (w/o\ sacc)}$ (core-top, Raitzsch et al., 2018)
- ★ $\delta^{11}B_{T.sacculifer\ (sacc)}$ (core-top, this study)
- ◇ $\delta^{11}B_{T.\ sacculifer\ (sacc)}$ (core-top, Foster et al., 2008)
- ── *T. sacculifer (w/o sacc and sacc)* calibration line (All data, this study)
- ── *T. sacculifer (w/o sacc and sacc)* calibration line (Core-top, this study)
- ── *T. sacculifer (s)* calibration line (Martinez-Boti et al., 2015)

[Figure]

- ● $\delta^{11}B_{O.\ universa}$ (core-top, this study)
- ○ $\delta^{11}B_{O.\ universa}$ (core-top, Henehan et al., 2016)
- □ $\delta^{11}B_{O.\ universa}$ (sediment trap, Henehan et al., 2016)
- ◇ $\delta^{11}B_{O.\ universa}$ (tow, Henehan et al., 2016)
- △ $\delta^{11}B_{O.\ universa}$ (core-top, Raitzsch et al., 2018)
- ── *O. universa* calibration line (core-top, this study)
- ── *O. universa* calibration line (this study, Henehan et al., 2016, Raitzsch et al., 2018)
- ── *O. universa* calibration line (wild, Henehan et al., 2016)

[Figure]

○ δ¹¹B$_{P.obliquiloculata}$ (Core-top, this study)

▽ δ¹¹B$_{P.obliquiloculata}$ (Henehan et al., 2016)

── *P. obliquiloculata* calibration line (this study, Henehan et al., 2016)

── *O. universa* calibration curve (Henehan et al., 2016)

[Figure]

● δ¹¹B$_{G. menardii}$ (this study)

── *O. universa* calibration curve (Henehan et al., 2016)

[Figure]

● δ¹¹B$_{N. dutertrei}$ (Core-top, this study)

◇ δ¹¹B$_{N. dutertrei}$ (Core-top, Foster et al., 2008)

── *O. universa* calibration line (This study)

── *O. universa* calibration line (This study, Foster et al., 2008)

── *O. universa* calibration line (Henehan et al., 2016)

[Figure]

● δ¹¹B$_{G. tumida}$ (this study)

── *O. universa* calibration curve (Henehan et al., 2016)

[Figure]

● δ¹¹B$_{O.universa}$

○ δ¹¹B$_{P.obliquiloculata}$

● δ¹¹B$_{N. dutertrei}$

● δ¹¹B$_{G. menardii}$

● δ¹¹B$_{G. tumida}$

△ δ¹¹B$_{deep-dweller}$ from literature

── Deep-dweller calibration line

── *O. universa* calibration line (Henehan et al., 2016)

[Figure]

[Figure]

Photosynthesis/respiration-calcification

**A**

Δ microenvironment pH

*T. sacculifer*
μ = +0.047
n = 5

*G. ruber*
μ = +0.027
n = 5

*T. sacculifer (w/o sacc)* (WEP)
μ = -0.022
n = 4

Compensation point

*O. universa*
μ = -0.153
n = 5

*N. dutertrei*
μ = -0.159
n = 5

*G. menardii*
μ = -0.163
n = 4

*G. tumida*
μ = -0.168
n = 3

*P. obliquiloculata*
μ = -0.240
n = 5

**B**

Light penetration WEP (%)

Depth (m)

ΔpH$_0$= -0.02
ΔpH$_1$ = -0.04
ΔpH$_2$ = -0.06

[revised manuscript text omitted]

**Figure S5:** Figure evaluating the circularity of our reconstructions. It is showing in the y-axis the difference between reconstruction utilizing calibrations derive from the entire dataset and compare to *in-situ* values and in the x-axis the difference between the reconstruction utilizing the species-specific calibrations derived excluding the site of interest (no circularity) compared to *in-situ* values. Results show that difference is not significant between the two reconstruction methods (e.g. following the 1:1 line), validating the method and the calibrations.

[revised manuscript text omitted]
 ΔpH$_1$ would lead to a decrease of 15 μEistn.m$^{-2}$.s$^{-1}$ and a decrease of ΔpH$_2$ would lead to a decrease of 24 μEistn.m$^{-2}$.s$^{-1}$ (Jorgensen et al., 1985). These results correspond in our case of a light penetration of 12% to reach Ec, 5% for a decrease of ΔpH$_1$ and 1% for a decrease of ΔpH$_2$. This means that in the WEP if *T. sacculifer* calcifies below 75m where E$_c$ is reached the δ$^{11}$B$_{carbonate}$ is below the theoretical 1:1 line. *T. sacculifer* (w/o sacc) in the WEP is decreasing its pH of ~ΔpH$_1$ which would imply a calcification depth of 110m consistent with the reconstruction of Rickaby et al., (2005).

$$\Delta \text{microenvironment pH} = -\log\left(\frac{(\delta 11 B \text{seawater} - \delta 11 B \text{carbonate}) \times Kb*}{\varepsilon - \delta 11 B \text{seawater} + \delta 11 B \text{carbonate}}\right) - \text{pHseawater}$$

**References**

[revised manuscript text omitted]

- G. ruber
- T. sacculifer (w/o sacc)
- T. sacculifer (sacc)
- O. universa
- P. obliquiloculata
- N. dutertrei
- G. menardii
- G. tumida

[Figure]

[Figure]

[Figure]

**Table S1**

| Elemental ratios | Li/Ca | B/Ca | Mg/Ca | Al/Ca | Sr/Ca | Cd/Ca | Ba/Ca | U/Ca | Mn/Ca | Fe/Ca |
| --- | --- | --- | --- | --- | --- | --- | --- | --- | --- | --- |
| | μmol/mol | μmol/mol | mmol/mol | mmol/mol | mmol/mol | μmol/mol | μmol/mol | nmol/mol | μmol/mol | mmol/mol |
| Standard solution 0 | 0.8 | 9 | 0.10 | 0.131 | 0.00 | 0.03 | 0.6 | 31 | 1 | 0.01 |
| Standard solution 1 | 2.3 | 38 | 0.31 | 0.112 | 0.49 | 0.05 | 1.9 | 38 | 12 | 0.02 |
| Standard solution 3 | 6.8 | 108 | 1.31 | 0.177 | 1.06 | 0.13 | 3.0 | 53 | 39 | 0.04 |
| Standard solution 5 | 14.6 | 216 | 3.17 | 0.223 | 1.57 | 0.23 | 5.1 | 62 | 129 | 0.08 |
| Standard solution 6 | 19.0 | 278 | 5.23 | 0.352 | 1.97 | 0.28 | 5.5 | 74 | 196 | 0.11 |
| Standard solution 8 | 25.0 | 281 | 6.07 | 0.602 | 2.99 | 0.50 | 20.1 | 390 | 501 | 0.50 |
| Standard solution 9 | | 408 | | | 4.89 | | | | | |
| Standard solution 10 | | 519 | | | 8.01 | | | | | |
| Standard solution 11 | | 607 | | | 9.93 | | | | | |

**Table S2**

[revised manuscript text omitted]

---

## Author Comment (AC2) · 23 Oct 2019

We wish to thank Michael Henehan for his helpful comments on the manuscript and previous help with the code. We believe that we addressed all of the major comments as indicated in the discussion below and the updated manuscript (see supplement).

Comment SC1 1: The authors on several occasions highlight the difference between their G. ruber data and the slope obtained by our culture experiments (Lines 385-387; 422-425; lines 525-526; 537-538; 563). In lines 422-425 the picture as presented is

[Figure]

particularly confusing, since the authors first suggest "there is a difference in calibrations", then say "this is particularly notable for G. ruber", but then say "the sensitivity of the species analysed are not statistically different". In truth, only the final sentence is true (with the exception of the clause "and are close to unity". A slope of 1.12 _ 1.67 is within uncertainty of our culture slope (0.6), and could technically allow a slope as low as -0.55: i.e. there is no significant difference between the slope they suggest and the slope that we observe in culture. Framing this as a difference seems even more odd given the authors do not draw any distinction between their T. sacculifer slope and that of previous calibrations, because their bounds of uncertainty do not allow it- so it seems logically inconsistent to draw distinctions for ruber where the statistical difference is equally unfounded. I would suggest the authors go through the manuscript and revise their phrasing to reflect this lack of statistically significant difference. Including "the sensitivity of d11B to pH is not statistically different from unity for G. ruber" as a main conclusion in line 563, for example, implies this study is in disagreement with cultures (which it isn't), and that we should consider a slope of 1 to be potentially suitable for this species. In reality, were we to calculate pCO2 with the slope and intercept that the authors suggest (m=1.12, c=-1.23), the fit of the downcore record of G. ruber from Chalk et al. 2017 with ice-core pCO2 would be considerably worse (see attached Fig. 1). The magnitude of pCO2 change between glacials and interglacials (i.e. the parameter that is driven by the slope of the calibration) is underestimated, with pCO2 too high in glacials by _50 ppm. The improved fit of the down-core record with pCO2 from ice-cores when our ruber calibration is used, however (see Chalk et al. 2017), offers support for the shallowerthan-unity slope we observed in this species. Incidentally, also, an R-squared of 0.98 for the ruber core-top data presented in this study seems anomalously high relative to the scatter/uncertainty bounds in the dataset- can the authors be clear how this Rsquared is computed? Is it the average R-squared of Monte Carlo regressions plotted through datapoints randomly subsampled from within the x- and y-uncertainties? Or is this simply a least-squares linear regression through the central tendencies of the datapoints? The former might be more representative, but as long as the authors are clear about what they are describing that is the main thing.

Response 1: We have mitigated our discussion, especially because our dataset is too limited and our trend driven by two datapoints (WP07-1 and FC01-a). The results presented for the sensitivity > 1 of $\delta11B_{carbonate}$ to $\delta11B_{borate}$ have been clearly written as speculative in the text; first of all because when doing the bootstrap on all compiled data the regression is similar to Raitzsch et al., (2018), sensitivity of $0.46 \pm 0.34$ (updated table 3) compared to $0.45 \pm 0.16$ for Raitzsch et al., (2018) and as highlighted the uncertainties based on 5 points is important and finally pCO2 reconstructions would not be consistent with the Vostok pCO2 record.

The $R^2$ were calculated with R doing a linear regression not taking into account all simulated values of the Monte Carlo Simulation. Because we did not find a way to extract those data, we decided to not present the $R^2$ and p-value.

Comment SC1 2: The authors report our generic culture intercept for ruber in their Table 3 (9.52), but erroneously list the size fraction as _250 _m. I would like to point them to Fig. 6 of this 2013 paper, where we give the average size fraction of our cultures to be _380 _m. A suggested size fraction correction on the intercept is given, such that for 300-355_m it would be 8.87. On this same note, the authors also combine a wide size fraction (250-400_m) for G. ruber, which given size-related offsets from the culture calibration (as shown in Fig. 6 of Henehan et al. 2013) has the potential over such a large range to skew the data, due to size-related changes in d11B. Can the authors give some estimate as to the distribution of test sizes within their broad sample range, so as to make them more easily comparable to published data?

Response 2: I changed the size fraction in Table 3, I missed this information in your paper. Most of the samples have been picked in the 250-300 size fraction (average weight/ shell of $11 \pm 4$ $\mu$g (n=4, SD) only when measurements were realized), we chose a restrained size fraction to avoid this size-related variability (at least in our calibration), we also chose this lower size fraction because some of the sites did not present shells in the higher size fractions and we wanted to stay consistent. Now, when compiling all the data this variability is not constrained. From your paper, our weight/shell variability could lead to an offset up to ∼1% that we acknowledged can explain most of the variability for our G. ruber data. I have added line 582 "Henehan et al., (2013) reported a lighter $\delta$11B with smaller test size, our sample add a weight/shell of 11 ± 4 $\mu$g (n=4, SD) which could also explain this variability."

Comment SC1 3: The authors screened for clay contamination using Ti/Ca ratios, as Al/Ca values were difficult to measure with their introduction system. However, they do not provide these data. Clay may carry isotopically-light sorbed d11B with it, and introduce bias towards lighter values. To allow maximum confidence in the data, and see which datapoints if any might have some influence of clay, can the authors please provide the Ti/Ca ratios in Table 2?

Response 3: We only had a contamination at one site which is not presented in the paper (site E035), with high Mn/Ca of 79$\mu$mol/mol and high Fe/Ca of 3.0 mmol/mol. We have added the Mn/Ca and Fe/Ca in Table 2. We didn't add the Ti/Ca ratios as we monitored it with the raw ratios and do not have the absolute values. However, a minor correlation was found between Ti/Ca and B/Ca (R2=0.0887). Some of our samples have elevated Fe/Ca concentration but no high Mn/Ca, we don't suspect contamination from those samples since this high Fe can potentially come from MnCO3 overgrowth and this over growth will have negligible quantity of Mg and B unlike the Fe-Mn oxide and hydroxides.

Comment SC1 4: In Section 4.2.3 (note the paper skips 4.2.4 and goes straight on to 4.2.5?), the authors pool 'deeper-dwelling' foraminiferal species together, but this seems a bit unfounded since these foraminifera don't even have the same symbiont types (crysophyte vs. dinoflagellate), and have quite different ecologies. I'm not convinced there's enough of an a priori reason to even do this in the first place. However, I see that the authors do already concede this may be unfounded.

Response 4: We developed the discussion about the symbionts type/photosynthesis, lines 485-543. Even if the data are limited, we don't want to reject a chrysophytes/insolation limiting threshold resulting in a respiration-driven environment where this calibration can make sense. I have tried to reply to this comment for reviewer 2:

"I have tried to improve the discussion, focusing on the symbionts/photosynthesis, because the story is of course more complex. From what I see in the literature is that T. sacculifer, G. ruber, O. universa have mostly dinoflagellates symbionts (can have chrysophyte as well) where G. tumida, G. menardii, P. obliquiloculata and N. dutertrei will have chrysophyte algal symbionts. The photosynthesis is dependent of the nature of the host/symbionts interactions, symbionts type (pigment associated for light absorption efficiency), symbionts density. The recent study from Tagaki et al., (2019) is really helpful as he constrained the photosynthesis activity, light absorption efficiency and the symbiont density of those species.

Fv/Fm (photosyntethic activity) T. sacculifer>G. menardii > O. universa> G. ruber (white) > N. dutertrei > P. obliquiloculata $\sigma$psi (light absorption efficiency) N. dutertrei > P. obliquiloculata > G. menardii > G. ruber > T. sacculifer > O. universa Chla/biomass T. sacculifer>O.universa>G. ruber> N. dutertrei>G. menardii>P. obliquiloculata

What I assume is that T. sacculifer, O. universa and G. ruber photosynthesis are likely to be more affected by changes in insolation than other species due to their symbiont density, high photosynthetic capacity and their light absorption efficiency. Which is still in line with the argumentation we are giving. Also, the fact that the deeper dwellers have this low boron isotopic signature is likely due to a lower symbiont density, lower photosynthetic activity and a reduced insolated environment. P. obliquiloculata has the lowest density and photosynthetic activity, which would translate in a respiration driven environment the fact that most of the species are following this trend would go in the sense of a respiration driven environment. Also the fact that O. universa is following this trend would, I think, whether be due light limitation and/or a different symbiont due to its deeper depth."
Comment SC1 5: In section 3.9, the authors make no mention of how they calculated pK*B for each foraminifera. I take it they did indeed account for changes in pK*B with temperature, salinity and pressure? It may sound blindingly obvious, but I'm constantly amazed at how many people make this error. On a similar theme Fig S3 the pH lines are no doubt helpful, but I'm not sure how the authors managed to calculate them, given the pK*B is different for each foram. Is this calculated using the mean pK*B of each of the forams plotted? This figure makes me worry that the authors just chose a single value of pK*B for all forams in all calculations of the paper, which would be wrong.

Response 5: The pKB* were calculated taking into account temperature, salinity and pressure in all calculations of the paper except in Fig S3 to draw the Δmicroenvironment pH line. We agree that individually, the parameters can significantly influence the calculations (figure below), especially temperature, then salinity and a minor effect for pressure. Maximum divergence was observed for site CD107-a due to colder temperature however for the other sites due to our uncertainties, the results were still consistent with our discussion. However, I have directly calculated the pH difference from the microenvironment for each of the species (every calculations taking into account, P, T and S). Results are shown in Figure 8 (or below).

Comment SC1 6: I think the decision to group 'shallow-dwelling' foraminifera (note it is not clearly defined what species this includes in any caption) in Fig. S4 (and in the text where this is referenced) is I think unfounded. It produces a correlation between B/Ca and Borate/DIC, sure, but it's entirely driven by the interspecies difference between ruber and sacculifer, and we know from Kat Allen's work for example that these species have fundamentally different B/Ca-Borate/DIC relationships.. hey shouldn't be lumped together in one group. As it is, it makes this look like a carbonate system relationship when on an intra-species level there is no significant correlation with the carbonate system (as we also observed elsewhere).

Response 6: We group G. ruber and T. sacculifer data in the "shallow-dwelling"

foraminifera. It is true that this calibration can be driven by interspecific differences. Then, I have added calibrations for T. sacculifer and G. ruber.

Line 429-432: "B/Ca ratios are presented in Table 2. Values are species specific consistent with previous work (e.g., compiled in Henehan et al., 2016) with ratios higher for G. ruber > T. sacculifer > T. sacculifer (w/o sacc) > P. obliquicloculata > O. universa > > G. menardii > N. dutertrei > G. tumida > G. inflata > N. pachyderma > G. bulloides (Fig. 7). This study supports interspecific B/Ca ratios (Allen and Hönisch, 2012; Henehan et al., 2016)."

Line 565-568: "Those interspecific differences still remain to be explained, however, part of this variability is likely due to changes of the carbonate chemistry of the microenvironment resulting in changing competition between borate and bicarbonate ion, but we can't exclude specific biological processes, and for the mixed-dweller (e.g. non respiration-driven microenvironment) day/night calcification ratios."

Please also note the supplement to this comment:
https://www.biogeosciences-discuss.net/bg-2019-266/bg-2019-266-AC2-supplement.pdf

———————————

[Figure]

[Figure]

[Figure]

**Fig. 1.**

[Figure]

**Fig. 2.**

**Supplement:**

**Response to SC1 - Michael Henehan**

We wish to thank Michael Henehan for his helpful comments on the manuscript and previous help with the code. We believe that we addressed all of the major comments as indicated in the discussion below and the updated document.

The updated manuscript and figures can be found at the end after this response.

**Comment SC1 1:** The authors on several occasions highlight the difference between their G. ruber data and the slope obtained by our culture experiments (Lines 385-387; 422-425; lines 525-526; 537-538; 563). In lines 422-425 the picture as presented is particularly confusing, since the authors first suggest "there is a difference in calibrations", then say "this is particularly notable for G. ruber", but then say "the sensitivity of the species analysed are not statistically different". In truth, only the final sentence is true (with the exception of the clause "and are close to unity". A slope of 1.12 _ 1.67 is within uncertainty of our culture slope (0.6), and could technically allow a slope as low as -0.55: i.e. there is no significant difference between the slope they suggest and the slope that we observe in culture.
Framing this as a difference seems even more odd given the authors do not draw any distinction between their T. sacculifer slope and that of previous calibrations, because their bounds of uncertainty do not allow it- so it seems logically inconsistent to draw distinctions for ruber where the statistical difference is equally unfounded. I would suggest the authors go through the manuscript and revise their phrasing to reflect this lack of statistically significant difference. Including "the sensitivity of d11B to pH is not statistically different from unity for G. ruber" as a main conclusion in line 563, for example, implies this study is in disagreement with cultures (which it isn't), and that we should consider a slope of 1 to be potentially suitable for this species. In reality, were we to calculate pCO2 with the slope and intercept that the authors suggest (m=1.12, c=-1.23), the fit of the downcore record of G. ruber from Chalk et al. 2017 with ice-core pCO2 would be considerably worse (see attached Fig. 1). The magnitude of pCO2 change between glacials and interglacials (i.e. the parameter that is driven by the slope of the calibration) is underestimated, with pCO2 too high in glacials by _50 ppm. The improved fit of the down-core record with pCO2 from ice-cores when our ruber calibration is used, however (see Chalk et al. 2017), offers support for the shallowerthan-unity slope we observed in this species. Incidentally, also, an R-squared of 0.98 for the ruber core-top data presented in this study seems anomalously high relative to the scatter/uncertainty bounds in the dataset- can the authors be clear how this Rsquared is computed? Is it the average R-squared of Monte Carlo regressions plotted through datapoints randomly subsampled from within the x- and y-uncertainties? Or is this simply a least-squares linear regression through the central tendencies of the datapoints? The former might be more representative, but as long as the authors are clear about what they are describing that is the main thing.

**Response 1:** *We have mitigated our discussion, especially because our dataset is too limited and our trend driven by two datapoints (WP07-1 and FC01-a). The results presented for the sensitivity > 1 of δ11Bcarbonate to δ11Bborate have been clearly written as speculative in the text; first of all because when doing the bootstrap on all compiled data the regression is similar to Raitzsch et al., (2018), sensitivity of 0.46 ± 0.34 (updated table 3) compared to 0.45 ± 0.16 for Raitzsch et al., (2018) and as highlighted the uncertainties based on 5 points is important and finally pCO2 reconstructions would not be consistent with the Vostok pCO2 record.*

*The R² were calculated with R doing a linear regression not taking into account all simulated values of the Monte Carlo Simulation. Because we did not find a way to extract those data, we decided to not present the R² and p-value.*

**Comment SC1 2:** The authors report our generic culture intercept for ruber in their Table 3 (9.52), but erroneously list the size fraction as _250 _m. I would like to point them to Fig. 6 of this 2013 paper, where we give the average size fraction of our cultures to be _380 _m. A suggested size fraction correction on the intercept is given, such that for 300-355_m it would be 8.87. On this same note, the authors also combine a wide size fraction (250-400_m) for G. ruber, which given size-related offsets from the culture calibration (as shown in Fig. 6 of Henehan et al. 2013) has the potential over such a large range to skew the data, due to size-related changes in d11B. Can the authors give some estimate as to the distribution of test sizes within their broad sample range, so as to make them more easily comparable to published data?

**Response 2:** *I changed the size fraction in Table 3, I missed this information in your paper.*
*Most of the samples have been picked in the 250-300 size fraction (average weight/ shell of 11 ± 4 µg (n=4, SD) only when measurements were realized), we chose a restrained size fraction to avoid this size-related variability (at least in our calibration), we also chose this lower size fraction because some of the sites did not present shells in the higher size fractions and we wanted to stay consistent. Now, when compiling all the data this variability is not constrained.*
*From your paper, our weight/ shell variability could lead to an offset up to ~1% that we acknowledged can explain most of the variability for our G. ruber data.*
*I have added line 582 "Henehan et al., (2013) reported a lighter $\delta^{11}B$ with smaller test size, our sample add a weight/shell of 11 ± 4 µg (n=4, SD) which could also explain this variability."*

**Comment SC1 3:** The authors screened for clay contamination using Ti/Ca ratios, as Al/Ca values were difficult to measure with their introduction system. However, they do not provide these data. Clay may carry isotopically-light sorbed d11B with it, and introduce bias towards lighter values. To allow maximum confidence in the data, and see which datapoints if any might have some influence of clay, can the authors please provide the Ti/Ca ratios in Table 2?

**Response 3:** *We only had a contamination at one site which is not presented in the paper (site E035), with high Mn/Ca of 79µmol/mol and high Fe/Ca of 3.0 mmol/mol. We have added the Mn/Ca and Fe/Ca in Table 2. We didn't add the Ti/Ca ratios as we monitored it with the raw ratios and do not have the absolute values. However, a minor correlation was found between Ti/Ca and B/Ca ($R^2=0.0887$). Some of our samples have elevated Fe/Ca concentration but no high Mn/Ca, we don't suspect contamination from those samples since this high Fe can potentially come from $MnCO_3$ overgrowth and this over growth will have negligible quantity of Mg and B unlike the Fe-Mn oxide and hydroxides.*

**Comment SC1 4:** In Section 4.2.3 (note the paper skips 4.2.4 and goes straight on to 4.2.5?), the authors pool 'deeper-dwelling' foraminiferal species together, but this seems a bit unfounded since these foraminifera don't even have the same symbiont types (crysophyte vs. dinoflagellate), and have quite different ecologies. I'm not convinced there's enough of an a priori reason to even do this in the first place. However, I see that the authors do already concede this may be unfounded.

**Response 4:** *We developed the discussion about the symbionts type/photosynthesis, lines 485-543.*
*Even if the data are limited, we don't want to reject a chrysophytes/insolation limiting threshold resulting in a respiration-driven environment where this calibration can make sense. I have tried to reply to this comment for reviewer 2:*

*"I have tried to improve the discussion, focusing on the symbionts/photosynthesis, because the story is of course more complex.*
*From what I see in the literature is that T. sacculifer, G. ruber, O. universa have mostly dinoflagellates symbionts (can have chrysophyte as well) where G. tumida, G. menardii, P. obliquiloculata and N. dutertrei will have chrysophyte algal symbionts. The photosynthesis is dependent of the nature of the host/symbionts interactions, symbionts type (pigment associated for light absorption efficiency), symbionts*

*density. The recent study from Tagaki et al., (2019) is really helpful as he constrained the photosynthesis activity, light absorption efficiency and the symbiont density of those species.*

*Fv/Fm (photosyntethic activity) T. sacculifer>G. menardii > O. universa> G. ruber (white) > N. dutertrei > P. obliquiloculata*

*σpsi (light absorption efficiency) N. dutertrei > P. obliquiloculata > G. menardii > G. ruber > T. sacculifer > O. universa*

*Chla/biomass       T. sacculifer>O.universa>G. ruber> N. dutertrei>G. menardii>P. obliquiloculata*

*What I assume is that T. sacculifer, O. universa and G. ruber photosynthesis are likely to be more affected by changes in insolation than other species due to their symbiont density, high photosynthetic capacity and their light absorption efficiency. Which is still in line with the argumentation we are giving. Also, the fact that the deeper dwellers have this low boron isotopic signature is likely due to a lower symbiont density, lower photosynthetic activity and a reduced insolated environment. P. obliquiloculata has the lowest density and photosynthetic activity, which would translate in a respiration driven environment the fact that most of the species are following this trend would go in the sense of a respiration driven environment.*
*Also the fact that O. universa is following this trend would, I think, whether be due light limitation and/or a different symbiont due to its deeper depth."*

**Comment SC1 5:** In section 3.9, the authors make no mention of how they calculated pK*B for each foraminifera. I take it they did indeed account for changes in pK*B with temperature, salinity and pressure? It may sound blindingly obvious, but I'm constantly amazed at how many people make this error. On a similar theme Fig S3 the pH lines are no doubt helpful, but I'm not sure how the authors managed to calculate them, given the pK*B is different for each foram. Is this calculated using the mean pK*B of each of the forams plotted? This figure makes me worry that the authors just chose a single value of pK*B for all forams in all calculations of the paper, which would be wrong.

**Response 5:** *The pKB* were calculated taking into account temperature, salinity and pressure in all calculations of the paper except in Fig S3 to draw the Δmicroenvironment pH line. We agree that individually, the parameters can significantly influence the calculations, especially temperature, then salinity and a minor effect for pressure. Maximum divergence was observed for site CD107-a due to colder temperature however for the other sites due to our uncertainties, the results were still consistent with our discussion.*

[Figure]

However, I have directly calculated the pH difference from the microenvironment for each of the species (every calculations taking into account, P, T and S). Results are shown in Figure 7 (or below).

[Figure]

**Comment SC1 6:** I think the decision to group 'shallow-dwelling' foraminifera (note it is not clearly defined what species this includes in any caption) in Fig. S4 (and in the text where this is referenced) is I think unfounded. It produces a correlation between B/Ca and Borate/DIC, sure, but it's entirely driven by the interspecies difference between ruber and sacculifer, and we know from Kat Allen's work for example that these species have fundamentally different B/Ca-Borate/DIC relationships.. hey shouldn't be lumped together in one group. As it is, it makes this look like a carbonate system relationship when on an intra-species level there is no significant correlation with the carbonate system (as we also observed elsewhere).

**Response 6:** *We group G. ruber and T. sacculifer data in the "shallow-dwelling" foraminifera. It is true that this calibration can be driven by interspecific differences.*
*Then, I have added calibrations for T. sacculifer and G. ruber.*

*Line 429-432: "B/Ca ratios are presented in Table 2. Values are species specific consistent with previous work (e.g., compiled in Henehan et al., 2016) with ratios higher for G. ruber > T. sacculifer > T. sacculifer (w/o sacc) > P. obliquicloculata > O. universa > > G. menardii > N. dutertrei > G. tumida > G. inflata > N. pachyderma > G. bulloides (Fig. 7). This study supports interspecific B/Ca ratios (Allen and Hönisch, 2012; Henehan et al., 2016)."*

*Line 565-568: "Those interspecific differences still remain to be explained, however, part of this variability is likely due to changes of the carbonate chemistry of the microenvironment resulting in changing competition between borate and bicarbonate ion, but we can't exclude specific biological processes, and for the mixed-dweller (e.g. non respiration-driven microenvironment) day/night calcification ratios."*

[revised manuscript text omitted]

In planktonic foraminifera, algal symbiosis is the more common symbiotic relationship. For most of
planktonic foraminifera, the host presents only one species of symbionts (Gast and Caron, 2001). The family
Globigerinidae, including *G. ruber*, *T. sacculifer* and *O. universa*, commonly have dinoflagellates or chrysophyte
algal symbionts (Anderson and Be, 1976; Spero, 1987). The families Pulleniatinidae, Globorotaliidae, including
*N. dutertrei*, *P. obliquiloculata, G. menardii* and *G. tumida*, have chrysophyte algal symbionts (Gastrich, 1988).
The relationship between the symbionts and the host is complex by nature. Nevertheless, this symbiotic
relationship provides energy (Hallock, 1981b) and promotes calcification of the foraminifera (Duguay, 1983;
Erez et al., 1983) by providing the inorganic carbon to the host (Jorgensen et al., 1985). Also, for *T. sacculifer*
and *O. universa* photosynthesis increases with higher insolation (Jorgensen et al., 1985; Rink et al., 1998).

Dinoflagellate-bearing foraminifera (*G. ruber*, *T. sacculifer* and *O. universa*) tend to have a higher symbiont density and photosynthesis activity while *P. obliquiloculata*, *G. menardii* and *N. dutertrei* have lowered symbiont density and *P. obliquiloculata*, *N. dutertrei* lower photosynthetic activity (Takagi et al., 2019).

*P. obliquiloculata* showed the minimum symbiont density and photosynthetic activity (Takagi et al., 2019).

It is now accepted that the foraminifera $\delta^{11}$B signature comes from the microenvironment pH

(Jorgensen et al., 1985; Rink et al., 1998; Köhler-Rink and Kühl, 2000, Hönisch et al., 2003; Zeebe et el., 2003).

Foraminifera with high photosynthetic activity and symbiont density like *G. ruber* and *T. sacculifer* present a pH

of microenvironment higher than ambient seawater, $\delta^{11}$B higher than 1:1 line (Foster et al., 2008, Henehan et al.,

2013, Raitzsch et al., 2018). The opposite can also be true, from our study, species with lower photosynthetic activity and lower symbiont density present microenvironments lower than ambient seawater, $\delta^{11}$B lower than

1:1 line (Martinez-Boti et al., 2015b; Henehan et al., 2016), this is the case in our data for *N. dutertrei*, *G.

menardii* and *P. obliquiloculata* and likely *G. tumida*. Nevertheless, the low $\delta^{11}$B of *O. universa* and *T.

sacculifer (w/o sacc)* from the WEP are difficult to reconcile with a high photosynthetic activity compared to *T.

sacculifer* et *G. ruber*.

The photosynthetic activity is also function of the light level they received which is, in the natural system, dependent of their depth in the water column, for the purpose of this study we will not consider turbidity which also influences the light penetration in the water column. In this case, the photosynthetically active foraminifera living close to the surface should see their microenvironment pH (thus $\delta^{11}$B) more sensitive to water depth changes. A deeper depth habitat will change the light intensity they received and as a consequence may lower their photosynthetic activity reducing their microenvironment pH. This thought is supported by the significant trend observed between our $\Delta^{11}$B and the calcification depth for *G. ruber* and *T. sacculifer* of our sites (Fig. S2). This trend basically supports the fact that the microenvironment pH decrease with calcification depth.

We observe a decrease of $\delta^{11}$B in the WEP for *T. sacculifer (w/o sacc)*, significantly different from the other sites (p<0.05). The $\Delta^{11}$B of *G. ruber*, *T. sacculifer* (w/o sacc and sacc) is also significantly lower in the WEP

compared to the other sites (p<0.05). To test if the $\delta^{11}$B signature was inferred to a light driven, we have been able to independently calculate the depth of the foraminifera based on various light insolation culture experiments (Jorgensen et al., 1985) and the $\Delta$microenvironment pH derived from our data (Fig. 8A and B). This exercise verified that this low $\delta^{11}$B can be explained by the reduced light environment due to a deeper depth habitat in the WEP (Fig. 8B). Also, *T. sacculifer* has the potential to support more photosynthesis due to its higher symbiont density. Higher photosynthetic activity is observed compared to other species potentially supporting higher symbiont/host interactions. Those results could be in line with a greater sensitivity of *T.

sacculifer* photosynthetic activity with changes in insolation/water depth. It can also be noted that this species presents the largest variations in symbiont density versus its test size. When applied to the other species *O.

[revised manuscript text omitted]

There are a few main inferences we can make. When compiled with data from the literature,
sensitivities of $\delta^{11}B_{carbonate}$ to $\delta^{11}B_{borate}$ for *G. ruber* and *T. sacculifer* are similar to previous studies (Martinez-
Boti et al., 2015b; Raitzsch et al., 2018) which is also supporting of previous paleo-reconstructions. Our data
also support the observations of Henehan et al., (2016) for *O. universa*.

In order to derive accurate reconstructions of past ambient pH and $pCO_2$, accurate species-specific
calibrations need to be used that are constrained by core-tops or samples from similar types of settings (Fig. 9,
10, S6). Lighter $\delta^{11}B$ signatures in *T. sacculifer* (w/o sacc) are observed in the WEP, which may be explained by
the deeper depth habitat for these taxa, where lower light levels might reduce symbiont photosynthetic activity.
Also, correction will be needed for *T. sacculifer* (w/o sacc) in the WEP. When applying the calibrations n°2 and
4 to *T. sacculifer* and *G. ruber* (compilation of all data, Table 3) our data show more variability, especially for *G.*
*ruber* which lead to the larger mismatch compared to *in-situ* parameters. Henehan et al., (2013) reported a lighter
$\delta^{11}B$ with smaller test size, our sample add a weight/shell of 11 ± 4 µg (n=4, SD) which, despite a narrow range,

[revised manuscript text omitted]

- ▬ (grey) *O. universa* calibration curve (Henehan et al., 2016)

- ⬤ (purple) δ$^{11}$B$_{G. menardii}$ (this study)
- ▬ (grey) *O. universa* calibration curve (Henehan et al., 2016)

[Figure]

[Figure]

- ⬤ (green) δ$^{11}$B$_{N. dutertrei}$ (Core-top, this study)
- ◇ δ$^{11}$B$_{N. dutertrei}$ (Core-top, Foster et al., 2008)
- ▬ (green) *O. universa* calibration line (This study)
- ▬ (light green) *O. universa* calibration line (This study, Foster et al., 2008)
- ▬ (grey) *O. universa* calibration line (Henehan et al., 2016)

- ⬤ (violet) δ$^{11}$B$_{G. tumida}$ (this study)
- ▬ (grey) *O. universa* calibration curve (Henehan et al., 2016)

[Figure]

- ⬤ (orange) δ$^{11}$B$_{O.universa}$
- ⬤ (yellow) δ$^{11}$B$_{P.obliquiloculata}$
- ⬤ (green) δ$^{11}$B$_{N. dutertrei}$
- ⬤ (purple) δ$^{11}$B$_{G. menardii}$
- ⬤ (violet) δ$^{11}$B$_{G. tumida}$
- △ δ$^{11}$B$_{deep-dweller}$ from literature
- ▬ (black) Deep-dweller calibration line
- ▬ (grey) *O. universa* calibration line (Henehan et al., 2016)

[Figure]

[Figure]

Photosynthesis/respiration-calcification

[Figure]

[Figure]

δ^11B_borate   pH   pCO_2

1:1 (all three panels)

Legend:
- *G. ruber* (white) [1]

[revised manuscript text omitted]

**Figure S5:** Figure evaluating the circularity of our reconstructions. It is showing in the y-axis the difference between reconstruction utilizing calibrations derive from the entire dataset and compare to *in-situ* values and in the x-axis the difference between the reconstruction utilizing the species-specific calibrations derived excluding the site of interest (no circularity) compared to *in-situ* values. Results show that difference is not significant between the two reconstruction methods (e.g. following the 1:1 line), validating the method and the calibrations.

[revised manuscript text omitted]

---

## Author Comment (AC3) · 23 Oct 2019

We wish to thank this reviewer for his thorough review of our manuscript and his helpful comments. We believe that we addressed all of the major comments as indicated in the discussion below and the updated manuscript (see supplement).

RC2: Depth habitats of the foraminifers are used to link the habitat to oceanographic conditions at the core locations. The supplement has a fairly thorough explanation of how the various depths were determined (using d18Oc, MgCa-derived T, and published

[Figure]

depth habitats; detailed in Table 6), but I couldn't determine which depth the authors chose to use for each species (or at least is wasn't consistently clear in the text) without seeking the information in Table 7. In the intro of the MS, the authors point the reader to table 3 for the depth habitats but table 3 doesn't include this information. Also, I don't understand why sometimes the authors used published references for depth habitats and in other instances is d18O or Mg/Ca derived temperatures. For example: For core FC-01a, the Sime reference is cited for the depths used for G. ruber, T. sacc, O. universa, but oxygen isotopes are used for P. obliquiloculata and Mg/Ca derived Ts are used for tumida and menardii. Clarification here is needed.

Response: A table was given in a previous version of this manuscript, but reviewers thought that was redundant with the table in the supplement (it was table 3), I have added the selected CDs in Table S7. You are right, it has been difficult to carefully select the CDs, I have tried to do it rigorously but sometimes it still remains arbitrary. To explain: I derived both calcification depth (CD) using $\delta$18O (CD1) and Mg/Ca (CD2). Because each of methods have their limits (species-specific calibrations for most of it, and analytical uncertainties for $\delta$18O) I compared the calcification depths to previously published in literature (CD3). I can't say what method is the best, but in my case I would tend toward Mg/Ca because we have species-specific calibrations for most of the species and the measurements were done on the same sample analyzed for $\delta$11B. For $\delta$18O I had to pick again. To select the CD, the first thing I did was to look at the water column structure at each site in order to have an idea of what depth I should expect the species to be depending on their habitats (mixed-dweller, within thermocline, below thermocline). Then I compared CD1, CD2 and CD3. If the CDs were in line with the water structure and two CDs were similar I chose that one. If CD1 and CD2 were different I chose CD3. I chose CD3 when possible because even if I have confidence in my data, there are papers that have more robust CD reconstructions (main goal of their papers). Now, for G. tumida and G. menardii, there were not a lot of data in the literature to compare with and there is no species-specific $\delta$18O calibrations but there are if using Mg/Ca. When I compared with the few data in the literature, my CD2 were in line than CD3 that is why I stick to CD2 for both species. For P. obliquiloculata, it was the same not a lot of data in the literature (only Sime et al.,), so we use Mg/Ca data (CD2), not $\delta$18O even if they are similar except (FCO2a).

RC2: I don't understand why O. universa is considered a deeper dweller in this MS. There are some major assumptions made about why the d11B of this species falls below the 1:1 line – this is also discussed in Henehan et al., 2013. Why O. universa d11B is more like the deeper dweller non-spinose forams is indeed puzzling (esp. since it has same symbionts as G. ruber and T. sacculifer), but I don't think it can be attributed to a deep depth, especially given the size fraction used (>500). The larger size fraction of the samples used would suggest that these are living at a shallower depth (See Spero and Parker, 2003) and likely in the mixed layer. The correct depth habitat will impact the calibration.

Response: Yes, it will change quite a lot the position of the data, especially for site FC0-1 that probably would be an outlier, but the calibration won't change significantly when compiling all data (see Figure below). Only based on the CD calculations, O. universa is found in average $\sim$60-70m which is in line with other depth habitats reconstructions, the fact that $\delta$11B is lowered than ambient pH can be explained by a depth habitat >50m. The data converge towards this low $\delta$11B due to a decrease in light insolation at deeper depth. Its high symbiont density and photosynthetic activity might make it, as T. sacculifer, more sensitive to water depth changes as well, or we could assume a change in symbiont assemblage to ones that are more similar to what are found in deeper dweller species, changing host/symbiont interactions, photosynthesis and related microenvironment pH.

RC2: Samples were cleaned using the full cleaning method including the reductive step (cite Boyle and Kiegwin 1985/1986 as well, since they developed the method). Why was the reductive step included here? Yu et al., 2007 suggests the reductive step isn't detrimental to B/Ca ratios, but the effect of this cleaning step on for d11B analysis is unknown and according to Rae et al., 2018, the reductive step is typically not used during sample cleaning. There are documented dissolution effects that the authors discuss in the supplement (preferential dissolution of ontogenetic calcite occurs relative to the light d11B of gam calcite) and if cleaning preferentially removes ontogenetic calcite, then the primary d11B signal has been altered by the reductive cleaning. Additionally, the reductive cleaning step IS detrimental to other elements, like Mg/Ca ratios (decreases ratios by up to 15%), which were used to estimate depth habitat for some of the species investigated here. Including the reductive step should be justified with an explanation of how this additional cleaning step could have affected results.

Response: Misra et al., (2014b) tested the different cleaning on the B/Ca and $\delta$11B and observed no effect. However, in nodules (Mn-Fe) the concentration of B can be up to 120 ppm (Axelsson et al., 2002), Fe-Mn oxide and hydroxides can then result in non-negligible content of Mg and B contamination. We utilized the reductive step because some of the sites where not previously studied, overall the sites did not present high Mn, except for E035 which was removed due to that contamination.

Also, results of Mg/Ca would lead to a deeper CDs which is not the case when comparing with CD1 and CD3. Axelsson, M. D., Rodushkin, I., Ingri, J. and Öhlander, B.: Multielemental analysis of Mn–Fe nodules by ICP-MS: optimisation of analytical method, Analyst, 127, 76–82, 2002. Misra, S., Owen, R., Kerr, J., Greaves, M. and Elderfield, H.: Determination of $\delta$11B by HR-ICP-MS from mass limited samples: Application to natural carbonates and water samples, Geochim. Cosmochim. Acta, 140, 531–552, 2014b.

RC2: Li/Ca ratios are included in table 2, but never discussed?? Are they relevant for this MS? If not, perhaps remove?

Response: Li/Ca are not discussed in this paper, however there is a growing interest for Li in foraminifera, we decided to publish those results in order to contribute to the existing data.

RC2: Table 3: In the text on line 329 it is stated that chemo stratigraphic data is used to constrain depths, but this information is not summarized in the table, probably just a typo?

Response: It was a typo, changed. The CD chosen are in Table S7.

RC2: Other suggestions: Line 56-57: Not sure I agree with the statement in the abstract that the other species follow O. universa because of light limitation by symbiont bearing foraminifera. All of the deep dwellers have symbionts, all live in the photic zone.

Response: I have tried to improve the discussion, focusing on the symbionts/photosynthesis, because the story is of course more complex. Yes, all of the species have algal symbionts but they can differ. From what I see in the literature is that T. sacculifer, G. ruber, O. universa have mostly dinoflagellates symbionts (can have chrysophyte as well) where G. tumida, G. menardii, P. obliquiloculata and N. dutertrei will have chrysophyte algal symbionts. The photosynthesis is dependent of the nature of the host/symbionts interactions, symbionts type (pigment associated for light absorption efficiency), symbionts density. The recent study from Tagaki et al., (2019) is really helpful as he constrained the photosynthesis activity, light absorption efficiency and the symbiont density of those species.

Fv/Fm (photosyntethic activity) T. sacculifer>G. menardii > O. universa> G. ruber (white) > N. dutertrei > P. obliquiloculata $\sigma$psi (light absorption efficiency) N. dutertrei > P. obliquiloculata > G. menardii > G. ruber > T. sacculifer > O. universa Chla/biomass T. sacculifer>O.universa>G. ruber> N. dutertrei>G. menardii>P. obliquiloculata

What I assume is that T. sacculifer, O. universa and G. ruber photosynthesis are likely to be more affected by changes in insolation than other species due to their symbiont density, high photosynthetic capacity and their light absorption efficiency. Which is still in line with the argumentation we are giving. Also, the fact that the deeper dwellers have this low boron isotopic signature is likely due to a lower symbiont density, lower photosynthetic activity and a reduced insolated environment. P. obliquiloculata has the lowest density and photosynthetic activity, which would translate in a respiration driven environment the fact that most of the species are following this trend would go in the sense of a respiration driven environment. Also the fact that O. universa is following this trend would, I think, whether be due light limitation and/or a different symbiont due to its deeper depth.

Lines 485-543.

I also changed Figure 7 to give more informations on the microenvironment pH.

I changed for "likely caused by light limitation and/or symbiont/host interactions"

RC2: Line 128: This sentence is poorly structured.

Response: Changed for: "Furthermore, $\delta$11B differences between foraminifera species from the same pH makes the acquisition of more core-top and culture data essential for testing and applying the proxy."

RC2: Secition 2.4: Origin of biological fractionation Paragraph beginning on line 172: very speculative and based upon benthic foraminifer experiments.

We have added a statement (line 176) to this effect, that "We acknowledge this is speculative as it is based upon benthic foraminifer experiments."

RC2: Section 2.5: The annual vs seasonal preferences for forams is largely dependent on temperatures. For example, in some regions G. ruber and G. sacculifer can be present throughout the year if T > 25C, but will have a summer/fall preference when T drops below 15. Their choice on seasonal vs. annual presence of these species will affect the hydrographic data used and perhaps impact results.

Response: Noted.

RC2: Section 3.2: Size fractions listed in this paragraph don't agree with size fractions in the Table.

Response: Changed for: "Around 50-100 foraminifera shells were picked from the 400-

$\mu$m fraction size for Globorotalia menardii and Globorotalia tumida, >500 $\mu$m for Orbulina universa, from the 250-400 $\mu$m fraction size for Trilobatus sacculifer (w/o sacc, without sacc-like final chamber), Trilobatus sacculifer (sacc, sacc-like final chamber), Globigerinoides ruber (white, sensu stricto), Neogloboquadrina dutertrei, Pulleniatina obliquiloculata. The samples picked for analyses were visually well preserved."

RC2: Section 3.8: This section could/should include some of the information in the supplement regarding the depth used to obtain hydrographic information.

Response: We acknowledge it was not clear in the manuscript. The information can be found in Table S7. Line 341: "The depth habitats utilized to derive in situ parameters are summarized in Table S7."

RC2: Line 339: Forams don't migrate in the water column (See Meiland et al., 2019), but deep dwellers may crust at depth during the END of their lifecycle, this should be clarified. This is later explained correctly (lines 452-453).

Response: Changed: line 338 "As foraminifera can migrate in the water column along their ontogeny". It is still vague but I don't know how constrained is this depth change at the end of their life cycle.

RC2: Lines 509-514: The concept of facultative symbiosis is outdated – all forams with symbionts are likely obligate and not facultative. See https://www.biogeosciences.net/16/3377/2019/. G. tumida doesn't have symbionts at all, so why does it align with the other species? Please discuss.

Response: But Tagaki et al., (2019) didn't constrain G. tumida in his study. From what I have read the family Globorotaliidae have algal chrysphyte symbionts, which should include G. tumida. If there are no symbionts or a low symbiont density/low photosynthetic activity (like P. obliquiloculata) the respiration driven microenvironment resulting from the parameters aforementioned or again light limitation would be a plausible explanation.

I have made changes in this section in order to improve the discussion. Lines 485-543.

Please also note the supplement to this comment:
https://www.biogeosciences-discuss.net/bg-2019-266/bg-2019-266-AC3-supplement.pdf

—————————————————————

[Figure]

[Figure]

Fig. 1.

[Figure]

**Fig. 2.**

**Supplement:**

**Response to RC2**

We wish to thank this reviewer for his thorough review of our manuscript and his helpful comments. We believe that we addressed all of the major comments as indicated in the discussion below and the updated document.

The updated manuscript and figures can be found at the end after this response.

**RC2:** Depth habitats of the foraminifers are used to link the habitat to oceanographic conditions at the core locations. The supplement has a fairly thorough explanation of how the various depths were determined (using d18Oc, MgCa-derived T, and published depth habitats; detailed in Table 6), but I couldn't determine which depth the authors chose to use for each species (or at least is wasn't consistently clear in the text) without seeking the information in Table 7. In the intro of the MS, the authors point the reader to table 3 for the depth habitats but table 3 doesn't include this information. Also, I don't understand why sometimes the authors used published references for depth habitats and in other instances is d18O or Mg/Ca derived temperatures. For example: For core FC-01a, the Sime reference is cited for the depths used for G. ruber, T. sacc, O. universa, but oxygen isotopes are used for P. obliquiloculata and Mg/Ca derived Ts are used for tumida and menardii. Clarification here is needed.

**Response:** *A table was given in a previous version of this manuscript, but reviewers thought that was redundant with the table in the supplement (it was table 3), I have added the selected CDs in Table S7. You are right, it has been difficult to carefully select the CDs, I have tried to do it rigorously but sometimes it still remains arbitrary. To explain:*
*I derived both calcification depth (CD) using δ18O (CD1) and Mg/Ca (CD2). Because each of methods have their limits (species-specific calibrations for most of it, and analytical uncertainties for δ18O) I compared the calcification depths to previously published in literature (CD3). I can't say what method is the best, but in my case I would tend toward Mg/Ca because we have species-specific calibrations for most of the species and the measurements were done on the same sample analyzed for δ11B. For δ18O I had to pick again.*
*To select the CD, the first thing I did was to look at the water column structure at each site in order to have an idea of what depth I should expect the species to be depending on their habitats (mixed-dweller, within thermocline, below thermocline). Then I compared CD1, CD2 and CD3. If the CDs were in line with the water structure and two CDs were similar I chose that one. If CD1 and CD2 were different I chose CD3. I chose CD₃ when possible because even if I have confidence in my data, there are papers that have more robust CD reconstructions (main goal of their papers).*
*Now, for G. tumida and G. menardii, there were not a lot of data in the literature to compare with and there is no species-specific δ18O calibrations but there are if using Mg/Ca. When I compared with the few data in the literature, my CD2 were in line than CD3 that is why I stick to CD2 for both species.*
*For P. obliquiloculata, it was the same not a lot of data in the literature (only Sime et al.,), so we use Mg/Ca data (CD2), not δ18O even if they are similar except (FCO2a).*

**RC2:** I don't understand why O. universa is considered a deeper dweller in this MS. There are some major assumptions made about why the d11B of this species falls below the 1:1 line – this is also discussed in Henehan et al., 2013. Why O. universa d11B is more like the deeper dweller non-spinose forams is indeed puzzling (esp. since it has same symbionts as G. ruber and T. sacculifer), but I don't think it can be attributed to a deep depth, especially given the size fraction used (>500). The larger size fraction of the samples used would suggest that these are living at a shallower depth (See Spero and Parker, 2003) and likely in the mixed layer. The correct depth habitat will impact the calibration.

**Response:** *Yes, it will change quite a lot the position of the data, especially for site FC0-1 that probably would be an outlier, but the calibration won't change significantly when compiling all data (see Figure below).*

*Only based on the CD calculations, O. universa is found in average ~60-70m which is in line with other depth habitats reconstructions, the fact that δ11B is lowered than ambient pH can be explained by a depth habitat >50m. The data converge towards this low δ11B due to a decrease in light insolation at deeper depth. Its high symbiont density and photosynthetic activity might make it, as T. sacculifer, more sensitive to water depth changes as well, or we could assume a change in symbiont assemblage to ones that are more similar to what are found in deeper dweller species, changing host/symbiont interactions, photosynthesis and related microenvironment pH.*

[Figure]

**RC2:** Samples were cleaned using the full cleaning method including the reductive step (cite Boyle and Kiegwin 1985/1986 as well, since they developed the method). Why was the reductive step included here? Yu et al., 2007 suggests the reductive step isn't detrimental to B/Ca ratios, but the effect of this cleaning step on for d11B analysis is unknown and according to Rae et al., 2018, the reductive step is typically not used during sample cleaning. There are documented dissolution effects that the authors discuss in the supplement (preferential dissolution of ontogenetic calcite occurs relative to the light d11B of gam calcite) and if cleaning preferentially removes ontogenetic calcite, then the primary d11B signal has been altered by the reductive cleaning. Additionally, the reductive cleaning step IS detrimental to other elements, like Mg/Ca ratios (decreases ratios by up to 15%), which were used to estimate depth habitat for some of the species investigated here. Including the reductive step should be justified with an explanation of how this additional cleaning step could have affected results.

**Response:** *Misra et al., (2014b) tested the different cleaning on the B/Ca and δ11B and observed no effect. However, in nodules (Mn-Fe) the concentration of B can be up to 120 ppm (Axelsson et al., 2002), Fe-Mn oxide and hydroxides can then result in non-negligible content of Mg and B contamination. We utilized the reductive step because some of the sites where not previously studied, overall the sites did not present high Mn, except for E035 which was removed due to that contamination.*

*Also, results of Mg/Ca would lead to a deeper CDs which is not the case when comparing with CD1 and CD3.*

*Axelsson, M. D., Rodushkin, I., Ingri, J. and Öhlander, B.: Multielemental analysis of Mn–Fe nodules by ICP-MS: optimisation of analytical method, Analyst, 127, 76–82, 2002.*

*Misra, S., Owen, R., Kerr, J., Greaves, M. and Elderfield, H.: Determination of δ11B by HR-ICP-MS from mass limited samples: Application to natural carbonates and water samples, Geochim. Cosmochim. Acta, 140, 531–552, 2014b.*

**RC2:** Li/Ca ratios are included in table 2, but never discussed?? Are they relevant for this MS? If not, perhaps remove?

**Response:** *Li/Ca are not discussed in this paper, however there is a growing interest for Li in foraminifera, we decided to publish those results in order to contribute to the existing data.*

**RC2:** Table 3: In the text on line 329 it is stated that chemo stratigraphic data is used to constrain depths, but this information is not summarized in the table, probably just a typo?

**Response:** *It was a typo, changed. The CD chosen are in Table S7.*

**RC2:** Other suggestions: Line 56-57: Not sure I agree with the statement in the abstract that the other species follow O. universa because of light limitation by symbiont bearing foraminifera. All of the deep dwellers have symbionts, all live in the photic zone.

**Response:** *I have tried to improve the discussion, focusing on the symbionts/photosynthesis, because the story is of course more complex.*
*Yes, all of the species have algal symbionts but they can differ. From what I see in the literature is that T. sacculifer, G. ruber, O. universa have mostly dinoflagellates symbionts (can have chrysophyte as well) where G. tumida, G. menardii, P. obliquiloculata and N. dutertrei will have chrysophyte algal symbionts. The photosynthesis is dependent of the nature of the host/symbionts interactions, symbionts type (pigment associated for light absorption efficiency), symbionts density. The recent study from Tagaki et al., (2019) is really helpful as he constrained the photosynthesis activity, light absorption efficiency and the symbiont density of those species.*

*Fv/Fm (photosynthethic activity)  T. sacculifer>G. menardii > O. universa> G. ruber (white) > N. dutertrei > P. obliquiloculata*

*σpsi (light absorption efficiency)  N. dutertrei > P. obliquiloculata > G. menardii > G. ruber > T. sacculifer > O. universa*

*Chla/biomass  T. sacculifer>O.universa>G. ruber> N. dutertrei>G. menardii>P. obliquiloculata*

*What I assume is that T. sacculifer, O. universa and G. ruber photosynthesis are likely to be more affected by changes in insolation than other species due to their symbiont density, high photosynthetic capacity and their light absorption efficiency. Which is still in line with the argumentation we are giving. Also, the fact that the deeper dwellers have this low boron isotopic signature is likely due to a lower symbiont density, lower photosynthetic activity and a reduced insolated environment. P. obliquiloculata has the lowest density and photosynthetic activity, which would translate in a respiration driven environment the fact that most of the species are following this trend would go in the sense of a respiration driven environment.*

*Also the fact that O. universa is following this trend would, I think, whether be due light limitation and/or a different symbiont due to its deeper depth.*

*Lines 485-543.*

*I also changed Figure 7 to give more informations on the microenvironment pH.*

*I changed for "likely caused by light limitation and/or symbiont/host interactions"*

**RC2:** Line 128: This sentence is poorly structured.

**Response:** *Changed for: "Furthermore, δ11B differences between foraminifera species from the same pH makes the acquisition of more core-top and culture data essential for testing and applying the proxy."*

**RC2:** Secition 2.4: Origin of biological fractionation Paragraph beginning on line 172: very speculative and based upon benthic foraminifer experiments.

*We have added a statement (line 176) to this effect, that "We acknowledge this is speculative as it is based upon benthic foraminifer experiments."*

**RC2:** Section 2.5: The annual vs seasonal preferences for forams is largely dependent on temperatures. For example, in some regions G. ruber and G. sacculifer can be present throughout the year if T > 25C, but will have a summer/fall preference when T drops below 15. Their choice on seasonal vs. annual presence of these species will affect the hydrographic data used and perhaps impact results.

**Response:** *Noted.*

**RC2:** Section 3.2: Size fractions listed in this paragraph don't agree with size fractions in the Table.

**Response:** *Changed for: "Around 50-100 foraminifera shells were picked from the 400-500 μm fraction size for Globorotalia menardii and Globorotalia tumida, >500 μm for Orbulina universa, from the 250-400 μm fraction size for Trilobatus sacculifer (w/o sacc, without sacc-like final chamber), Trilobatus sacculifer (sacc, sacc-like final chamber), Globigerinoides ruber (white, sensu stricto), Neogloboquadrina dutertrei, Pulleniatina obliquiloculata. The samples picked for analyses were visually well preserved."*

**RC2:** Section 3.8: This section could/should include some of the information in the supplement regarding the depth used to obtain hydrographic information.

**Response:** *We acknowledge it was not clear in the manuscript. The information can be found in Table S7. Line 341: "The depth habitats utilized to derive in situ parameters are summarized in Table S7."*

**RC2:** Line 339: Forams don't migrate in the water column (See Meiland et al., 2019), but deep dwellers may crust at depth during the END of their lifecycle, this should be clarified. This is later explained correctly (lines 452-453).

**Response:** *Changed: line 338 "As foraminifera can migrate in the water column along their ontogeny". It is still vague but I don't know how constrained is this depth change at the end of their life cycle.*

**RC2:** Lines 509-514: The concept of facultative symbiosis is outdated – all forams with symbionts are likely obligate and not facultative. See https://www.biogeosciences.net/16/3377/2019/. G. tumida doesn't have symbionts at all, so why does it align with the other species? Please discuss.

**Response:** *But Tagaki et al., (2019) didn't constrain G. tumida in his study.*
*From what I have read the family Globorotaliidae have algal chrysphyte symbionts, which should include G. tumida. If there are no symbionts or a low symbiont density/low photosynthetic activity (like P. obliquiloculata) the respiration driven microenvironment resulting from the parameters aforementioned or again light limitation would be a plausible explanation.*

*I have made changes in this section in order to improve the discussion. Lines 485-543.*

[revised manuscript text omitted]

In planktonic foraminifera, algal symbiosis is the more common symbiotic relationship. For most of planktonic foraminifera, the host presents only one species of symbionts (Gast and Caron, 2001). The family

Globigerinidae, including *G. ruber*, *T. sacculifer* and *O. universa*, commonly have dinoflagellates or chrysophyte algal symbionts (Anderson and Be, 1976; Spero, 1987). The families Pulleniatinidae, Globorotaliidae, including

*N. dutertrei*, *P. obliquiloculata, G. menardii* and *G. tumida*, have chrysophyte algal symbionts (Gastrich, 1988).

The relationship between the symbionts and the host is complex by nature. Nevertheless, this symbiotic relationship provides energy (Hallock, 1981b) and promotes calcification of the foraminifera (Duguay, 1983;

Erez et al., 1983) by providing the inorganic carbon to the host (Jorgensen et al., 1985). Also, for *T. sacculifer*

and *O. universa* photosynthesis increases with higher insolation (Jorgensen et al., 1985; Rink et al., 1998).

499    Dinoflagellate-bearing foraminifera (*G. ruber*, *T. sacculifer* and *O. universa*) tend to have a higher

500 symbiont density and photosynthesis activity while *P. obliquiloculata*, *G. menardii* and *N. dutertrei* have

501 lowered symbiont density and *P. obliquiloculata*, *N. dutertrei* lower photosynthetic activity (Takagi et al., 2019).

502 *P. obliquiloculata* showed the minimum symbiont density and photosynthetic activity (Takagi et al., 2019).

503    It is now accepted that the foraminifera $\delta^{11}B$ signature comes from the microenvironment pH

504 (Jorgensen et al., 1985; Rink et al., 1998; Köhler-Rink and Kühl, 2000, Hönisch et al., 2003; Zeebe et el., 2003).

505 Foraminifera with high photosynthetic activity and symbiont density like *G. ruber* and *T. sacculifer* present a pH

506 of microenvironment higher than ambient seawater, $\delta^{11}B$ higher than 1:1 line (Foster et al., 2008, Henehan et al.,

507 2013, Raitzsch et al., 2018). The opposite can also be true, from our study, species with lower photosynthetic

508 activity and lower symbiont density present microenvironments lower than ambient seawater, $\delta^{11}B$ lower than

509 1:1 line (Martinez-Boti et al., 2015b; Henehan et al., 2016), this is the case in our data for *N. dutertrei*, *G.*

510 *menardii* and *P. obliquiloculata* and likely *G. tumida*. Nevertheless, the low $\delta^{11}B$ of *O. universa* and *T.*

511 *sacculifer (w/o sacc)* from the WEP are difficult to reconcile with a high photosynthetic activity compared to *T.*

512 *sacculifer* et *G. ruber*.

513    The photosynthetic activity is also function of the light level they received which is, in the natural

514 system, dependent of their depth in the water column, for the purpose of this study we will not consider turbidity

515 which also influences the light penetration in the water column. In this case, the photosynthetically active

516 foraminifera living close to the surface should see their microenvironment pH (thus $\delta^{11}B$) more sensitive to water

517 depth changes. A deeper depth habitat will change the light intensity they received and as a consequence may

518 lower their photosynthetic activity reducing their microenvironment pH. This thought is supported by the

519 significant trend observed between our $\Delta^{11}B$ and the calcification depth for *G. ruber* and *T. sacculifer* of our sites

520 (Fig. S2). This trend basically supports the fact that the microenvironment pH decrease with calcification depth.

521 We observe a decrease of $\delta^{11}B$ in the WEP for *T. sacculifer (w/o sacc)*, significantly different from the other sites

522 (p<0.05). The $\Delta^{11}B$ of *G. ruber*, *T. sacculifer* (w/o sacc and sacc) is also significantly lower in the WEP

523 compared to the other sites (p<0.05). To test if the $\delta^{11}B$ signature was inferred to a light driven, we have been

524 able to independently calculate the depth of the foraminifera based on various light insolation culture

525 experiments (Jorgensen et al., 1985) and the $\Delta$microenvironment pH derived from our data (Fig. 8A and B). This

526 exercise verified that this low $\delta^{11}B$ can be explained by the reduced light environment due to a deeper depth

527 habitat in the WEP (Fig. 8B). Also, *T. sacculifer* has the potential to support more photosynthesis due to its

528 higher symbiont density. Higher photosynthetic activity is observed compared to other species potentially

529 supporting higher symbiont/host interactions. Those results could be in line with a greater sensitivity of *T.*

530 *sacculifer* photosynthetic activity with changes in insolation/water depth. It can also be noted that this species

531 presents the largest variations in symbiont density versus its test size. When applied to the other species *O.*

532 *universa* data suggest a microenvironment pH 0.10 to 0.20 lower than ambient seawater pH which would be in

533 line with species living deeper than 50m (light compensation point (Ec), Rink et al., 1998) also consistent with

534 our calcification depth reconstructions.  $\Delta$microenvironment pH is higher in *T. sacculifer* > *G. ruber* > *T.*

535 *sacculifer* (w/o sacc - WEP) > *O. universa*, *N. dutertrei*, *G. menardii*, *G. tumida* > *P. obliquiloculata* in line with

536 photosymbiosis findings from Tagazaki et al., (2019).  Also, the higher $\delta^{11}B$ data from the African upwelling

537 published by Raitzsch et al., (2018) for *G. ruber* and *O. universa* might reflect the higher microenvironment pH

538 due to a shallower depth habitat. This could highlight a potential issue with calibration when applied to sites with different oceanic regimes as the $\delta^{11}B$ specie-specific calibrations could be also location-specific for the mixed dweller species.

   Microenvironment pH results for *N. dutertrei, G. menardii, G. tumida,* are similar to *O. universa* and suggest a threshold for respiration driven $\delta^{11}B$ signature. This threshold can be driven by a change of photosynthetic activity due to lower light intensity at deeper depth and/or a change in the symbiont assemblage with non-dinoflagellate symbionts at deeper depth. We can explain this threshold because deep dweller species do not experience important changes of insolation at those depths so their microenvironments should be respiration driven and relatively stable. We can also note that *P. obliquiloculata* which has the lowest symbiont density and photosynthetic activity has the lowest microenvironment pH compared to other deeper dweller species supporting this respiration driven microenvironment.

**5.3 $\delta^{11}B$ sensitivity to $\delta^{11}B_{borate}$ and relationship with B/Ca signatures**

   $\delta^{11}B_{carbonate}$ and B/Ca data have shown to be sensitive to precipitation rate with at higher precipitation rate increasing $\delta^{11}B_{carbonate}$ (Farmer et al., 2019) and B/Ca (Farmer et al., 2019; Gabitov et al., 2014; Kaczmarek et al., 2016; Mavromatis et al., 2015; Ushikawa et al., 2015). A recent study from Farmer et al, (2019) has proposed that in foraminifera at higher precipitation rates, more borate ion is incorporated into the carbonate mineral, while at lower precipitation rates, more boric acid is incorporated. They also suggest this may explain low sensitivies of culture experiments.

   When combining all literature data, *T. sacculifer* and *G. ruber* have sensitivities of $\delta^{11}B_{carbonate}$ to

$\delta^{11}B_{borate}$ of 0.83 ± 0.48 and 0.46 ± 0.34 respectively in line with previous literature and paleo-$CO_2$

reconstructions. Also, if we only take into account our data, the observation that the sensitivity of $\delta^{11}B_{carbonate}$ to

$\delta^{11}B_{borate}$ are not statistically different from unity for most of the species investigated we can speculate that for these taxa, changes in precipitation rate and contributions of boric acid are not likely to be important. If considering only the data from this study, *G. ruber* (1.12 ± 1.67) and *T. sacculifer* (1.38± 1.35) present higher sensitivities of $\delta^{11}B_{carbonate}$ to $\delta^{11}B_{borate}$. We can then again speculate that the observed high values for $\delta^{11}B_{carbonate}$

at high seawater pH can be due to higher precipitation rates. We note this could also be consistent with the higher sensitivity of B/Ca signatures in these two surface dwelling species to ambient $[B(OH)_4^-]/[HCO_3^-]$ relative to deeper dwelling species. Those interspecific differences still remain to be explained, however, part of this variability is likely due to changes in the carbonate chemistry of the microenvironment resulting in changing competition between borate and bicarbonate. A caveat is that we can not exclude specific biological processes, and that in taxa with a non respiration-driven microenvironment, changes in day/night calcification ratios also impacting observed values. As indicated by Farmer et al., (2019), studies of calcite precipitation rates in foraminifera may help to improve our understanding of the fundamental basis of boron-based proxies.

**5.4 Evaluation of species for pH reconstructions and water depth pH reconstructions**

   This data set allows us to reassess the utility of boron-based proxies for the carbonate system. The main interest with utilizing boron-based proxies relates to the reconstruction of past oceanic conditions - specifically pH and p$CO_2$. Mixed-layer species (eg. *G. ruber* and *T. sacculifer)* are potential archives for atmospheric $CO_2$

reconstructions. Other species can shed light on other aspects of the carbon cycle including the physical and biological carbon pumps.

There are a few main inferences we can make. When compiled with data from the literature,
sensitivities of $\delta^{11}B_{carbonate}$ to $\delta^{11}B_{borate}$ for *G. ruber* and *T. sacculifer* are similar to previous studies (Martinez-
Boti et al., 2015b; Raitzsch et al., 2018) which is also supporting of previous paleo-reconstructions. Our data
also support the observations of Henehan et al., (2016) for *O. universa*.

In order to derive accurate reconstructions of past ambient pH and pCO$_2$, accurate species-specific
calibrations need to be used that are constrained by core-tops or samples from similar types of settings (Fig. 9,
10, S6). Lighter $\delta^{11}B$ signatures in *T. sacculifer* (w/o sacc) are observed in the WEP, which may be explained by
the deeper depth habitat for these taxa, where lower light levels might reduce symbiont photosynthetic activity.
Also, correction will be needed for *T. sacculifer* (w/o sacc) in the WEP. When applying the calibrations n°2 and
4 to *T. sacculifer* and *G. ruber* (compilation of all data, Table 3) our data show more variability, especially for *G.*
*ruber* which lead to the larger mismatch compared to *in-situ* parameters. Henehan et al., (2013) reported a lighter
$\delta^{11}B$ with smaller test size, our sample add a weight/shell of 11 ± 4 µg (n=4, SD) which, despite a narrow range,

[revised manuscript text omitted]

(A)

δ¹¹B$_{carbonate}$ (‰)

δ¹¹B$_{borate}$ (‰)

*G. ruber*

- ● δ¹¹B$_{G. ruber}$ (core-top, this study)
- ◇ δ¹¹B$_{G. ruber}$ (core-top, Foster et al., 2008)
- ◐ δ¹¹B$_{G. ruber}$ (core-top, 250-300μm, Henehan et al., 2013)
- ○ δ¹¹B$_{G. ruber}$ (core-top, Henehan et al., 2013)
- □ δ¹¹B$_{G. ruber}$ (sediment trap, Henehan et al., 2013)
- ◈ δ¹¹B$_{G. ruber}$ (tow, Henehan et al., 2013)
- ▽ δ¹¹B$_{G. ruber}$ (grab sample, Henehan et al., 2013)
- △ δ¹¹B$_{G. ruber}$ (Raizsch et al., 2018)
- ── *G. ruber* calibration line (All data, this study)
- ── *G. ruber* calibration line (Core-top, this study)
- ── *G. ruber* calibration line (Culture, Henehan et al., 2013)

[Figure]

(B)

δ¹¹B$_{carbonate}$ (‰)

δ¹¹B$_{borate}$ (‰)

*T. sacculifer*

- ● δ¹¹B$_{T.sacculifer (w/o sacc)}$ (core-top, this study)
- △ δ¹¹B$_{T. sacculifer (w/o sacc)}$ (core-top, Raitzsch et al., 2018)
- ★ δ¹¹B$_{T.sacculifer (sacc)}$ (core-top, this study)
- ◇ δ¹¹B$_{T. sacculifer (sacc)}$ (core-top, Foster et al., 2008)
- ── *T. sacculifer (w/o sacc and sacc)* calibration line (All data, this study)
- ── *T. sacculifer (w/o sacc and sacc)* calibration line (Core-top, this study)
- ── *T. sacculifer (s)* calibration line (Martinez-Boti et al., 2015)

[Figure]

(C)

δ¹¹B$_{carbonate}$ (‰)

δ¹¹B$_{borate}$ (‰)

*O. universa*

- ● δ¹¹B$_{O. universa}$ (core-top, this study)
- ○ δ¹¹B$_{O. universa}$ (core-top, Henehan et al., 2016)
- □ δ¹¹B$_{O. universa}$ (sediment trap, Henehan et al., 2016)
- ◇ δ¹¹B$_{O. universa}$ (tow, Henehan et al., 2016)
- △ δ¹¹B$_{O. universa}$ (core-top, Raitzsch et al., 2018)
- ── *O. universa* calibration line (core-top, this study)
- ── *O. universa* calibration line (this study, Henehan et al., 2016, Raitzsch et al., 2018)
- ── *O. universa* calibration line (wild, Henehan et al., 2016)

[Figure]

[Figure]

[Figure]

- ⬤ (yellow) $\delta^{11}B_{P.obliquiloculata}$ (Core-top, this study)
- ▽ $\delta^{11}B_{P.obliquiloculata}$ (Henehan et al., 2016)
- ▬ (yellow) *P. obliquiloculata* calibration line (this study, Henehan et al., 2016)
- ▬ (grey) *O. universa* calibration curve (Henehan et al., 2016)

- ⬤ (purple) $\delta^{11}B_{G.\ menardii}$ (this study)
- ▬ (grey) *O. universa* calibration curve (Henehan et al., 2016)

[Figure]

[Figure]

- ⬤ (green) $\delta^{11}B_{N.\ dutertrei}$ (Core-top, this study)
- ◇ $\delta^{11}B_{N.\ dutertrei}$ (Core-top, Foster et al., 2008)
- ▬ (green thick) *O. universa* calibration line (This study)
- ▬ (green thin) *O. universa* calibration line (This study, Foster et al., 2008)
- ▬ (grey) *O. universa* calibration line (Henehan et al., 2016)

- ⬤ (violet) $\delta^{11}B_{G.\ tumida}$ (this study)
- ▬ (grey) *O. universa* calibration curve (Henehan et al., 2016)

[Figure]

- ⬤ (orange) $\delta^{11}B_{O.universa}$
- ⬤ (yellow) $\delta^{11}B_{P.obliquiloculata}$
- ⬤ (green) $\delta^{11}B_{N.\ dutertrei}$
- ⬤ (blue) $\delta^{11}B_{G.\ menardii}$
- ⬤ (violet) $\delta^{11}B_{G.\ tumida}$
- △ $\delta^{11}B_{deep-dweller}$ from literature
- ▬ (black) Deep-dweller calibration line
- ▬ (grey) *O. universa* calibration line (Henehan et al., 2016)

[Figure]

[Figure]

Photosynthesis/respiration-calcification

**A**

$\Delta$ microenvironment pH

0.2

0.1

0.0

-0.1

-0.2

-0.3

-0.4

*T. sacculifer*
$\mu = +0.047$
n = 5

*G. ruber*
$\mu = +0.027$
n = 5

*T. sacculifer (w/o sacc)* (WEP)
$\mu = -0.022$
n = 4

Compensation point

*O. universa*
$\mu = -0.153$
n = 5

*N. dutertrei*
$\mu = -0.159$
n = 5

*G. menardii*
$\mu = -0.163$
n = 4

*G. tumida*
$\mu = -0.168$
n = 3

*P. obliquiloculata*
$\mu = -0.240$
n = 5

**B**

Light penetration WEP (%)

20   40   60   80   100

Depth (m)

$\Delta pH_0 = -0.02$
$\Delta pH_1 = -0.04$
$\Delta pH_2 = -0.06$

[revised manuscript text omitted]

**Figure S5:** Figure evaluating the circularity of our reconstructions. It is showing in the y-axis the difference between reconstruction utilizing calibrations derive from the entire dataset and compare to *in-situ* values and in the x-axis the difference between the reconstruction utilizing the species-specific calibrations derived excluding the site of interest (no circularity) compared to *in-situ* values. Results show that difference is not significant between the two reconstruction methods (e.g. following the 1:1 line), validating the method and the calibrations.

[revised manuscript text omitted]

---

## Author Response (AR1)

Dear Reviewers,

First of all, we would like to thank you for your detailed comments that helped to improve the manuscript. We think we have addressed all of them. You will find the new version of the manuscript, with changes highlighted attached to this letter, along with a clean version of the manuscript.

To comment on a few points:

One of the main comments we adressed is the sensitivities of $\delta^{11}B_{carbonate}$ to $\delta^{11}B_{borate}$, >1 for *G. ruber* and *T. sacculifer*. We acknowledge that the sensitivities are not statistically different from published literature. When doing the bootstrap analysis on all compiled data, the regression for *G. ruber* was similar to Raitzsch et al., (2018), with a sensitivity of $0.46 \pm 0.34$ (updated table 3) compared to $0.45 \pm 0.16$ for Raitzsch et al., (2018) and the sensitivity was of $0.83 \pm 0.48$ for *T. sacculifer* in line with Martinez-Boti et al., (2015b) and Raitzsch et al., (2018). We also take into account the $pCO_2$ based $\delta^{11}B$ reconstructions against the Vostok $pCO_2$ record as pointed out. We also have clearly written that possible sensitivities of >1 are speculative.

Another major change is the reconstruction of microenvironment pH for each species (Figure 8) which we believe are in line with results of a study of photosynthesis/symbiont density from Takagi et al., (2019) for different species. This also helps to explain a possible light-limiting threshold for calibrations. We have also added discussion around symbionts-host interactions of different species.

We do observe changes in calcification depth with thermocline depth for most of the deeper-dwelling species (*G. ruber, T. sacculifer*) which might be problematic for future reconstructions over periods when there is evidence for changes in stratification. However, our data are too limited to derive a robust trend.

We understand that our discussion on the Western Equatorial Pacific in the earlier draft was limited as it was based on a single site. In order to strengthen the discussion, we added two other sites, ODP Sites 806A and 807A. The data record similar, low values to what we reported for site WPO7, and add to our confidence in the results.

Also, a reviewer pointed out a potential issue with the circularity of some of the calculations presented in the last manuscript. In order to address this issue, we recalculate for each species a calibration that excludes results for a single site, and then use the modified calibration that contains a subset of calibration data to reconstruct vertical profiles of carbonate chemistry as shown in Figure 9. What Fig. S5 shows is the comparison of vertical profiles using these two approaches, which yield consistent results. On average, the difference in $\delta^{11}B$ is 0.8% for $^{11}B_{borate}$, 0.2% for pH and 5% for $pCO_2$. The fact that using a subset of the calibration data and all of the calibration data yields similar results validates the work.

Reviews are adressed in the commented version of the manuscript and in our response to reviewers. We thank you for your time.

Best Regards,

Seawater pH reconstruction using boron isotopes in multiple planktonic foraminifera species with
different depth habitats and their potential to constrain pH and pCO$_2$ gradients

Maxence Guillermic[1,2], Sambuddha Misra[3,4], Robert Eagle[1,2], Alexandra Villa[2,5], Fengming Chang[6],
                     Aradhna Tripati [1,2]
[1] Department of Earth, Planetary, and Space Sciences, Department of Atmospheric and Oceanic
Sciences, Institute of the Environment and Sustainability, UCLA, University of California – Los
Angeles, Los Angeles, CA 90095 USA
[2] Laboratoire Géosciences Océan UMR6538, UBO, Institut Universitaire Européen de la Mer, Rue
Dumont d'Urville, 29280, Plouzané, France
[3] Indian Institute of Science, Centre for Earth Sciences, Bengaluru, Karnataka 560012, India
[4] The Godwin Laboratory for Palaeoclimate Research, Department of Earth Sciences, University of
Cambridge, UK
[5] Department of Geology, University of Wisconsin-Madison, Madison, WI 53706 USA
[6] Key Laboratory of Marine Geology and Environment, Institute of Oceanology, Chinese Academy of
Sciences, Qingdao 266071, China

Submitted to Biogeosciences

*Corresponding author:
E-mail address: maxence.guillermic@gmail.com

**ABSTRACT**

Boron isotope systematics of planktonic foraminifera from core-top sediments and culture experiments have been studied to investigate the sensitivity of $\delta^{11}B$ of their calcite tests to seawater pH. However, our knowledge of the relationship between $\delta^{11}B$ and pH remains incomplete for many taxa. Thus, to expand the potential scope of application of this proxy, we report $\delta^{11}B$ data for 7 different species of planktonic foraminifera from sediment core-tops. We utilize a method for the measurement of small samples of foraminifera and calculate the $\delta^{11}B$- calcite sensitivity to pH for *Globigerinoides ruber*, *Trilobus sacculifer* (sacc or w/o sacc), *Orbulina universa*,

*Pulleniatina obliquiloculata*, *Neogloboquadrina dutertrei*, *Globorotalia menardii* and *Globorotalia tumida,*

including for unstudied core-tops and species. The sensitivity of $\delta^{11}B_{carbonate}$ to $\delta^{11}B_{borate}$ (eg.

$\Delta\delta^{11}B_{carbonate}/\Delta\delta^{11}B_{borate}$) in core-tops is consistent with previous studies for *T. sacculifer* and *G. ruber* and close to unity for *N. dutertrei*, *O. universa* and combined deep-dwelling species. Deep-dwelling species closely follow the core-top calibration for *O. universa,* which is attributed to respiration-driven microenvironments, likely caused by light limitation and/or symbiont/host interactions. These taxa have diverse ecological preferences and are from sites that span a range of oceanographic regimes, including some that are in regions of air-sea equilibrium and others that are out of equilibrium with the atmosphere. Our data support the premise that utilizing boron isotope measurements of multiple species within a sediment core can be utilized to constrain vertical profiles of pH and $pCO_2$ at sites spanning different oceanic regimes, thereby constraining changes in vertical pH gradients and yielding insights into the past behavior of the oceanic carbon pumps.

**1. Introduction**

The oceans are absorbing a substantial fraction of anthropogenic carbon emissions resulting in declining surface ocean pH (Fig. 1; IPCC, 2014). Yet there is a considerable uncertainty over the magnitude of future pH change in different parts of the ocean and the response of marine biogeochemical cycles to physio-chemical parameters (T, pH) caused by climate change (Bijma et al., 2002; Ries et al., 2009). Therefore, there is an increased interest in reconstructing past seawater pH (Hönisch and Hemming, 2005; Liu et al., 2009; Wei et al., 2009; Douville et al., 2010), in understanding spatial variability in aqueous pH and carbon dioxide ($p\text{CO}_2$) (Foster et al., 2008; Martinez-Boti et al., 2015b; Raitzsch et al., 2018), and in studying the response of the biological carbon pump utilizing geochemical proxies (Yu et al., 2007, 2010, 2016).

Although proxies for carbon cycle reconstruction are complex in nature (Pagani et al., 2005; Tripati et al., 2009, 2011; Allen and Hönisch, 2012), the boron isotope composition of foraminiferal tests is emerging as one of the more robust candidates (Hönisch et al., 2005, 2009; Ni et al., 2007; Foster et al., 2008, 2012; Bartoli et al., 2011; Henehan et al., 2013; Martinez-Boti et al., 2015b; Chalk et al., 2017). The study of laboratory cultured foraminifera has demonstrated a systematic dependence of the boron isotope composition of tests on ambient pH (Sanyal et al., 1996, 2001; Henehan et al., 2013, 2016). Core-top measurements on globally distributed samples also show a $\delta^{11}\text{B}$ sensitivity to pH with taxa-specific offsets from the theoretical fractionation line of borate ion (Rae et al., 2011; Henehan et al., 2016; Raitzsch et al., 2018).

Knowledge of seawater pH, in conjunction with constraints on one other carbonate system parameter (Total Alkalinity (TA), DIC (dissolved inorganic carbon), $[\text{HCO}_3^-]$, $[\text{CO}_3^{2-}]$), can be utilized to constrain aqueous $p\text{CO}_2$. Application of empirical calibrations for boron isotopes, determined for select species of foraminifera from core-tops and laboratory cultures, has resulted in accurate reconstructions of $p\text{CO}_2$ utilizing downcore samples from sites that are in quasi-equilibrium with the atmosphere at present. $\delta^{11}\text{B}_{carbonate}$ based reconstructed values of $p\text{CO}_2$ are analytically indistinguishable from ice core $\text{
[revised manuscript text omitted]

In planktonic foraminifera, algal symbiosis is the more common symbiotic relationship. For most of planktonic foraminifera, the host presents only one species of symbionts (Gast and Caron, 2001). The family

Globigerinidae, including *G. ruber*, *T. sacculifer* and *O. universa*, commonly have dinoflagellates or chrysophyte algal symbionts (Anderson and Be, 1976; Spero, 1987). The families Pulleniatinidae, Globorotaliidae, including

*N. dutertrei, P. obliquiloculata, G. menardii* and *G. tumida*, have chrysophyte algal symbionts (Gastrich, 1988).

The relationship between the symbionts and the host is complex by nature. Nevertheless, this symbiotic relationship provides energy (Hallock, 1981b) and promotes calcification of the foraminifera (Duguay, 1983;

Erez et al., 1983) by providing the inorganic carbon to the host (Jorgensen et al., 1985). Also, for *T. sacculifer* and *O. universa* photosynthesis increases with higher insolation (Jorgensen et al., 1985; Rink et al., 1998).

Dinoflagellate-bearing foraminifera (*G. ruber*, *T. sacculifer* and *O. universa*) tend to have a higher symbiont density and photosynthesis activity while *P. obliquiloculata*, *G. menardii* and *N. dutertrei* have lowered symbiont density and *P. obliquiloculata*, *N. dutertrei* lower photosynthetic activity (Takagi et al., 2019). *P. obliquiloculata* showed the minimum symbiont density and photosynthetic activity (Takagi et al., 2019).

It is now accepted that the foraminifera $\delta^{11}$B signature comes from the microenvironment pH (Jorgensen et al., 1985; Rink et al., 1998; Köhler-Rink and Kühl, 2000, Hönisch et al., 2003; Zeebe et el., 2003). Foraminifera with high photosynthetic activity and symbiont density like *G. ruber* and *T. sacculifer* present a pH of microenvironment higher than ambient seawater, $\delta^{11}$B higher than 1:1 line (Foster et al., 2008, Henehan et al., 2013, Raitzsch et al., 2018). The opposite can also be true, from our study, species with lower photosynthetic activity and lower symbiont density present microenvironments lower than ambient seawater, $\delta^{11}$B lower than 1:1 line (Martinez-Boti et al., 2015b; Henehan et al., 2016), this is the case in our data for *N. dutertrei*, *G. menardii* and *P. obliquiloculata* and likely *G. tumida*. Nevertheless, the low $\delta^{11}$B of *O. universa* and *T. sacculifer (w/o sacc)* from the WEP are difficult to reconcile with a high photosynthetic activity compared to *T. sacculifer* et *G. ruber*.

The photosynthetic activity is also function of the light level they received which is, in the natural system, dependent of their depth in the water column, for the purpose of this study we will not consider turbidity which also influences the light penetration in the water column. In this case, the photosynthetically active foraminifera living close to the surface should see their microenvironment pH (thus $\delta^{11}$B) more sensitive to water depth changes. A deeper depth habitat will change the light intensity they received and as a consequence may lower their photosynthetic activity reducing their microenvironment pH. This thought is supported by the significant trend observed between our $\Delta^{11}$B and the calcification depth for *G. ruber* and *T. sacculifer* of our sites (Fig. S2). This trend basically supports the fact that the microenvironment pH decrease with calcification depth. We observe a decrease of $\delta^{11}$B in the WEP for *T. sacculifer (w/o sacc)*, significantly different from the other sites (p<0.05). The $\Delta^{11}$B of *G. ruber*, *T. sacculifer* (w/o sacc and sacc) is also significantly lower in the WEP compared to the other sites (p<0.05). To test if the $\delta^{11}$B signature was inferred to a light driven, we have been able to independently calculate the depth of the foraminifera based on various light insolation culture experiments (Jorgensen et al., 1985) and the Δmicroenvironment pH derived from our data (Fig. 8A and B). This exercise verified that this low $\delta^{11}$B can be explained by the reduced light environment due to a deeper depth habitat in the WEP (Fig. 8B). Also, *T. sacculifer* has the potential to support more photosynthesis due to its higher symbiont density. Higher photosynthetic activity is observed compared to other species potentially supporting higher symbiont/host interactions. Those results could be in line with a greater sensitivity of *T. sacculifer* photosynthetic activity with changes in insolation/water depth. It can also be noted that this species presents the largest variations in symbiont density versus its test size. When applied to the other species *O. universa* data suggest a microenvironment pH 0.10 to 0.20 lower than ambient seawater pH which would be in line with species living deeper than 50m (light compensation point (Ec), Rink et al., 1998) also consistent with our calcification depth reconstructions. Δmicroenvironment pH is higher in *T. sacculifer > G. ruber > T. sacculifer* (w/o sacc - WEP) *> O. universa*, *N. dutertrei*, *G. menardii*, *G. tumida > P. obliquiloculata* in line with photosymbiosis findings from Tagazaki et al., (2019). Also, the higher $\delta^{11}$B data from the African upwelling published by Raitzsch et al., (2018) for *G. ruber* and *O. universa* might reflect the higher microenvironment pH

due to a shallower depth habitat. This could highlight a potential issue with calibration when applied to sites with different oceanic regimes as the $\delta^{11}B$ specie-specific calibrations could be also location-specific for the mixed dweller species.

Microenvironment pH results for *N. dutertrei, G. menardii, G. tumida,* are similar to *O. universa* and suggest a threshold for respiration driven $\delta^{11}B$ signature. This threshold can be driven by a change of photosynthetic activity due to lower light intensity at deeper depth and/or a change in the symbiont assemblage with non-dinoflagellate symbionts at deeper depth. We can explain this threshold because deep dweller species do not experience important changes of insolation at those depths so their microenvironments should be respiration driven and relatively stable. We can also note that *P. obliquiloculata* which has the lowest symbiont density and photosynthetic activity has the lowest microenvironment pH compared to other deeper dweller species supporting this respiration driven microenvironment.

**5.3 $\delta^{11}B$ sensitivity to $\delta^{11}B_{borate}$ and relationship with B/Ca signatures**

$\delta^{11}B_{carbonate}$ and B/Ca data have shown to be sensitive to precipitation rate with at higher precipitation rate increasing $\delta^{11}B_{carbonate}$ (Farmer et al., 2019) and B/Ca (Farmer et al., 2019; Gabitov et al., 2014; Kaczmarek et al., 2016; Mavromatis et al., 2015; Ushikawa et al., 2015). A recent study from Farmer et al, (2019) has proposed that in foraminifera at higher precipitation rates, more borate ion is incorporated into the carbonate mineral, while at lower precipitation rates, more boric acid is incorporated. They also suggest this may explain low sensitivities of culture experiments.

When combining all literature data, *T. sacculifer* and *G. ruber* have sensitivities of $\delta^{11}B_{carbonate}$ to

$\delta^{11}B_{borate}$ of 0.83 ± 0.48 and 0.46 ± 0.34 respectively in line with previous literature and paleo-CO$_2$

reconstructions. Also, if we only take into account our data, the observation that the sensitivity of $\delta^{11}B_{carbonate}$ to

$\delta^{11}B_{borate}$ are not statistically different from unity for most of the species investigated we can speculate that for these taxa, changes in precipitation rate and contributions of boric acid are not likely to be important. If considering only the data from this study, *G. ruber* (1.12 ± 1.67) and *T. sacculifer* (1.38± 1.35) present higher sensitivities of $\delta^{11}B_{carbonate}$ to $\delta^{11}B_{borate}$. We can then again speculate that the observed high values for $\delta^{11}B_{carbonate}$

at high seawater pH can be due to higher precipitation rates. We note this could also be consistent with the higher sensitivity of B/Ca signatures in these two surface dwelling species to ambient $[B(OH)_4^-]/[HCO_3^-]$ relative to deeper dwelling species. Those interspecific differences still remain to be explained, however, part of this variability is likely due to changes in the carbonate chemistry of the microenvironment resulting in changing competition between borate and bicarbonate. A caveat is that we can not exclude specific biological processes, and that in taxa with a non respiration-driven microenvironment, changes in day/night calcification ratios also impacting observed values. As indicated by Farmer et al., (2019), studies of calcite precipitation rates in foraminifera may help to improve our understanding of the fundamental basis of boron-based proxies.

**5.4 Evaluation of species for pH reconstructions and water depth pH reconstructions**

This data set allows us to reassess the utility of boron-based proxies for the carbonate system. The main interest with utilizing boron-based proxies relates to the reconstruction of past oceanic conditions - specifically pH and pCO$_2$. Mixed-layer species (eg. *G. ruber* and *T. sacculifer)* are potential archives for atmospheric CO$_2$

reconstructions. Other species can shed light on other aspects of the carbon cycle including the physical and biological carbon pumps.

There are a few main inferences we can make. When compiled with data from the literature, sensitivities of $\delta^{11}B_{carbonate}$ to $\delta^{11}B_{borate}$ for *G. ruber* and *T. sacculifer* are similar to previous studies (Martinez-

Boti et al., 2015b; Raitzsch et al., 2018) which is also supporting of previous paleo-reconstructions. Our data also support the observations of Henehan et al., (2016) for *O. universa*.

In order to derive accurate reconstructions of past ambient pH and $pCO_2$, accurate species-specific calibrations need to be used that are constrained by core-tops or samples from similar types of settings (Fig. 9,

10, S6). Lighter $\delta^{11}B$ signatures in *T. sacculifer* (w/o sacc) are observed in the WEP, which may be explained by the deeper depth habitat for these taxa, where lower light levels might reduce symbiont photosynthetic activity.

Also, correction will be needed for *T. sacculifer* (w/o sacc) in the WEP. When applying the calibrations n°2 and

4 to *T. sacculifer* and *G. ruber* (compilation of all data, Table 3) our data show more variability, especially for *G.*

*ruber* which lead to the larger mismatch compared to *in-situ* parameters. Henehan et al., (2013) reported a lighter

$\delta^{11}B$ with smaller test size, our sample add a weight/shell of 11 ± 4 µg (n=4, SD) which, despite a narrow range, could still explain this variability. The higher 
[revised manuscript text omitted]

**Figure S4:** Boron geochemistry against water depth. A) $\delta^{11}B_{carbonate}$ versus water depth, B) B/Ca against water depth and C) $\delta^{11}B_{carbonate}$ versus calcification depth and linear regressions for *G. ruber*, *T. sacculifer* (w/o sacc), *T. sacculifer* (sacc) and *O. universa*.

**Figure S5:** Figure evaluating the circularity of our reconstructions. It is showing in the y-axis the difference between reconstruction utilizing calibrations derive from the entire dataset and compare to *in-situ* values and in the x-axis the difference between the reconstruction utilizing the species-specific calibrations derived excluding the site of interest (no circularity) compared to *in-situ* values. Results show that difference is not significant between the two reconstruction methods (e.g. following the 1:1 line), validating the method and the calibrations.

[revised manuscript text omitted]

---

## Author Response (AR2)

Dear Reviewers,

Thank you for your patience and your comments on the manuscript. We are grateful for your time and believe the feedback has improved the manuscript. We hope you find it suitable for publication.

This version has incorporated all of your comments. As suggested, we developed and re-formatted section 5.2 following the structure proposed by J. Farmer and we believe that it is written more clearly and will be easier for the reader to follow.

As suggested by M. Henehan, due to evidence of size-fraction effects on the $\delta^{11}B_{carbonate}$ for *G. ruber*, we made a linear regression based only on smaller size fractions. The resulting calibration did not show a significant change in the sensitivity of $\delta^{11}B_{carbonate}$ to $\delta^{11}B_{borate}$ compared to culture experiments (Henehan et al., 2013) but showed an offset of ~ -0.4‰.

The regression on smaller size fractions for *T. sacculifer* did not show a significant offset from the calibration from a culture study (Martinez-Boti et al., 2015b), which supports our discussion for *T. sacculifer* in the WEP.

The anonymous reviewer (AR) was concerned that although both *O. universa* and *T. sacculifer* have overlapping calcification depths at sites FC-02a and WP07-a, they record different $\delta^{11}B_{carbonate}$ signatures. We attribute the difference in $\delta^{11}B_{carbonate}$ to the different photosymbiosis characteristics of each species, and discuss this.

We also thank the AR for the note regarding gender assumptions during this process.

Best, Regards,

Revision review by Jesse Farmer

Guillermic et al. have made great progress in revising their manuscript. The introduction, background, materials and methods and results are publishable with very minor revisions. However, the manuscript loses focus and cohesiveness in the discussion, where the text becomes quite difficult to read and understand in its current iteration. I do note that, to my reading, this is an issue only with the writing presentation; the technical aspects of the discussion are sound and the figures are generally excellent and supportive of the text. Still, unfortunately I cannot yet recommend this for publication until the discussion text is improved. I encourage the coauthors to churn through a few rounds of revision to the discussion, with a particular focus on English grammar, and have provided detailed comments below that hopefully aid their work. I look forward to approving the revised manuscript for publication.

L84. Change to "samples from sites that are currently in quasi-equilibrium with the atmosphere" (less ambiguous)

Changed

L125-127. Suggest change to "Furthermore, d11B differences between foraminifera species inhabiting waters of the same pH makes the acquisition of more coretop and culture data essential for applications of the d11B-pH proxy."

Changed

L149. State here which fractionation value you use in this study, e.g., "We use the fractionation of 27.2 ‰ from Klochko et al. (2006) in this study."

Changed

L176-177. Be a little more specific here, e.g., "The extent to which these results apply to the planktonic foraminifera studied here are not known. Nonetheless, pH modulation of the calcifying fluid may influence the d11B of planktonic foraminifera."

Changed

Section 2.5. Please clarify- To what extent are these depth and habitat preferences global? Do foraminifera can show regional deviations in their depth & habitat preferences? Do we know? Many of the referenced studies of foraminifera habitats focus on the tropical/subtropical Atlantic.

"Although the studies listed above showed evidence for species-specific living depth-habitat affinities, recent direct observations showed that environmental conditions (e.g. temperature, light) was locally responsible for the variability in the living depth of certain foraminifera species in the eastern North Atlantic (Rebotim et al., 2017). The same study showed evidence for a correspondence between living depth habitat and indirectly-derived calcification depth, supporting the approach utilized in this study. "

L232. Change "drilled" to "recovered" or "cored". It is only appropriate to say "drilled" when a drilling system was used for core recovery (as is the case for ODP/IODP)
Changed for "cored"

L255. Good points on the response to my comment; please incorporate your response into a sentence in the manuscript here.
L256. "Hydrochloric acid was used to allow complete dissolution of the sample including Fe-Mn oxide and hydroxides if present. No matrix effect resulting from the mix HCl/HF was observed on the $\delta^{11}B$.

L266. Typo, should read "3.5"
Fixed

L269. Change to "70 uL of carbonate sample dissolved in 1N HCl was loaded...". This gets around the ambiguity of acid used in my above comment on L256-257.
Changed

L289. Remove "internal"
Removed

L291. Change "boron isotopes liquid standard" to "boric acid standard"
Changed

L303. Again on the 1N HCl dissolution from L256-257. If I follow your protocol, you have a foram aliquot dissolved in 1N HCl, to which you have added HNO3 and HF for the ICP-MS measurement. Was HCl added to the standards to properly matrix-match? Or was the volume of HCl sufficiently small that you ignored it? Please specify.
The volume of HCl was sufficiently small (<1%) to be ignored.
Line 205 "any matrix effect (Misra et al., 2014b), the remaining HCl (<1%) was negligible."

L339. Suggest "during their growth" in lieu of "along their ontogeny"
Removed instead following reviewer 2 comment. "We applied (based on uncertainties of our measurements) an uncertainty of ±10m for calcification depths > 70 m and an uncertainty of ± 20 m when calcification depths < 70 m."

L339-341. Are these uncertainties truly representative of foram depth migration during their lifespan, or are they more indicative of uncertainty in different measurements of foram depth habitats? Please specify.
"Direct observations of living depths of foraminifera remain limited. However, the depth uncertainties reported here are in line with the uncertainties calculated based on direct observations in the eastern North Atlantic which give a standard error on average living depths ranging from 6-22 m for the same species (Rebotim et al., 2017)."

Section 4.1. This is a fair response to my comment; please incorporate this response into the text by adding a paragraph in Section 3.8. It is certainly fine to use a best guess and reasoned approach to derive CDs, as long as the reader is aware of your approach. (Note- it might also

help to mock up a figure showing the data in Tables S6 and S7).

"Because both methods have their uncertainties (in one case, use of taxon-specific calibrations, and in the other, analytical limitations), both estimates of calcification depth were compared to published values for the basin, and where available, for the same site (Table S6). To select which calcification depth to use for further calculations, we first looked at $CD_1$, $CD_2$ and $CD_3$. If, two CDs were similar we selected that one, if $CD_1$ and $CD_2$ were different we chose literature values ($CD_3$) when available. For some less studied species, like *G. tumida*, *G. menardii* or *P. obliquiloculata*, $CD_3$ was not always available but showed good correspondence with our $CD_2$, moreover due to availability of Mg/Ca-temperature taxon-specific calibrations we preferentially use $CD_2$ for those species."

Section 4.2
To what extent are your foraminifera samples and those from the literature in Fig. 5 from different size fractions? If so, given the noted d11B differences between size fractions, is it worth regressing against literature data from different size fractions? How much of the "large uncertainty given variability in the data" (L399) might reflect size class differences? Please address in the text.

We have added more informations to the section 4.2.1. for *G. ruber*. Results can be in line with a size fraction effect for *G. ruber*, but not for *T. sacculifer*. However, no significant differences are observed yet between the different calibrations due to the limited datasets.

Starting line 400: "Samples were picked from the 250-300 µm fraction, except for the WEP sites where they were picked from the 250-400 µm fraction. Weight per shell averaged 11 ± 4 µg (n=4, SD) although the weight was not measured on the same sub-sample analyzed for $\delta^{11}B$ and trace elements or at the WEP sites. In comparison to literature, the size fraction used for this study was smaller: Foster et al. (2008) used the 300-355µm fraction, Henehan et al. (2013) utilized multiple size fractions (250-300, 250-355, 300-355, 355-400 and 400-455 µm) and Raitzsch et al. (2018) used the 315-355 µm fraction.

Our results for *G. ruber* (Fig. 5) are in close agreement with published data from other core-tops, sediment traps, tows, and culture experiments for $\delta^{11}B_{borate}$>19 ‰ (Foster et al., 2008, Henehan et al., 2013, Raitzsch et al., 2018). However, the two datapoints from $\delta^{11}B_{borate}$<19 ‰ are lower compared to previous studies. Elevated $\delta^{11}B_{carbonate}$ values relative to $\delta^{11}B_{borate}$ has been explained by the high photosynthetic activity (Hönisch et al., 2003; Zeebe et al., 2003). Three calibrations have been derived (Table 3). Linear regression on our data alone yields a slope of 1.12 (±1.67). While this regression is not significantly different from a 1:1 line, the uncertainty term are significant given limited data in our study. Therefore, the sensitivity of $\delta^{11}B_{carbonate}$ to $\delta^{11}B_{borate}$ of our linear regression is not statistically different from 1, the uncertainty on this regression is important due to our small dataset, thus not inconsistent with the low sensitivity trend of the culture experiments from Sanyal et al., (2001) or Henehan et al., (2013). The second calibration made compiling all data from literature shows a sensitivity similar (e.g. 0.46 (±0.34) to the one recently published by Raitzsch et al., (2018) (e.g. 0.45 (±0.16), Table 3). The third linear regression made only on data from the 250-400 µm fraction from our study and from the 250-300 µm from Henehan et al. (2013) yields a slope of 0.58 (±0.91) similar to culture experiments from Henehan et al., (2013) (e.g. 0.6 (±0.16), Table 3). This third calibration is offset by ~ -0.4 ‰ (p>0.05) compared to culture calibration from Henehan et al. (2013). The variability in our weight per shell based data from Henehan et al., (2013) can potentially imply a deviation down to 1‰ relative to its calibration line, which can be in line with the maximum deviation observed in our data (~1.2 ‰) and not inconsistent with a size effect explaining the offset in our calibration.

Line 431: "It is also noticeable that *T. sacculifer* (w/o sacc) samples from the WEP have a $\delta^{11}B_{carbonate}$ close to expected $\delta^{11}B_{borate}$ and are significantly lower compared to the combined *T. sacculifer* of other sites (p=0.01, unpaired t-test). When doing the regression using data from the 250-400 µm fraction, our results are not significantly different from the regression through data that combine all size fractions (Fig. 5).

Also, suggest changing " fall (above/below) the 1:1 line" to "exhibit (higher/lower) d11B compared to expected d11B_borate at their collection location" throughout the text. It is less confusing and more precise. (see L396, 400)
Changed throughout the text

Section 4.2.1 G. ruber. This section is confusing as currently written. Some rephrasing ideas:
• L383: Rephrase to "However, our two datapoints from d11B_borate < 19 ‰ are lower compared to previous studies." "Lighter d11B" is not correct; use either "lighter B isotopic composition" or "lower d11B".
Changed

• L383-384: Regarding the comment from M. Henehan and response, are your samples from different size fractions than
I think I have given more informations in this section.

• L384-385: "Whilst..." Delete this unnecessary sentence. **Deleted**
• L385-386: "The positive offset from the 1:1 curve..." change to say elevated d11B foram values relative to d11B_borate. **Changed**
• If you want to discuss the properties of the calibration curves, you should state the curves in the text for the reader. You should say something like you do for T. sacculifer on L396-397: "Linear regression on our data alone yields a regression of ______. While this regression is not significantly different from a 1:1 line, the uncertainty terms are significant given limited data in our study. Therefore, our data are not inconsistent with the low sensitivity trend.... (append the rest of the paragraph as written)".

Line 410: "Three calibrations have been derived (Table 3). Linear regression on our data alone yields a slope of 1.12 (±1.67). While this regression is not significantly different from a 1:1 line, the uncertainty term are significant given limited data in our study. Therefore, the sensitivity of $\delta^{11}B_{carbonate}$ to $\delta^{11}B_{borate}$ of our linear regression is not statistically different from 1, the uncertainty on this regression is important due to our small dataset, thus not inconsistent with the low sensitivity trend of the culture experiments from Sanyal et al., (2001) or Henehan et al., (2013). The second calibration made compiling all data from literature shows a sensitivity similar (e.g. 0.46 (±0.34) to the one recently published by Raitzsch et al., (2018) (e.g. 0.45 (±0.16), Table 3). The third linear regression made only on data from the 250-400 µm fraction from our study and from the 250-300 µm from Henehan et al. (2013) yields a slope of 0.58 (±0.91) similar to culture experiments from Henehan et al., (2013) (e.g. 0.6 (±0.16), Table 3)."

L400. Remove "below the 1:1 line" as no data are significantly below this line. "close to" is fine. Can you indicate the WEP samples on this plot with a star or other indication?
**Changed, I have highlighted the WEP samples in Fig. 5 for *T. sacculifer*.**

L408-409. Rephrase to ", and is not significantly different from (p >0.05) the O. universa calibration previously reported by Henehan et al. (2016) (0.95 ± 0.17)."
**Changed**

L413. "For O. universa and all deep-dwelling species,"
**Changed**

L417. Change "may remain" to "remains"
**Changed**

L430. Typo T. sacculifer (sacc)
**Fixed**

L432-433. What do you mean by interspecific B/Ca ratios? Please elaborate. I do not think it comes as any surprise that B/Ca ratios are different in different foraminifera species.
**This has been changed to: "This study supports species-specific B/Ca ratios as previously published (Yu et al., 2007; Tripati et al., 2009, 2011; Allen and Hönisch, 2012; Henehan et al., 2016)."**

L441-444. Confusing. Please split into two sentences, with one about core site depth and one about calcification depth. Also please note that you see a weak decrease in B/Ca with increasing calcification depth, although it is significant (p<0.05).
**"When comparing data from all sites together, a weak decrease in B/Ca with increasing calcification depth is observed ($R^2$=0.11, p<0.05, Fig. S4). A correlation also exists between B/Ca and the water depths of the cores (not significant, Fig. S4)."**

L454. Typo (w/o sacc)
**Fixed**

L487-488. Instead of saying seasonality is not important, rephrase to "seasonality is of relatively minor impact on the carbonate system parameters at the sites we examined."
**Changed**

L499-503. Specify symbiont photosynthesis
**Added**

L502-503. lower/lowest
**Changed: "Dinoflagellate-bearing foraminifera (*G. ruber*, *T. sacculifer* and *O. universa*) tend to have a higher symbiont density and photosynthesis activity while *P. obliquiloculata*, *G. menardii* and *N. dutertrei* have lower symbiont density and *P. obliquiloculata*, *N. dutertrei* lowest photosynthetic activity (Takagi et al., 2019)."**

L508-549. This is difficult to understand as written and needs revising, otherwise these points will be completely lost by the reader.

L508-513. Present items in logical order: First what you observe (low d11B of deep-dwelling species relative to d11B borate), then context (lower symbiont density and photosynthetic activity in these forams), and combine this into an interpretation (lower symbiont activity leads to lower microenvironment pH and may explain the low d11B of these taxa).

We reformatted this section according to your suggestions.

L513. "and" instead of "et"

Changed

L514 and throughout. Do not use "they" or any pronouns, as it is not clear to what you are referring. Use the noun itself. Here, "they" = "symbionts"

Changed

L518. Be specific. "A deeper depth habitat will reduce the light intensity the symbionts receive, and as a consequence may lower symbiont photosynthetic activity, possibly reducing pH in the microenvironment surrounding the foraminifera".

Changed

L521. What does "basically support" mean? Either the trend supports the fact or it does not.

Fixed

L524. Start a new paragraph here. "To test if the d11B signature was inferred to a light driven"- what does this mean?

I changed this section according to your previous comment.

L524-525. change to "we have independently calculated foraminifera (calcification?) dpeth based on various light insolation culture experiments and the microenvironment ΔpH derived from our data".

Changed

L535. Change to Microenvironment ΔpH

Changed

L537-541. Please rephrase these sentences. I think this may be a key point, but I cannot follow it as currently written.

We have edited this paragraph.

L542. Change to "G. menardii and G. tumida are similar..."

Changed

L547-549. Need to add some commas here to make this understandable.

We changed for: "We can also note that *P. obliquiloculata,* which has the lowest symbiont density and photosynthetic activity (Takagi et al., 2019), has the lowest microenvironment pH

compared to other deeper-dweller species, supporting this respiration driven microenvironment"

Changed

Edited

Changed

Edited: "Our $\delta^{11}B_{carbonate}$ date and their sensitivity to $\delta^{11}B_{borate}$ for *O. universa* support previous data from Henehan et al., (2016)."

I removed this part, as I have developed it in the other sections 4.2.1 and section 5.4.

"Results for *G. ruber* are the most scattered, potentially due to difference in test sizes (Henehan et al., 2013), or depth habitat, althought we can not exclude undocumented diagenetic effects. Results reaffirm the importance of working with narrow size fractions (Henehan et al., 2013), the utilization of calibrations derived from the same size fraction or use of offsets to take into account this size fraction effect, and the importance of core-top studies before paleo-application."

"We also find that for two species, the boron proxy is a relatively straightforward recorder of ambient pH, with sensitivities close to unity for *O. universa and N. dutertrei*."

Checked

Changed

Report #2
Submitted on 02 Dec 2019
Anonymous Referee #2

This is my second review of this MS. As it stands, the edits are generally ok, but in some cases the authors chose to only reply to edits in the response and not always insert their response into the MS. Sometimes this was ok, but if a reviewer requests an explanation, the response should generally be incorporated into the manuscript. Reviewer 1 should also review the edits as some of Dr. Farmer's suggestions were also not implemented into the revised MS. I have specific comments below. The new section on symbiosis and the findings of Tagaki was very poorly written and should be edited prior to final publication.

We have reviewed the edits and incorporated them, including the explanation for the depth habitat and the cleaning. We also have rewritten section 5.2.

As an aside: please do not assume that anonymous reviewers are CIS gendered males. The opening response to my review was "We wish to thank this reviewer for HIS thorough review of our manuscript and HIS helpful comments". Instead: We thank the reviewer for helpful comments and suggestions.

We thank the reviewer for pointing this out.

There is no evidence that foraminifers change their symbiont assemblages to ones more similar to what are found in deeper dwelling species. In addition, the paragraph beginning on line 195 where differences in the boron isotope data between T. sacculifer, G. ruber, and O. universa is explored states that O. universa exhibits lower d11B because it lives deeper in the water column, but the depth habitats chosen for these species IN THIS STUDY overlap or are at the same depth according to table S7.

Line 176: Change: "We acknowledge this is speculative as it is based upon benthic foraminifer experiments" to "We acknowledge this process may not be the same for planktic species as these findings were based upon benthic foraminifer experiments"

This has been changed from "The extent to which these results apply to the planktonic foraminifera studied here are not known. Nonetheless, pH modulation of the calcifying fluid may influence the $\delta^{11}B$ of planktonic foraminifera."

Section 3.3: the reviewers responded to my comment but did not add any additional text to the MS. The reason the full reductive cleaning protocol was used should be included in this paragraph so that other readers are aware of the study by Misra 2014 and the purpose of the full cleaning protocol.

This states: "Samples were then cleaned using full reductive and oxidative cleaning (Boyle and Keigwin, 1985; Barker et al., 2003). We utilized the reductive cleaning because some of the sites where not previously studied and previous comparison have shown no effect on B/Ca (Misra et al., 2014b), nevertheless, Fe-Mn oxide and hydroxides can result in non-negligible content of Mg and B contamination. Overall, the samples did not present high Mn concentration. Reductive cleaning leads to a decrease in Mg/Ca which would result in deeper CDs, which is not the case when comparing with CD1 and CD3, we then no longer assume this decrease problematic for the purpose of this study."

Line 339: I suggest removing the clause "As foraminifera can migrate in the water column along their ontogeny" because it is misleading. Foraminifers can occupy a deeper depth habitat at the end of their ontogeny, but as written it still seems to imply that foraminifers migrate up and down, which they likely do not. Removing this clause bears no impact on the rest of the sentence.
Removed

Line 456: Again: I think this should be more carefully written. The use of the word "migrate" with foraminifer depth habitats usually implies that they move up and down in the water column, which they do not. This should be reworded to state that at the end of their life cycle they often transition to deeper waters prior to gametogenesis.
Changed for: "We note that calculation of absolute calcification depths can be challenging in some cases as many species often transition to deeper waters at the end of their life cycle prior to gametogenesis"

Line 495: G. ruber, T. sacculifer and O. universa do not have chrysophyte algal symbionts, only dinoflagellates. There is no mention of chrysophytes in the Anderson and Be paper nor in Spero 1987. N. dutertrei has pelagophyte symbionts not chrysophytes, confirmed using genetics, see Bird et al., 2018.
Thank you for those inputs. We have added the pelagophyte symbionts for *N. dutertrei.*

Paragraph beginning line 500 is poorly worded.
The substance does not change, but we have edited it.

Line 509: have microenvironments with lower "pH" than ambient seawater (insert pH)
Changed
Line 513: 'et'?
Changed
Line 514: insert 'a' between also and function.
Added
Line 515: Should "for the purpose of this study" be a new sentence?
Split in two sentences
Line 537: Tagazaki should be Tagaki.
Changed
Line 540: specie is spelled incorrectly
Changed

Paragraph beginning on line 540: O. universa in this study occupy similar depths to the G. ruber and G. sacculifer. Thus, I do not agree with the discussion here.
That is actually where Tagaki's study is relevant, as it shows that each species has their own characteristics in terms of photosymbiosis. What we found interesting is that potentially, *T. sacculifer*, which seems to have a higher potential for photosynthesis, might also be more sensitive to changes of insolation depending of its habitat in the water column. Also, its photosynthetic activity might be more effective at depth due to its higher symbiont density, reflected in higher $\delta^{11}B$carbonate than *O. universa* which has a lower potential for photosynthesis **for the same depth.**

Line 560: "*T. sacculifer* has the potential to support more photosynthesis due to its higher symbiont density. Higher photosynthetic activity is observed compared to other species potentially supporting higher symbiont/host interactions (Tagaki et al., 2019). Those results could be in line with a greater sensitivity of *T. sacculifer's* photosynthetic activity with changes in insolation/water depth."
Line 577: "The low $\delta^{11}B_{carbonate}$ of *O. universa* compared to *T. sacculifer* for the similar calcification depth at few sites (e.g. FC-02a, WP07-a) might reflect difference in photosynthetic potential between the two species, Tagaki et al. (2019) showed a lower photosynthetic potential for *O. universa* compared to *T. sacculifer*."

Section 5.4 is very confusing. It seems haphazardly put together and some sentences are poorly structured. Paragraph 580 could be deleted.
We have edited this part following suggestions from Jesse Farmer

Report #3 Henehan

The manuscript is greatly improved from its previous incarnation. Although it is for the previous reviewers to judge whether their comments are adequately dealt with, my sense is that they seem to be largely adequately discussed.

I am personally still a bit unconvinced in the point of plotting a calibration line through data that come from different size fractions for G. ruber and T. sacculifer, when there is clearly a known size fraction effect which is muddling the pH (/borate) signal and potentially influencing the slope. I guess particularly also in this study using such wide size fractions (perhaps by necessity due to sample limitations) means that there is always the possibility that the size distribution could have varied within this range and introduced inter-site differences that change the slope. The authors now provide some shell weight data to inform a little as to the possibility of there being inter-site differences in the sampled shell size fraction – I presume because there are no photos of the samples to measure the actual size distribution. This is at least something, but just telling us that the average shell weight varied is only of limited use- why not give us this data in the tables, so we can see if those boron samples that diverge most strongly from the existing calibration line of ruber/sacculifer were indeed on average the smallest test sizes? It would be helpful.

Unfortunately, no weight per shell determinations was done on the WEP samples.  Also, the weight per shell we discuss in the manuscript are from other sub-samples picked for oxygen and carbon isotopes only.
We have derived the calibrations from the smaller size fractions, results for *G. ruber* can be in line with a size effect on the $\delta^{11}B$carbonate, same sensitivity to  $\delta^{11}B_{borate}$ but offset of ~0.4‰ from your culture calibration.

There are also still quite a few oddities in sentence structure, word choice, spelling/typos which I will outline below.

Otherwise, with these edits/additions, I have no objections the paper being published. Congratulations to the authors on some nice work.

Line 94: Anagnostou et al. (2016) also did this for core-top and Eocene samples.
Added

Line 144: B(OH)3 and B(OH)4-, as used here, are the more common notations for these aqueous species.. I would favour using these. But then for whatever reason in Fig. 2 and its caption (but I think nowhere else?) you have called them H4B(OH)3 etc. I think you should be consistent so the unfamiliar reader can relate what you are talking about better. Why not stick to what you have here to be more consistent with the literature?
Changed

Line 157: as far as I am aware Noireaux et al 2015 did not look at taxonomic differences in forams- rather inorganic carbonate polymorphs. Either rephrase the sentence or delete the reference.
We removed it, this is a typo from previous version.

Line 196: within the parentheses, insert comma after the subscript borate, followed by 'hereafter'
added

Line 204: Note that the culture slopes of T. sacculifer and O. universa are also within error of 1, when one propagates the error on each datapoint (which for universa in particular is rather large).
"More core-top and culture calibrations are needed to refine those slopes and fully understand why different slopesif significant differences are observed, which is part of the motivation for this study"

Line 273: I may be being ignorant, but I have never heard of H or X sample cones, only skimmer cones. Is this correct here?
Thank you for picking on that, you are right, skimmer are X or H but sample are normal or jet. We utilized "normal" sample cones.

Line 282: delete 'the' before 11B.
Removed

Lines 285, 293, possibly elsewhere: Inconsistent nomenclature of JCp-1 coral standard. How I just wrote it is the official notation.
Changed

Line 300: Subscript needed for 3 on nitric acid.
Changed

Line 329: 'prerequisite' would be a better word than 'postulate'.
Changed

Line 331: apostrophe after the 's' of species.
Added

Line 426: typo in obliquiloculata.
Changed

Line 427: What do the authors mean by 'This study supports interspecific B/Ca ratios'.. ? Rewrite.
" B/Ca data are species-specific and consistent with previous work (e.g., compiled in Henehan et al., 2016) with ratios higher for *G. ruber* > *T. sacculifer* (sacc) > *T. sacculifer* (w/o sacc) > *P. obliquiloculata* > *O. universa* > > *G. menardii* > *N. dutertrei* > *G. tumida* > *G. inflata* > *N. pachyderma* > *G. bulloides* (Fig. 7)."

Line 451: correct names are 'Kemle-von Mücke and Oberhänsli'
Changed

Line 486: Is this statement really true? See Shaked and de Vargas (2008), doi:10.3354/meps325059

I removed this statement and added the reference

Line 517-518: Sentence poorly written. Not exactly sure what it is the authors want to say.

We have reedited this part

Line 532: missing s on species.

Added

Line 537: 'deep-dwelling species' or 'deeper-dwelling species', not 'deep dweller species'

Changed thorough the text

Line 546: Joji Uchikawa, not Ushikawa.

Changed thorough the text

Lines 550-551: Not sure it's wise to pool all different size fractions like this- what does this truly tell you about pH?

We made regressions based on the different size fractions.

Line 552: missing 'and' before 'the observation'? Also 'is', not 'are'.

Changed

Line 567-568: "the main interest with utilizing boron-based proxies relates to the…" poorly written sentence. "The main aim of utilizing" or "The primary applications of boron-based proxies are"?

Changed for: "The main aim of utilizing boron-based proxies relates to the reconstruction of past oceanic conditions"

Line 567-575: Odd paragraph separation, and first line of second paragraph in line 572 isn't particularly self-contained. In line 574: 'which is also supporting of' should be 'which supports'.

Changed

Lines 576-578: I would suggest that Chalk et al. (2017)'s accurate downcore data would suggest there isn't necessarily need for a calibration from the same location or setting? Is it fair to make such a bold statement, and advocate for every downcore reconstruction from now on to have a calibration point from the same setting, based on your sacculifer data alone (that could in theory be different in some places just because the size distribution isn't exactly the same in all sample sites)?

Yes, only the WEP presents those low d11B in our data, I think it is still best explained by their calcification depths. At this stage, for reconstruction in the WEP, correction / calibration will have to be applied. Nevertheless, this is reassuring to observe that the sensitivity of the calibrations for G. ruber is the not significantly different between size fractions, which would make the use of an offset ok for the reconstruction.

Lines 579-580: What correction do you mean? And why? Correction relative to what, the culture calibration?

"Results for *G. ruber* are the most scattered, potentially due to difference in test sizes (Henehan et al., 2013), or depth habitat, althought we can not exclude undocumented diagenetic effects. Results reaffirm the importance of working with narrow size fractions (Henehan et al., 2013), the utilization of calibrations derived from the same size fraction or use of offsets to take into account this size fraction effect, and the importance of core-top studies before paleo-application."

Line 583: what is meant by 'our sample add a weight/shell'?
Edited

Line 584: Suggest "Greater divergence" rather than "The higher divergence"
Changed

Line 588: insert 'isotope-pH' after boron.
Added

Line 592 (and elsewhere): in situ is not hyphenated.
Addressed

Lines 592-594: This sentence could be rewritten to explain a bit more clearly what you did.
This section has been reedited

Line 600: 'Sparsest' seems an odd choice of word- implies not only that there is more scatter, but there is also not as much data. 'most scattered' might be better?
Changed

Line 606: 'can impact' rather than 'impacts', as for example your deep dwellers are not very affected.
Changed

Line 614: insert 'similarly' before respiration
Added

Line 617: "those calibrations" – which calibrations are you referring to?
"the calibrations published to date"

Line 618: 'taxon-specific', not 'taxa-specific', I would think here?
Changed

Figure 5: Include the size fraction in each ruber and sacculifer dataset's legend entry, not just one.. and why are the ruber culture data not plotted?
Added and plotted the culture data

Figure captions: In general the figure and table captions are a little spartan, and could be a bit more in-depth and informative. Examples, for instance:

Fig. 4 caption, 'Pre-industrial data' is vague.. hydrographic data (also 'for the sites', rather than 'of the sites')? Are these actual measurements or interpolations? What are the dotted lines? Why are the temperatures seasonally defined but nothing else?
We utilized the GLODAP database that does not resolve seasonal variations. We have added "Dotted lines are the calculated uncertainties based errors on TA and DIC from the GLODAP database."

Fig.5 caption: d11Bborate was, not were. Are the measurements from core-tops, tows, cultures, etc? Are these just the MC-ICPMS data, if so state, so it's clear why there's no NTIMS measurements in here? How are the error bars defined, are they one sigma or two?
" Boron isotopic measurements of mixed-layer foraminifera plotted against the $\delta11B$borate. $\delta11B$borate was characterized by determination of the calcification depth of the foraminifera utilizing data presented in Fig. 4, A) G. ruber, B) T. sacculifer, C) O. universa. Mono-specific calibrations (Table 3) and error bars on $\delta11B$borate were derived utilizing the wild bootstrap code from Henehan et al. (2016), errors on the $\delta11B$carbonate for this study are reported as $2\sigma$ of measured AE121 standards during the session of the sample. Calibrations were also derived on the 250-400 size fraction for G. ruber and T. sacculifer (black dashed lines). Data reported on those graphs have been measured with an MC-ICP-MS."

Fig. 6: 'plotted', not 'plot'.
Changed

Fig. 8: watch your superscripts. Does the light attenuation coefficient have a unit? State how the calcification depth marked by the grey band was derived.
So, no units were reported in Rink et al., 1998, but we believe it may be $m^{-1}$

Fig. 10: say a little bit more about what the 'reconstructed' values come from and how they were calculated.
" **Figure 9:** Water depth pH profiles reconstructed at every site applying the mono-specific calibrations derived from our results (Table 3). Figure is showing measured $\delta^{11}B_{calcite}$, $\delta^{11}B_{borate}$ calculated according to different calibrations (see Table 3 and text), calculated pH based on $\delta^{11}B$ (pH$_{\delta11B}$) and pCO$_2$ calculated from pH$_{\delta11B}$ and alkalinity.
**Figure 10:** Evaluation of the reconstructed parameters, $\delta^{11}B_{borate}$, pH and pCO$_2$ versus *in situ* parameter calculated in Fig. 9 (based on $\delta^{11}B$ and alkalinity). The recalculated parameters are consistent with *in situ* data, except for *G. ruber*, this variability might be explained by the different test sizes within our size fractions."

Table 2: Is it necessary to have a column for the cleaning method when it's the same for every sample?
We have removed this column.

Quick question- Site A14 (to a lesser extent also FC-02a) has a really weird vertical pH gradient.. any idea what causes this?
We are not sure how to explain the A14 profile.

**Résultats de la comparaison**

| Ancien fichier : | par rapport à | Nouveau fichier : |
|---|---|---|
| **Maxence Guillermic coretop manuscript - 03092020.pdf** | | **Maxence Guillermic coretop manuscript - 11122019.pdf** |
| **31 pages (736 KB)** | | **30 pages (742 KB)** |
| 3/9/2020 9:14:32 AM | | 2/3/2020 4:10:37 PM |

**Total des modifications**

**1187**

Comparaison du texte seulement

**Contenu**

remplacements insertions suppressions

**Styles et annotations**

style annotations

Consulter la 1re modification (page 1)

[revised manuscript text omitted]

In planktonic foraminifera, algal symbiosis is the more common symbiotic relationship. For most of planktonic foraminifera, the host presents only one species of symbionts (Gast and Caron, 2001). The family

Globigerinidae, including *G. ruber*, *T. sacculifer* and *O. universa*, commonly have dinoflagellates or chrysophyte algal symbionts (Anderson and Be, 1976; Spero, 1987). The families Pulleniatinidae, Globorotaliidae, including

*N. dutertrei*, *P. obliquiloculata, G. menardii* and *G. tumida*, have chrysophyte algal symbionts (Gastrich, 1988).

The relationship between the symbionts and the host is complex by nature. Nevertheless, this symbiotic relationship provides energy (Hallock, 1981b) and promotes calcification of the foraminifera (Duguay, 1983; Erez et al., 1983) by providing the inorganic carbon to the host (Jorgensen et al., 1985). Also, for *T. sacculifer* and *O.*

*universa* photosynthesis increases with higher insolation (Jorgensen et al., 1985; Rink et al., 1998).

Dinoflagellate-bearing foraminifera (*G. ruber*, *T. sacculifer* and *O. universa*) tend to have a higher symbiont density and photosynthesis activity while *P. obliquiloculata*, *G. menardii* and *N. dutertrei* have lowered symbiont density and *P. obliquiloculata*, *N. dutertrei* lower photosynthetic activity (Takagi et al., 2019). *P.*

*obliquiloculata* showed the minimum symbiont density and photosynthetic activity (Takagi et al., 2019).

It is now accepted that the foraminifera $\delta^{11}B$ signature comes from the microenvironment pH (Jorgensen et al., 1985; Rink et al., 1998; Köhler-Rink and Kühl, 2000, Hönisch et al., 2003; Zeebe el el., 2003). Foraminifera with high photosynthetic activity and symbiont density like *G. ruber* and *T. sacculifer* present a pH of microenvironment higher than ambient seawater, $\delta^{11}B$ higher than 1:1 line (Foster et al., 2008, Henehan et al., 2013, Raitzsch et al., 2018). The opposite can also be true, from our study, species with lower photosynthetic activity and lower symbiont density present microenvironments lower than ambient seawater, $\delta^{11}B$ lower than 1:1 line (Martinez-Boti et al., 2015b; Henehan et al., 2016), this is the case in our data for *N. dutertrei*, *G. menardii* and *P. obliquiloculata* and likely *G. tumida*. Nevertheless, the low $\delta^{11}B$ of *O. universa* and *T. sacculifer (w/o sacc)* from the WEP are difficult to reconcile with a high photosynthetic activity compared to *T. sacculifer* et *G. ruber.*

The photosynthetic activity is also function of the light level they received which is, in the natural system, dependent of their depth in the water column, for the purpose of this study we will not consider turbidity which also influences the light penetration in the water column. In this case, the photosynthetically active foraminifera living close to the surface should see their microenvironment pH (thus $\delta^{11}B$) more sensitive to water depth changes. A deeper depth habitat will change the light intensity they received and as a consequence may lower their photosynthetic activity reducing their microenvironment pH. This thought is supported by the significant trend observed between our $\Delta^{11}B$ and the calcification depth for *G. ruber* and *T. sacculifer* of our sites (Fig. S2). This trend basically supports the fact that the microenvironment pH decrease with calcification depth. We observe a decrease of $\delta^{11}B$ in the WEP for *T. sacculifer (w/o sacc)*, significantly different from the other sites (p<0.05). The $\Delta^{11}B$ of *G. ruber*, *T. sacculifer* (w/o sacc and sacc) is also significantly lower in the WEP compared to the other sites (p<0.05). To test if the $\delta^{11}B$ signature was inferred to a light driven, we have been able to independently calculate the depth of the foraminifera based on various light insolation culture experiments (Jorgensen et al., 1985) and the Δmicroenvironment pH derived from our data (Fig. 8A and B). This exercise verified that this low $\delta^{11}B$ can be explained by the reduced light environment due to a deeper depth habitat in the WEP (Fig. 8B). Also, *T. sacculifer* has the potential to support more photosynthesis due to its higher symbiont density. Higher photosynthetic activity is observed compared to other species potentially supporting higher symbiont/host interactions. Those results could be in line with a greater sensitivity of *T. sacculifer* photosynthetic activity with changes in insolation/water depth. It can also be noted that this species presents the largest variations in symbiont density versus its test size. When applied to the other species *O. universa* data suggest a microenvironment pH 0.10 to 0.20 lower than ambient seawater pH which would be in line with species living deeper than 50m (light compensation point (Ec), Rink et al., 1998) also consistent with our calcification depth reconstructions. Δmicroenvironment pH is higher in *T. sacculifer* > *G. ruber* > *T. sacculifer* (w/o sacc - WEP) > *O. universa*, *N. dutertrei*, *G. menardii*, *G. tumida* > *P. obliquiloculata* in line with photosymbiosis findings from Tagazaki et al., (2019). Also, the higher $\delta^{11}B$ data from the African upwelling published by Raitzsch et al., (2018) for *G. ruber* and *O. universa* might reflect the higher microenvironment pH due to a shallower depth habitat. This could highlight a potential issue with calibration when applied to sites with different oceanic regimes as the $\delta^{11}B$ specie-specific calibrations could be also location-specific for the mixed dweller species.

Microenvironment pH results for *N. dutertrei, G. menardii, G. tumida,* are similar to *O. universa* and suggest a threshold for respiration driven $\delta^{11}B$ signature. This threshold can be driven by a change of photosynthetic activity due to lower light intensity at deeper depth and/or a change in the symbiont assemblage with non-dinoflagellate symbionts at deeper depth. We can explain this threshold because deep dweller species do not experience important changes of insolation at those depths so their microenvironments should be respiration driven and relatively stable. We can also note that *P. obliquiloculata* which has the lowest symbiont density and photosynthetic activity has the lowest microenvironment pH compared to other deeper dweller species supporting this respiration driven microenvironment.

**5.3 $\delta^{11}$B sensitivity to $\delta^{11}$B$_{borate}$ and relationship with B/Ca signatures**

$\delta^{11}$B$_{carbonate}$ and B/Ca data have shown to be sensitive to precipitation rate with at higher precipitation rate increasing $\delta^{11}$B$_{carbonate}$ (Farmer et al., 2019) and B/Ca (Farmer et al., 2019; Gabitov et al., 2014; Kaczmarek et al., 2016; Mavromatis et al., 2015; Ushikawa et al., 2015). A recent study from Farmer et al, (2019) has proposed that in foraminifera at higher precipitation rates, more borate ion is incorporated into the carbonate mineral, while at lower precipitation rates, more boric acid is incorporated. They also suggest this may explain low sensitivities of culture experiments.

When combining all literature data, *T. sacculifer* and *G. ruber* have sensitivities of $\delta^{11}$B$_{carbonate}$ to $\delta^{11}$B$_{borate}$ of 0.83 ± 0.48 and 0.46 ± 0.34 respectively in line with previous literature and paleo-CO$_2$ reconstructions. Also, if we only take into account our data, the observation that the sensitivity of $\delta^{11}$B$_{carbonate}$ to $\delta^{11}$B$_{borate}$ are not statistically different from unity for most of the species investigated we can speculate that for these taxa, changes in precipitation rate and contributions of boric acid are not likely to be important. If considering only the data from this study, *G. ruber* (1.12 ± 1.67) and *T. sacculifer* (1.38± 1.35) present higher sensitivities of $\delta^{11}$B$_{carbonate}$ to $\delta^{11}$B$_{borate}$. We can then again speculate that the observed high values for $\delta^{11}$B$_{carbonate}$ at high seawater pH can be due to higher precipitation rates. We note this could also be consistent with the higher sensitivity of B/Ca signatures in these two surface dwelling species to ambient [B(OH)$_4^-$]/[HCO$_3^-$] relative to deeper dwelling species. Those interspecific differences still remain to be explained, however, part of this variability is likely due to changes in the carbonate chemistry of the microenvironment resulting in changing competition between borate and bicarbonate. A caveat is that we can not exclude specific biological processes, and that in taxa with a non respiration-driven microenvironment, changes in day/night calcification ratios also impacting observed values. As indicated by Farmer et al., (2019), studies of calcite precipitation rates in foraminifera may help to improve our understanding of the fundamental basis of boron-based proxies.

**5.4 Evaluation of species for pH reconstructions and water depth pH reconstructions**

This data set allows us to reassess the utility of boron-based proxies for the carbonate system. The main interest with utilizing boron-based proxies relates to the reconstruction of past oceanic conditions - specifically pH and pCO$_2$. Mixed-layer species (eg. *G. ruber* and *T. sacculifer)* are potential archives for atmospheric CO$_2$ reconstructions. Other species can shed light on other aspects of the carbon cycle including the physical and biological carbon pumps.

There are a few main inferences we can make. When compiled with data from the literature, sensitivities of $\delta^{11}$B$_{carbonate}$ to $\delta^{11}$B$_{borate}$ for *G. ruber* and *T. sacculifer* are similar to previous studies (Martinez-Boti et al., 2015b; Raitzsch et al., 2018) which is also supporting of previous paleo-reconstructions. Our data also support the observations of Henehan et al., (2016) for *O. universa*.

In order to derive accurate reconstructions of past ambient pH and pCO$_2$, accurate species-specific calibrations need to be used that are constrained by core-tops or samples from similar types of settings (Fig. 9, 10, S6). Lighter $\delta^{11}$B signatures in *T. sacculifer* (w/o sacc) are observed in the WEP, which may be explained by the deeper depth habitat for these taxa, where lower light levels might reduce symbiont photosynthetic activity. Also, correction will be needed for *T. sacculifer* (w/o sacc) in the WEP. When applying the calibrations n°2 and 4 to *T.*
*sacculifer* and *G. ruber* (compilation of all data, Table 3) our data show more variability, especially for *G. ruber*
which lead to the larger mismatch compared to *in-situ* parameters. Henehan et al., (2013) reported a lighter $\delta^{11}B$
with smaller test size, our sample add a weight/shell of $11 \pm 4$ µg (n=4, SD) which, despite a narrow range, could

[revised manuscript text omitted]

---

## Author Response (AR3)

**Response to reviewer and editor**

We want to thank Jesse Farmer for his thoughtful review. We have addressed all of the comments, including removing pronouns, adding the text that was requested, adding missing references, defining terms and rewording some sentences. Below is a point-by-point response.

We hope this manuscript is now easier to follow for the reader and suitable for publication.

All the Best, On behalf of the authors

Review of Guillermic et al., "Seawater pH reconstruction using boron isotopes in multiple planktonic foraminifera species with different depth habitats and their potential to constrain pH and pCO2 gradients", revised for Biogeosciences, by Jesse Farmer

This is now my nth review of the manuscript by Guillermic and colleagues, and I am pleased to report that the authors have made excellent progress in both the text and figure presentation. At this point, while I still have a laundry list of recommended changes, specifications and clarifications, I am comfortable with publication of this manuscript in principle. Nice work.

General comments:
I strongly recommend that the authors address the following issues to improve readability and precision of the manuscript. Note particularly three items:
1. Terms such as CD1 and d11B must be defined at their first use.
2. The readability of this manuscript suffers from a lack of specificity in multiple places. Pronouns and vague descriptions should be removed and replaced with specific terms at every possible instance.
3. Responses to previous reviewer comments are often in the location of the comment, and not the location in the text where the response would be most helpful to the reader. As a result, there is unnecessary overlap between Materials and Methods, Results, and Discussion that should be cleaned up for clarity.

Specific comments:
Title
A bit bulky; what about cutting the end and making this "Seawater pH reconstruction using boron isotopes in multiple planktonic foraminifera species with different depth habitats"?
Although it is lengthy, this is a major part of the discussion, so we kept it as is.

Introduction
L71. Change "utilizing" to "using".
Changed
L74. Insert "available": "one of the more robust available tools".
Added
L76. Change "ambient" to "solution"
Changed
L72 and thereafter: You need to define terms at their first use. For example, on L73 "the boron isotopic composition (expressed as d11B)…"
Related comment on L137-140: Here you define d11B in Equation 1. However, by this point you have already used d11B in the text without the definition. You should address this by either:
1. Move the equation and text on L137-140 up to L72, so that all instances of d11B occur after this equation. Or,
2. Keep the text and equation in place, use "boron isotope ratio" instead of d11B for all instances in the Introduction, and then use d11B after L140.
We haven't moved the equation, thus replaced "d11B" by "boron isotope ratio" until the equation. We have defined the boron isotopic composition of foraminifera and borate in the introduction, lines 73 and 153.
L84-86. You have not defined d11B_carbonate yet, and either way it is not specific enough for this sentence. Rephrase to "Values of pCO2 reconstructed from planktonic foraminifera d11B are indistinguishable…"
Line 84: "Values of $p$CO$_2$ reconstructed from planktonic foraminifera boron isotopes are analytically indistinguishable from ice core CO$_2$ records (Hönisch et al., 2004, 2009; Foster et al., 2008; Henehan et al., 2013; Chalk et al., 2017)."
L87. Remove "therefore"; this does not logically follow the previous statement.
Removed
L90-91. Delete "in a few studies".
Removed
L96. Suggest rephrase to "Here we add to the emerging pool of boron isotope data in planktonic foraminifera…"
Changed
L99. Add "analysis method for small samples (down to ~250 µg…)"
Added
L100-101. End sentence at "planktonic foraminifera". Start new sentence with "The studied sediment core-tops span a range…".

Changed

L120. Important addition: "using d11B data from multiple taxa". "Data" by itself is not specific and should be avoided without context.

Changed

Added

L149-150. Hopefully the third time on this terminology is a charm…. Fractionation factors are not in per mil units. Either change the text to "fractionation" or change "27.2‰" to "1.0272" to make this statement correct.

Worse is that I have looked carefully. I apologize for this. Changed for: "We use a fractionation of 27.2 ‰ from Klochko et al. (2006) in this study."

Changed

L184. Change "it" to "microenvironment pH". Please avoid using pronouns.

Changed

Changed

L228-229. Since you have not yet outlined the approach used in this study, you may wish to save this sentence for the discussion. Or else, add "as described below".

Added: "add described below"

Changed

Changed

L258. How was the clay removed? Methanol? MQ water? Ultrasonication? Please add some brief details.

"Briefly, picked foraminifera were gently cracked open, clay removed with successive ultrasonication steps in MQ water and methanol and then were checked for coarse-grained silicates."

L261-262. Using reductive cleaning because the sites were not studied before is not a sufficient justification for using reductive cleaning. Please state why you used the reductive cleaning. Were you concerned about high Mn/Fe levels? Was this following typical laboratory protocols? (in which case, include reference). I agree with the authors that reductive cleaning should not make a huge difference for B, but it does for Mg (and hence your CDs), and so it needs justification.

"Samples from the South China Sea (sites A14, E035) presented high Mn and high Fe. Due to potential Fe-Mn oxide and hydroxides the reductive cleaning was used. Previous comparisons of cleaning methods have shown there is no impact of the reductive step on B/Ca (Misra et al., 2014b) but there is an impact of the reductive step on Mg/Ca (Barker et al., 2003 and others), nevertheless, it is possible that Fe-Mn oxide and hydroxides can result in non-negligible Mg and B contamination. Because this study was designed to investigate boron proxies and in order to be consistent in methodology, the reductive cleaning was used at all sites. Cleaned samples selected for this study did not yield high Mn concentrations (see supplement for discussion on contamination)."

L264-266. This is out of place as you have not yet defined or discussed CDs, and so this is very confusing to the reader. The answers to reviewers' queries must be in the appropriate section. Please move this text to either Section 3.8 or Section 5.1.1.

Moved to section 3.8 line 344 : "However, we note that reductive cleaning leads to a decrease in Mg/Ca that in turn would result in a bias towards deeper calcification depths (CD), which is not the case when we utilize non-Mg/Ca-based methodologies."

L268-269. Matrix effects should not be addressed in the sample cleaning section. Again, the response to the reviewer comment is out of place. Please move to an appropriate text location.

We haven't moved this sentence to the discussion because we should add another paragraph. We have moved this sentence to section 3.5, line 296 where we are presenting the method.

Changed

L297. Change to "and contributed less than…." (remove unnecessary parentheses)

Parentheses removed

Changed

Removed "- this study)"

L310-311. Also note the JCp-1 values published by Farmer et al. (2016) Chemical Geology here.

Added line 307: "Chemically cleaned JCp-1 samples were measured at 24.06 ± 0.20 (2SD, n=6) and is within error of published values of 24.37 ± 0.32 ‰, 24.11± 0.43 ‰ and 24.42 ± 0.28 ‰ by Holcomb et al. (2015), Farmer et al. (2016) and Sutton et al. (2018) respectively."

Reference also added.

L317-138. For clarity, please make a different sentence; e.g., "Levels of remaining HCl in these diluted samples were negligible and did not contribute to matrix effects".

Changed

L329-331. Are these uncertainties ± 2SD or the absolute 2SD range? For example, is it ±7 µmol/mol for B/Ca 2SD, or 7 µmol/mol B/Ca 2SD (±3.5 µmol/mol)? Please specify.

Uncertainties is "The analytical uncertainties (2SD, n=31, Table S3) on the X/Ca ratios are: ±…"

L336. Again, how was clay removed? If the same procedure was followed as for B analyses, please say so.

"Around 20 shells were weighed, crushed and clay removed following the same method described in section 3.3 (Barker et al., 2003)."

L338-339. Again, are uncertainties ± or absolute ranges?

Uncertainties ±

L342-349. Please define your methods as CD1, CD2, and CD3 in the text. For example, "The first method (CD1) utilizes d18O measurements of the carbonate…"

Added

L353-357. This approach is not clear, please rephrase. What is "that one"? What did you do if literature values were not available? How do you deal with reductive cleaning lowering Mg/Ca here?

Line 352: "To select which calcification depth to use for further calculations, we first looked at $CD_1$, $CD_2$ and $CD_3$. If, CD1 and CD2 were similar we selected this CD, if $CD_1$ and $CD_2$ were different we chose literature values, $CD_3$, when available."

Line 362: "The decrease in Mg/Ca due to reductive cleaning was not taking into account, because it has not been studied for most of the species used in this study and because the depth uncertainty applied based on $\delta^{18}O$ analytical error is conservative relative to the uncertainty of a 10% decrease in Mg/Ca equivalent that would be equivalent to ~1.2°C. "

L369. World Ocean Database (capital first letters).

Changed

L371-372. "Uncertainty… was similar to the one" I am confused, please rephrase. What one? Do you mean to say "Uncertainty… was similar to uncertainty calculated by…"?

Yes, "We utilized the R© code in Henehan et al, (2016) (courtesy of Michael Henehan) to calculate the $\delta^{11}B_{borate}$, $\delta^{11}B_{borate}$ uncertainty and derive our calibrations. Uncertainty for $\delta^{11}B_{borate}$ utilizing the Henehan's code was similar to uncertainty calculated by applying 2 standard deviations of the $\delta^{11}B_{borate}$ profiles within the limits imposed by our calcification depth."

L377. What is k? Please state/clarify for the reader.

"with k (number of wild bootstrap replicates) equal to 500."

Results.
L382-392. I still would like to see a small figure or a table in the main text with the calcification depths. But perhaps other reviewers feel differently.

It was too redundant for the anonymous reviewer.

L389. What are "both approaches"? Be specific.

"Data from both approaches (CD1 and CD2)  implies…"

L393-394. Which two calcification depth determination methods? You have three…

"Some differences are observed between the two methods for calcification depth determination that are based on $\delta^{18}O$ and Mg/Ca (CD1 and CD2, respectively).

L403. Change to "where G. ruber shells". Greater specificity would make this far easier to read.

Changed

L412. "by the high photosynthetic activity"…. Of what? Presumably of G. ruber symbionts? Please specify.

Changed

L414-417. Rephrase. "the uncertainty is significant given limited data in our study, and given this large uncertainty, our sensitivity of d11B_carbonate to d11B_borate is also consistent with the low sensitivity rend of culture experiments from…"

Added

L419. Missing a closing parentheses

Added

L423-426. This response to a reviewer comment should be included in the discussion. Here you should only present results, and you should not discuss why your data may (or may not) be different.

Moved to line 636

L437. Change to "When regressing data from the 250-400 µm size fraction, ..."

Changed

L449. "the points are in good agreement" what does that mean? What are points? Be specific.

"The range of $\delta^{11}B_{borate}$ from the samples we have of *G. menardii* and *G. tumida* is not sufficient to derive calibrations, but the $\delta^{11}B_{carbonate}$ measured for those species are in good agreement with the *N. dutertrei* calibration and Henehan et al. (2016) calibration for *O. universa*."

L455-457. This statement is an interpretation of results and should be moved to the discussion.

Moved to line 641

Discussion

L511-512. This wording is awkward to me. I am not sure it is that "seasonality can have a major impact on hydrographic carbonate parameters", as the authors state (Indeed, Gutjahr et al. (2018) argue otherwise). Instead, hydrographic parameters related to carbonate chemistry may change across seasons at a given water depth.

"As discussed by Raitzsch et al. (2018), depending of the study area, foraminiferal fluxes can change throughout the year. Hydrographic parameters related to carbonate chemistry may change across seasons at a given water depth."

L541. "and in previous studies (reference list)"

"Foraminifera with high photosynthetic activity and symbiont density, such as *G. ruber* and *T. sacculifer,* are expected to have a microenvironment pH higher than ambient seawater, and a $\delta^{11}B_{carbonate}$ higher than expected $\delta^{11}B_{borate}$, which is the case in our study and in previous studies (Foster et al., 2008, Henehan et al., 2013, Raitzsch et al., 2018)."

L544. What do you mean by "mixed-dweller"? Do you mean "mixed-layer dwelling"? Please clarify.

We changed for "mixed-layer dwelling", this is what we meant

L549. Comma use and grammar editing: "…photosynthetic activity, which may explain why P. obliquiloculata exhibited the lowest microenvironment pH as recorded by d11B."

Changed

L563. Do you mean d11B? If Δ11B is correct here, please define what Δ11B means.

"Additionally, the $\Delta^{11}B$ ($\Delta^{11}B = \delta^{11}B_{carbonate} - \delta^{11}B_{borate}$) of *G. ruber*, *T. sacculifer* (w/o sacc and sacc) is significantly lower in the WEP compared to the other sites (p<0.05)."

L573-577. This is still too confusing to be published as-is. How can a lower size fraction induce a change in photosynthetic activity? Do you not mean that a lower size fraction may have had lower photosynthetic activity and hence d11B?

Regardless, please change the final sentence to "Unfortunately, no weight per shell data were determined on foraminifera samples to constrain whether test size was significantly different across sites. Future studies could use shell weights to test these relationships."

Line 574: "It can also be noted that *T. sacculifer* exhibits the largest variation in symbiont density versus test size (Takagi et al., 2019), suggesting that lower size fraction reported for the WEP (250-400 µm) compared to the 300-400 µm at the other sites can be related to a decrease in photosynthetic activity and a lower $\delta^{11}B$. Unfortunately, no weight per shell data were determined on foraminifera samples to constrain whether test size was significantly different across sites. Future studies could use shell weights to test these relationships."

L583-584. Rephrasing: "which is supported by observation of a lower photosynthetic potential in O. universa than in T. sacculifer (Tagaki et al., 2019)."

Changed

L605-611. Add "In inorganic calcite, d11B_carboate and B/Ca have been shown to be sensitive…." You must specify that these results are for inorganic calcite. Also, and I swear this is not personal bias, but Farmer et al. (2019) is missing from the references.

Added

L644. Change "lighter" to "lower"

Changed

Changed

"The comparison of the two methods, first using all the data to derive the calibration and recalculate pH and $pCO_2$ (circular) and the second by excluding the site of interest, derive calibrations and calculate pH and $pCO_2$ (not circular), does not show significant differences and validates the robustness of the calibrations (Fig. S5)."

L661-667. This section is difficult to follow as written; please reword.

We have reworded one paragraph of the discussion:

Line 682: "In order to accurately develop downcore reconstructions, constraining the depth habitat using core-tops studies is important, as a same species can record the seawater pH at different water depth potentially introducing biases when comparing between different locations. Also, we speculate that a change of the thermocline depth in the past could imply variations of depth habitat and introduce biases in the reconstructions but further work is needed to test this assertion. "

Figure 1. To repeat myself once again, this figure is unnecessary and should be removed.

Removed

Figure 2. Change chemical formulae on figure to $B(OH)_3$ and $B(OH)_4^-$ for consistency with text.

Changed

Table 1. There is no such thing as a $\Delta 14C$ age (unless you mean projection ages ala Adkins, and I doubt that you do). Please change to 14C. Also, some of these ages are in years, while others (Pacific Ocean) are in kyr. Please make consistent.

Changed

Table 2. You could make this table smaller by having separate uncertainty columns, and sharing one uncertainty column for d11B_C1 and d11B_C2 (as the uncertainties are exactly the same)

We found it more explicit like this event if redundant.

**Résultats de la comparaison**

| Ancien fichier : | | Nouveau fichier : |
|---|---|---|
| **Maxence Guillermic coretop manuscript - 03092020- clean version.pdf** | par rapport à | **Maxence Guillermic coretop manuscript - 04282020 clean version.pdf** |
| **31 pages (736 KB)** | | **31 pages (702 KB)** |
| 3/9/2020 9:14:32 AM | | 4/28/2020 2:01:20 AM |

**Total des modifications**

**1156**

Comparaison du texte seulement

**Contenu**

remplacements insertions suppressions

**Styles et annotations**

style annotations

Consulter la 1re modification (page 2)

[revised manuscript text omitted]